# IL-1R signaling drives enteric glia-macrophage interactions in colorectal cancer

Lies van Baarle [1,14], Veronica De Simone[1,14], Linda Schneider [2,14], Sneha Santhosh[1,3], Saeed Abdurahiman[1], Francesca Biscu [1,4], Reiner Schneider [2], Lisa Zanoletti [1,5], Renata Siqueira de Mello [1], Sara Verbandt[6], Zedong Hu[6], Michelle Stakenborg[1], Bo-Jun Ke[1], Nathalie Stakenborg [7], Raquel Salvador Laureano[8], Balbina García-Reyes[2,9], Jonas Henn [2], Marieta Toma[10], Maxime Vanmechelen[11,12], Guy Boeckxstaens [7], Frederik De Smet [11,12], Abhishek D. Garg [8], Sales Ibiza[13], Sabine Tejpar [6], Sven Wehner [2,15] ✉ & Gianluca Matteoli [1,12,15] ✉

Enteric glia have been recently recognized as key components of the colonic tumor microenvironment indicating their potential role in colorectal cancer pathogenesis. Although enteric glia modulate immune responses in other intestinal diseases, their interaction with the colorectal cancer immune cell compartment remains unclear. Through a combination of single-cell and bulk RNA-sequencing, both in murine models and patients, here we find that enteric glia acquire an immunomodulatory phenotype by bi-directional communication with tumor-infiltrating monocytes. The latter direct a reactive enteric glial cell phenotypic and functional switch via glial IL-1R signaling. In turn, tumor glia promote monocyte differentiation towards pro-tumorigenic SPP1+ tumor-associated macrophages by IL-6 release. Enteric glia cell abundancy correlates with worse disease outcomes in preclinical models and colorectal cancer patients. Thereby, our study reveals a neuroimmune interaction between enteric glia and tumor-associated macrophages in the colorectal tumor microenvironment, providing insights into colorectal cancer pathogenesis.

Identified as the world's third most common tumor, colorectal cancer (CRC) represents one of the preeminent causes of cancer-associated deaths worldwide. Although innovative technologies have significantly impacted the diagnosis and treatment of CRC, patients with advanced disease still have a poor prognosis. In fact, while the 5-year survival rates of patients with early-stage CRC can reach up to 90%, the survival rate plummets dramatically to as low as 10% for patients diagnosed with metastatic disease[1]. Hence, a better understanding of the pathogenesis of CRC is crucial to develop advanced therapeutic strategies along with advanced patient stratification for precision medicine. CRC consists of rapidly evolving neoplasms where acquired mutations in

oncogenes and tumor-suppressor genes lead to increasing complexity of the tumor microenvironment (TME), unleashing interaction of the tumor cells with the stroma and the immune system, including fibroblasts, tumor-infiltrating immune cells, and cells of the enteric nervous system[2,3]. This process contributes to the formation of a complex network of cell types within the TME, which is leading to increased tumor fitness.

In recent years, enteric glial cells (EGCs) have also been identified as a constituent of the colon carcinoma microenvironment[4,5]. EGCs, once regarded as merely supportive and accessory cells for neurons within the enteric nervous system[6], have now gained increased

---

attention for their more complex roles in both health and disease[7]. In homeostasis, EGCs regulate intestinal reflexes and support neurotransmission via communication with enteric neurons. However, accumulating evidence highlights EGCs as crucial mediators of interactions not only among enteric neurons but also intestinal epithelium, enteroendocrine cells, and immune cells[7–10]. Of particular interest is their significant role in modulating immune responses in various intestinal diseases[11–14]. Being highly responsive to inflammatory mediators, including Adenosine triphosphate (ATP), Interleukin (IL)−1 cytokines, or Lipopolysaccharides, EGCs are rapidly activated during intestinal diseases. Upon activation during intestinal pathologies, EGCs contribute to the shaping of the inflammatory milieu through the secretion of a plethora of cytokines and chemokines[15–18]. In this regard, we recently demonstrated the profound influence EGCs exert on macrophage dynamics in the setting of acute intestinal inflammation, promoting the recruitment of monocytes and their differentiation into pro-resolving macrophages[17,19].

So far, in the context of CRC, a few studies suggested that EGCs exert a pro-tumorigenic effect during tumor development[4,5]. In a study by Yuan et al., glial cell depletion led to reduced tumor burden in a CRC mouse model[5], indicating a central role of EGCs in CRC development. A xenograft model confirmed this role, and in vitro work suggested that EGC activation by IL-1 resulted in a pro-tumorigenic EGC phenotype[4], pointing to a direct interaction of EGCs with the TME. However, the mechanisms by which EGCs interact with the different components of the colorectal cancer TME to exert their pro-tumorigenic role remain poorly understood. Especially the molecular and cellular communication pathways involved are so far insufficiently explored and display a substantial lack of in vivo evidence.

In this work, we demonstrate, using in vitro and in vivo models, that upon exposure to the colorectal TME, EGCs undergo a reactive phenotypic switch, leading to the activation of immunomodulatory processes that promote the differentiation of tumor-associated macrophages (TAMs). Tumor-infiltrating monocytes are found to influence the phenotype and function of CRC EGCs through IL-1 signaling. In turn, EGC-derived IL-6 promotes the differentiation of monocytes towards SPP1[+] TAMs. This IL-1R/IL-6 axis is found to be essential for the tumor-supportive functions of EGCs. Our findings uncover a critical neuroimmune interaction in the colon cancer microenvironment, potentially facilitating the development of additional therapeutic approaches to treat this devastating disease.

## Results

### EGCs shape the CRC immune compartment
Recent studies identified EGCs as an important component of the colon TME[4,5]. However, their contribution to CRC pathogenesis and their possible interaction with the tumor immune compartment remains largely unexplored. Hence, to study the immunomodulatory role of EGCs in CRC in vivo, we utilized an established murine model[20], in which MC38 colorectal cancer cells were orthotopically injected into the colonic submucosa (Fig. 1a and Supplementary Fig. 1a). This model is characterized by a strong immune cell infiltration and tumor epithelial cell proliferation (Supplementary Fig. 1b, c). After initial tamoxifen exposure, diphtheria toxin (DT) was administered via colonoscopy-guided injections into the colonic wall of PLP1[CreERT2]iDTR mice on days −5 and −3 prior to the submucosal injection of MC38 tumor cells (Fig. 1b). This approach enabled the temporal and localized depletion of enteric glia during the development of colon tumors (Supplementary Fig. 1d–h). Of note, the DT-triggered glial cell death by d0 was not associated with any significant differences in immune infiltrate between vehicle-treated and DT-treated mice (Supplementary Fig. 1i). Seven days after colonic MC38 cells injection, a significant reduction of tumor size was observed in DT pre-treated mice compared to the vehicle group (Fig. 1c). Tumors treated with DT exhibited a sustained reduction in glial marker

expression at day 7, with no impact on colon length (Supplementary Fig. 1j–l). Although absolute numbers of both immune and non-immune cells per milligram of tumor tissue decreased within the tumor microenvironment in the absence of glial cells, the composition of the tumor microenvironment remained unchanged, as evidenced by the consistent percentage of immune and non-immune cells. Thus, it is evident that the reduction in tumor size is not solely attributable to a decline in immune cells (Supplementary Fig. 1m, n). To address any potential non-specific DT toxicity or unspecific immune activation, we subjected C57BL/6J mice to the same injection protocol used for PLP1[CreERT2]iDTR mice. Our findings revealed no difference in tumor growth or immune infiltration between DT- and saline-injected WT mice, excluding non-specific DT toxicity in this model (Supplementary Fig. 1o–q).

Interestingly, during this early phase of tumor growth, the depletion of EGCs resulted also in fewer TAMs (Fig. 1d and Supplementary Fig. 2a). Furthermore, a decrease in the numbers of monocytes and eosinophils was observed in tumors after enteric glia depletion, whereas no differences were observed for neutrophils, T and B cells (Fig. 1d and Supplementary Fig. 2b).

To further validate the effect of EGCs on the CRC immune compartment, we established a co-injection model of MC38 cells together with primary EGCs in the colonic mucosa of C57BL/6J mice (Fig. 1e). Co-injection of MC38 and EGCs resulted in increased tumor growth associated with higher numbers of TAMs, as well as CD4[+] T cells, CD8[+] T cells, and T$_{reg}$ cells compared to mice orthotopically injected with MC38 cells alone (Fig. 1f, g and Supplementary Fig. 2c). No differences were observed in the numbers of monocytes, eosinophils, neutrophils, and B cells (Fig. 1g and Supplementary Fig. 2c). Spatial tissue mapping via confocal microscopy confirmed proximity between EGCs (GFAP[+]) and TAMs (F4/80[+]) within orthotopic colonic tumors injected in C57BL/6J mice (Fig. 1h), further suggesting the existence of a glial-macrophage interplay within the TME.

Overall, these findings suggest that EGCs participate in shaping the CRC immune microenvironment, by expanding the TAM population.

### EGCs display an immunomodulatory phenotype in CRC
To examine the molecular mechanisms by which EGCs influence the immune CRC compartment with particular regard to the TAMs, we first investigated their transcriptional adaptations upon CRC onset. To this end, we established an in vitro tumor EGC model, able to mimic the response of enteric glia to the factors secreted by the colonic TME. Primary embryonic-derived EGCs were treated with conditioned medium (CM) of digested murine MC38 orthotopic tumors, from now onwards, defined as TME-CM EGCs (Fig. 2a). Control groups consisted of EGCs stimulated with supernatants derived from healthy colonic tissue (H-CM) and naive unstimulated EGCs. At 6 h, 12 h, and 24 h post-stimulation, bulk RNA sequencing (RNA-seq) was performed to uncover the transcriptional differences among the various EGC groups. Principal component analysis (PCA) demonstrated a significant similarity between the H-CM and unstimulated EGC samples (Fig. 2b). In contrast, TME-CM EGCs exhibited a distinct separation from both H-CM and unstimulated EGCs, suggesting a noticeable difference in their transcriptional programs. At 24 h post-stimulation, the segregation of the samples was highly driven by the upregulation of several chemokines, cytokines and typical markers for pan-reactive astrocytes (*Lnc2* and *Timp1*)[21], suggesting an activated and immunomodulatory role for EGCs within the tumor microenvironment (Supplementary Fig. 3a, b).

Next, using weighted gene correlation network analysis (WGCNA) we identified 12 gene co-expression modules. TME-CM EGCs showed a specific correlation to modules 7 and 8 and an inverse correlation to module 4 (Fig. 2c and Supplementary Data 1). Module 7 showed a functional association with several pro-tumoral processes and

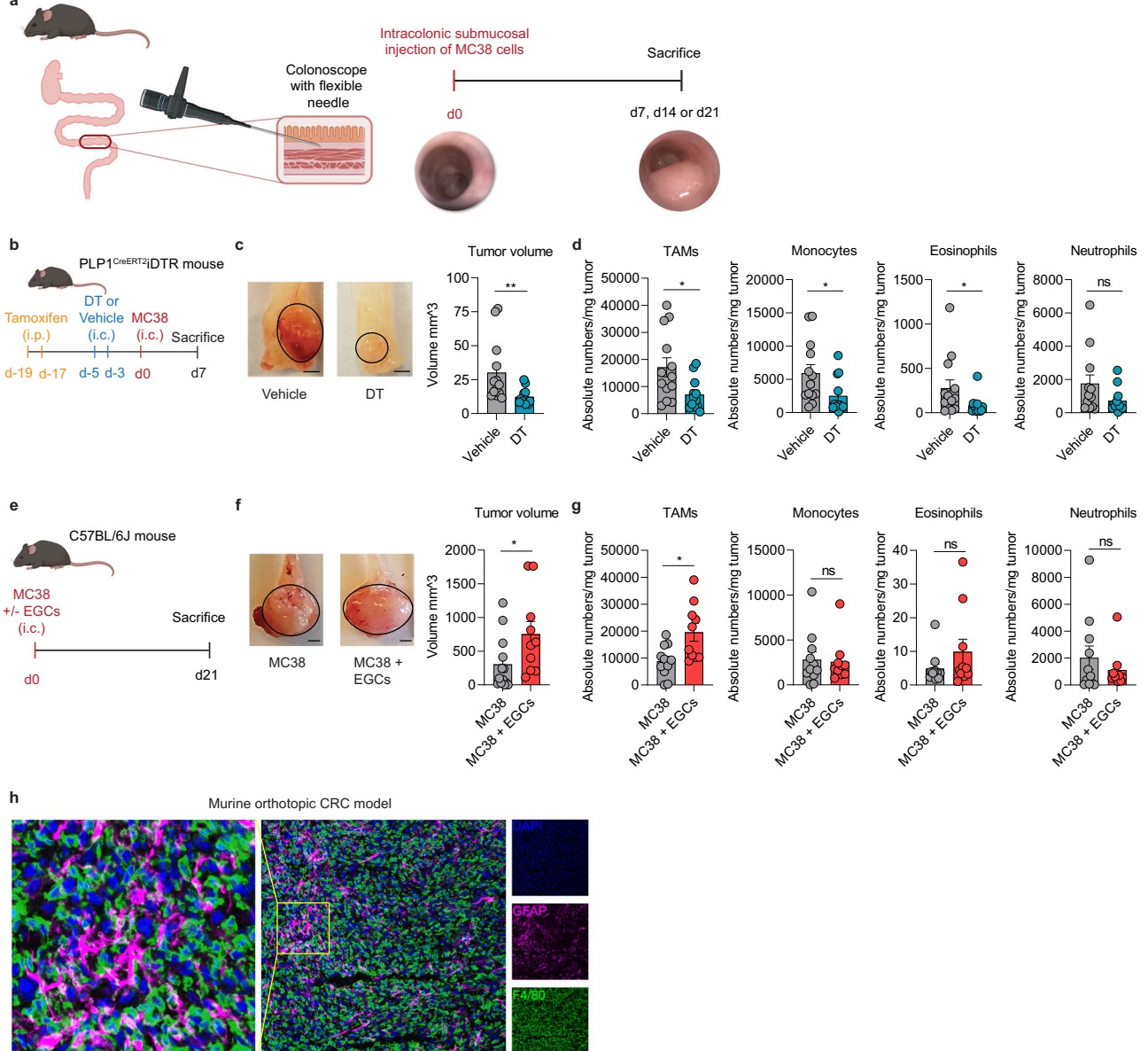

**Fig. 1 | EGCs shape the CRC immune compartment. a** Schematic representation of the murine orthotopic CRC model. Adult mice were injected endoscopically in the colonic submucosa with MC38 cells and tumors were assessed at day (d)7, d14 or d21. **b–d** PLP1$^{CreERT2}$iDTR mice were intraperitoneally (i.p.) injected with tamoxifen at d-19 and d-17, followed by intracolonic (i.c.) injection at d-5 and d-3 with 40 ng Diphtheria toxin (DT) or saline (Vehicle). At d0, MC38 cells were i.c. injected in both groups. Tumor growth and immune infiltration were assessed at d7. Schematic representation of EGCs depletion mouse model (**b**) with representative pictures (scale bar 2 mm) and quantitative comparison of tumor volume (**c**). Data show absolute tumor-infiltrating myeloid immune cell numbers per mg tumor tissue (**d**) ($n = 13$ Vehicle, $n = 12$ DT). **e–g** WT C57BL/6J mice were i.c. injected with

MC38 cells with or without embryonic neurosphere-derived EGCs (1:1 ratio). Tumor growth and immune infiltration were assessed at d21. Schematic representation of EGCs co-injection mouse model (**e**) with representative pictures (scale bar 2 mm) and quantitative comparison of tumor volume (**f**). Data show absolute tumor-infiltrating myeloid immune cell numbers per mg tumor tissue (**g**) ($n = 11$ MC38, $n = 10$ MC38 + EGCs). **h** Immunostaining of orthotopic murine tumor sections showing GFAP (magenta), F4/80 (green) and DAPI (blue) (scale bar 70 μm and 25 μm) representative of 4 independent experiments. Data show mean ± SEM (**c, d, f, g**). Statistical analysis: unpaired two-tailed Mann-Whitney test (**c, d, f, g**) *$p < 0.05$, **$p < 0.005$, ns not significant. Source data and exact $p$ values are provided as a Source Data file.

contained the glial reactivity markers *Lcn2* and *Timp1* (Fig. 2d and Supplementary Fig. 3c). Module 8 consisted of genes, associated with immunomodulatory functions of EGCs, such as "myeloid cell differentiation", "macrophage activation" and "myeloid leukocyte migration" (e.g., *Ccl2, Cxcl1, Cxcl10* and *Il6*). Conversely, the genes of module 4, such as *Ntsr1* and *Sparcl1*, were associated with the homeostatic functions of EGCs[22]. In line, gene set enrichment analysis of the 24 h TME-CM EGCs signature revealed impairment for functions previously ascribed to healthy EGCs, including GO terms like "*Positive regulation of stem cell differentiation*", "*Regulation of glial cell differentiation and*

*gliogenesis*", "*Neuron projection guidance*", and "*Positive regulation of neurogenesis*"[7,23] (Fig. 2e and Supplementary Data 2). Notably, in line with the previous in vitro findings of Valès et al., TME-CM EGCs were enriched for the GO terms "*Positive regulation of prostaglandin biosynthetic process*", and "*Interleukin 1 receptor activity*", supporting the possible paracrine effect of IL-1/PGE$_2$ signaling[4]. Lastly, gene set enrichment analysis predicted a direct interaction of CRC EGCs with TAMs, reflected by functional enrichment for the GO terms "*Macrophage differentiation*" and "*Positive regulation of macrophage activation and migration*" (Fig. 2e).

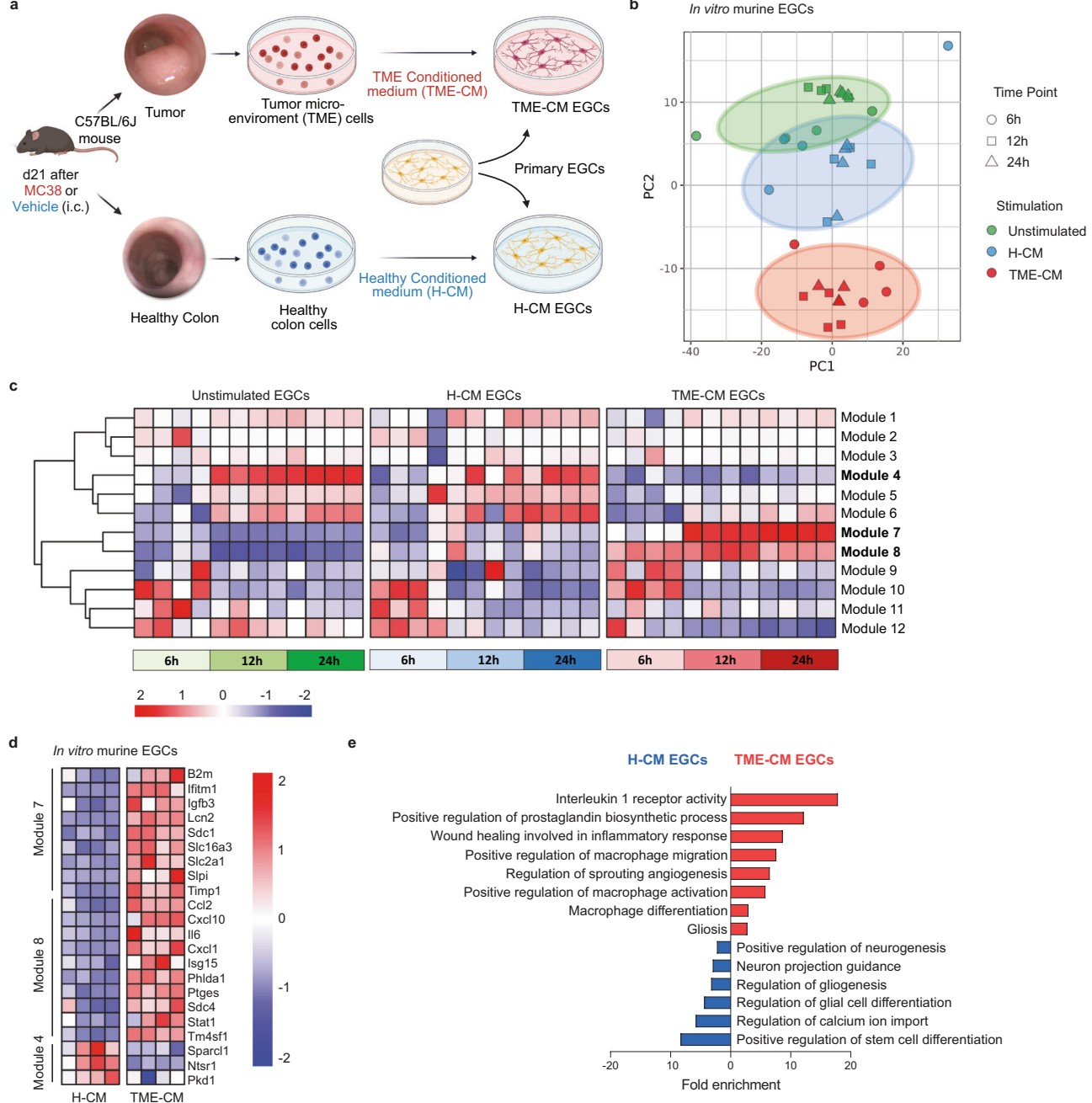

**Fig. 2 | EGCs display an activated and immunomodulatory phenotype in CRC.** Transcriptome analysis of in vitro primary embryonic neurosphere-derived EGCs alone or stimulated with healthy conditioned medium (H-CM) or tumor microenvironment conditioned medium (TME-CM) at different time points (6 h, 12 h, and 24 h, *n* = 4). **a** Schematic representation of the in vitro tumor EGCs model. **b** Principal component analysis (PCA) plot of EGCs gene signature identified by 3′mRNA bulk RNA-seq. Each dot represents an individual sample. **c** Heatmap showing the transcriptional modules identified by weighted gene correlation network analysis (WGCNA). **d** Heatmap of differentially expressed genes in modules 4, 7, and 8 of in vitro murine EGCs stimulated for 24 h with H- or TME-CM. **e** Gene set enrichment analysis for the differentially up- and down-regulated genes in TME-CM versus H-CM stimulated EGCs after 24 h (*n* = 4). Source data are provided as a Source Data file.

Taken together, upon exposure to the CRC TME, EGCs undergo a phenotypic switch associated with the activation of immunomodulatory programs related to macrophage interplay.

**Tumor EGC-derived IL-6 favors SPP1⁺ TAM differentiation**

Considering that tissue location and transcriptomic data suggest direct communication between EGCs and TAMs in the CRC microenvironment, we aimed to decipher the molecular mechanisms underpinning their interaction. Firstly, single-cell transcriptomic analysis was used to characterize the immune landscape of colorectal MC38 orthotopic tumors (Supplementary Fig. 4a). Interestingly, among the identified immune populations, monocytes and macrophages accounted for 60% of the tumor-infiltrating immune cells. Unsupervised clustering of the myeloid cells (*Lyz2, Cd68, H2-Ab1, Mrc1, C1qa, Ly6c2, Ccr2,* and *Fn1*) revealed 1 monocyte and 4 distinct macrophage sub-populations (Fig. 3a-b). The most abundant macrophage cluster was characterized by simultaneous marker gene expression for monocytes (*Ccr2* and *Ly6c2*) and for differentiated macrophages (*H2-*

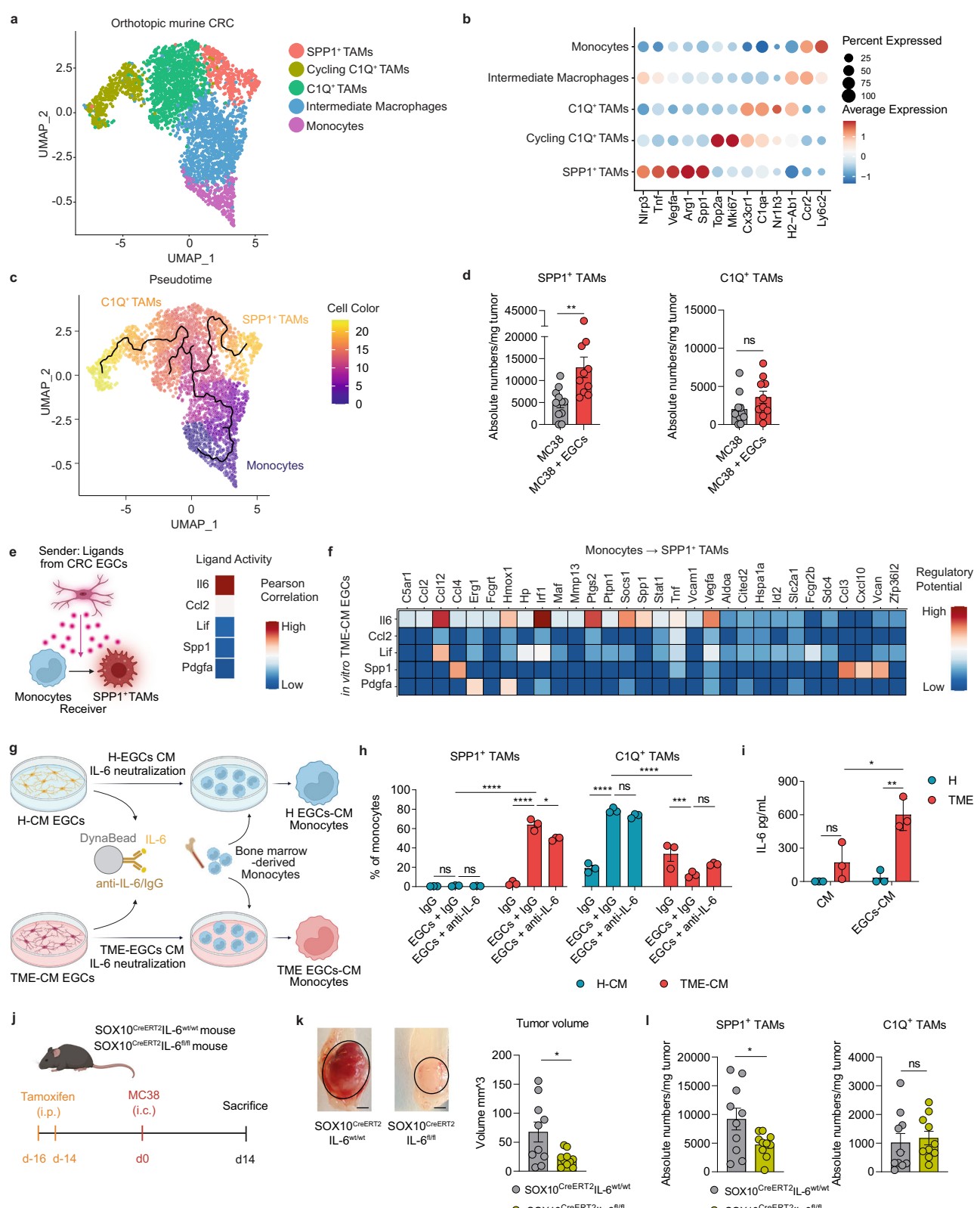

*Ab1* and *Nlrp3*), indicating a possible transitional cell state, which we termed "Intermediate Macrophages" (Fig. 3b). Additionally, we identified two clusters of TAMs, "SPP1+ TAMs" and "C1Q+ TAMs". The "SPP1+ TAMs" co-express *Spp1* and *Arg1*, along with genes involved in angiogenesis (*Vegfa*) and extracellular matrix remodeling (*Spp1* and *Tnf*) (Fig. 3b and Supplementary Fig. 4b). The "C1Q+ TAMs" are characterized by genes involved in phagocytosis (*Nr1h3*), antigen

presentation (*H2-Ab1*) and the complement cascade (*C1qa*). A second of C1Q+ TAM cluster, distinguished by high expression of cell cycle genes such as *Mki67* and *Top2a*, was classified as 'Cycling C1Q+ TAMs'.

Overall, our findings are in line with the study of Zhang et al., which reported very similar dichotomous functional phenotypes of TAMs in CRC patients[24]. Additionally, Zhang and colleagues predicted a dichotomic differentiation trajectory of monocytes towards SPP1+

**Fig. 3 | Tumor EGC-derived IL-6 favors SPP1⁺ TAM differentiation. a-c** Analysis of monocytes and macrophages from the scRNA-seq data of orthotopic MC38 tumors in WT C57BL/6J mice, 21 days(d) after tumor induction (n = 3). UMAP of sub-clustering (**a**), dot plot of differentially expressed marker genes (**b**) and differentiation trajectory (**c**). **d** WT C57BL/6J mice were intracolonically injected with MC38 cells ± embryonic neurosphere-derived EGCs (1:1 ratio). SPP1⁺ TAMs and C1Q⁺ TAMs infiltration was assessed on d21. Data represent absolute numbers/mg tumor (n = 11 MC38, n = 10 MC38 + EGCs). **e, f** Nichenet analysis was performed considering ligands expressed by 24 h tumor microenvironment-conditioned medium (TME-CM) EGCs, (CRC EGCs bulk RNAseq; Fig. 2) and considering the differentially expressed genes between monocytes and SPP1⁺ TAMs as target genes (orthotopic murine CRC dataset; Fig. 3a). Pearson correlation of top TME-CM EGCs-released ligands predicted to induce monocytes to SPP1⁺ TAM differentiation (**e**). Heatmap showing regulatory potential scores between top EGC-released ligands and their target genes differentially expressed between monocytes and SPP1⁺ TAMs (**f**). **g–h** Supernatant of healthy (H)-CM, TME-CM, H EGCs-CM, and TME EGCs-CM were incubated with IgG or anti-IL-6 (both 5 μg/mL) along with Dynabeads™ Protein G followed by removal of the protein-antibody-bead complex. Murine monocytes were cultured for 48 h with the indicated supernatants (n = 3 H-CM and TME-CM). Experimental design (**g**). Percentages of SPP1⁺ and C1Q⁺ TAMs in monocyte cultures after stimuli (**h**). **i** IL-6 concentration in H-CM, TME-CM, H-EGCs-CM, and TME-EGCs-CM (n = 3 H-CM and TME-CM). After tamoxifen treatment were SOX10^CreERT2IL-6^wt/wt and SOX10^CreERT2IL-6^fl/fl mice intracolonic (i.c.) injected with MC38 cells. Tumor growth and immune infiltration were assessed at d14 (n = 10). Schematic representation (**j**), representative pictures (scale bar 2 mm) and comparison of tumor volume (n = 10) (**k**). Tumor-infiltrating TAM numbers/mg tumor (n = 10) (**l**). Data are presented as mean ± SEM (**d, h, i, k, l**). Statistical analysis: unpaired two-tailed Mann Whitney test (**d**), two-way ANOVA with correction for multiple comparisons (**h, i**), unpaired two-tailed t-test (**k, l**). *p < 0.05, **p < 0.005, ***p < 0.0005, ****p < 0.00005, ns not significant. Source data and exact p values are provided as a Source Data file.

TAMs or C1QC⁺ TAMs in CRC patients. In line, we identified a strong directional flow from tumor-infiltrating monocytes towards intermediate macrophages, which in turn further branched into two opposite paths, ending either in SPP1⁺ TAMs or C1Q⁺ TAMs (Fig. 3c).

SPP1⁺ macrophages represent a significant cell population within the tumor immune cell compartment with negative prognostic value in CRC[24,25]. However, the current knowledge regarding microenvironmental cues, promoting the differentiation of tumor-infiltrating monocytes towards SPP1⁺ TAMs or C1Q⁺ TAMs, remains limited. Thus, we explored the possible role of EGCs in promoting monocyte to SPP1⁺ TAM or C1Q⁺ TAM differentiation using our EGC co-injection CRC model (Fig. 1e). Strikingly, supplementation of EGCs within the CRC TME resulted in more than a twofold increase in SPP1⁺ TAMs, while no significant difference was found for C1Q⁺ TAMs (Fig. 3d and Supplementary Fig. 4c).

Next, to identify key EGC-derived mediators accountable for SPP1⁺ TAM differentiation in CRC, we applied NicheNet, a computational tool designed to infer relationships between signaling molecules and their target gene expression[26]. By using the genes differentially expressed between SPP1⁺ TAMs and monocytes [data extracted from our single-cell RNA-seq (scRNAseq) murine orthotopic CRC dataset] as target genes, we prioritized candidate ligands derived from TME-CM EGCs (data extracted from bulk RNAseq in vitro CRC EGCs dataset) as potential drivers of this differentiation process (Fig. 3e). Here, TME-CM EGC-derived IL-6 was identified as the top predicted candidate factor involved in driving the SPP1⁺ TAM phenotype (Fig. 3e, f).

In line with this prediction, IL-6 neutralization in the TME-CM EGC supernatant attenuated the differentiation of monocytes into SPP1⁺ TAMs, further reflected by reduced Spp1 and Arg1 expression (Fig. 3g, h and Supplementary Fig. 4d). Of note, TME-CM EGCs did not promote C1Q⁺ TAM differentiation from monocytes. In contrast, H-CM EGCs had no effect on SPP1⁺ TAM polarization while they promoted C1Q⁺ TAM differentiation in an IL-6 independent fashion (Fig. 3h and Supplementary Fig. 4d). Consistent with this, high level of IL-6 could be measured only in the supernatant of TME-CM EGCs, with almost undetectable level in H-CM EGCs (Fig. 3i).

To finally investigate the effect of EGC-derived IL-6 on the SPP1⁺ TAM differentiation and tumor development in vivo, we orthotopically injected MC38 cells in the colonic mucosa of mice with a glial-specific deletion of IL-6 (SOX10^CreERT2IL-6^fl/fl) and their littermates (SOX10^CreERT2IL-6^wt/wt) (Fig. 3j). Strikingly, glial-specific IL-6 deletion led to a reduced tumor growth, which correlated with a reduction in SPP1⁺ TAMs while no effect was detected for C1Q⁺ TAMs (Fig. 3k-l). These results underline the importance of glial-derived IL-6 in both colonic tumor development and TAM differentiation in vivo. Glial reporter mouse line SOX10^CreERT2Ai14^fl/fl showing co-localization of tdTomato signals together with GFAP immunostaining in glial cells, was used to confirm enteric glia targeting specificity (Supplementary Fig. 4e).

Altogether, our data reveal a significant and previously unknown interaction between EGCs and TAMs in the CRC TME, with EGC-derived IL-6 acting as a crucial regulator in driving SPP1⁺ TAM differentiation.

## Monocyte-derived IL-1 promotes the CRC EGC phenotype

Given the evidence that EGCs modulate their functions based on microenvironmental cues[12,15,19], we aimed to define the factors driving the EGC phenotypic switch in CRC. Considering the heavy infiltration of the colon TME by immune cells, which recent studies have pinpointed as sources of EGC-activating factors[12], we investigated the possibility of immune cells driving CRC EGC transition, potentially creating a reinforcing neuroimmune feedback loop. To investigate the cellular circuits coordinating this interaction we used again NicheNet, linking ligands derived from immune cells within the TME (from the scRNAseq murine orthotopic CRC dataset) to target genes differentially expressed between H-CM EGCs and TME-CM EGCs (data extracted from Bulk RNAseq in vitro CRC EGCs dataset) (Fig. 4a). In this analysis, IL-1β and IL-1α emerged as the top-ranked ligands driving the transcriptional transition from H-CM EGCs to TME-CM EGCs (Fig. 4a, b). Notably, we could confirm elevated levels of IL-1 in the TME samples compared to healthy colon cells, both at RNA and protein levels (Fig. 4c and Supplementary Fig. 5a). Next, we utilized primary EGC cultures derived from embryonic or adult neurospheres to analyze their response to IL-1 treatments. Of note, both EGC culture systems were characterized by strong enrichment of EGC markers (Plp1, Sox10, and S100b) with minimal expression of markers for other cell types such as epithelial cells (Epcam and Cdh1), smooth muscle cells (Prkg1 and Foxp2), endothelial cells (Cdh5 and Pecam1), neurons (Nefl and Syp), mesenchymal cells (Pdgfra and Wt1), and immune cells (Ikzf1 and Itgam) (Supplementary Fig. 5b). Consistent with our prediction, treating primary EGCs with IL-1β markedly activated the CRC EGC phenotype, with induction of Lcn2 and Timp1, and immunomodulatory factors Ccl2 and Il6, both at RNA and protein level, highlighting a specific molecular EGC response independent of embryonic or adult neurosphere origin (Fig. 4d, e, Supplementary Fig. 5c, d and Supplementary Data 2). Conversely, inhibiting IL-1R signaling in EGCs during stimulation with the TME-CM, completely abrogated the induction of the CRC EGC key markers (Fig. 4f, g). Taken together, these findings underscore the pivotal role of IL-1 in the reprogramming of EGCs upon exposure to the colonic TME.

Next, to identify the cellular source of IL-1 within the TME, we quantified IL-1 expression across epithelial, stromal, and immune cells (Fig. 4h and Supplementary Fig. 5e–g). While Valès et al.[4] concluded that in vitro IL-1 is released by the tumor epithelial cells, we herein demonstrate that in vivo, in particular within the MC38 orthotopic model, the TME IL-1 secretion is restricted to myeloid cells, while no expression could be found in epithelial, nor in stromal cells (Fig. 4h and Supplementary Fig. 5g). Further analyses identified tumor monocytes

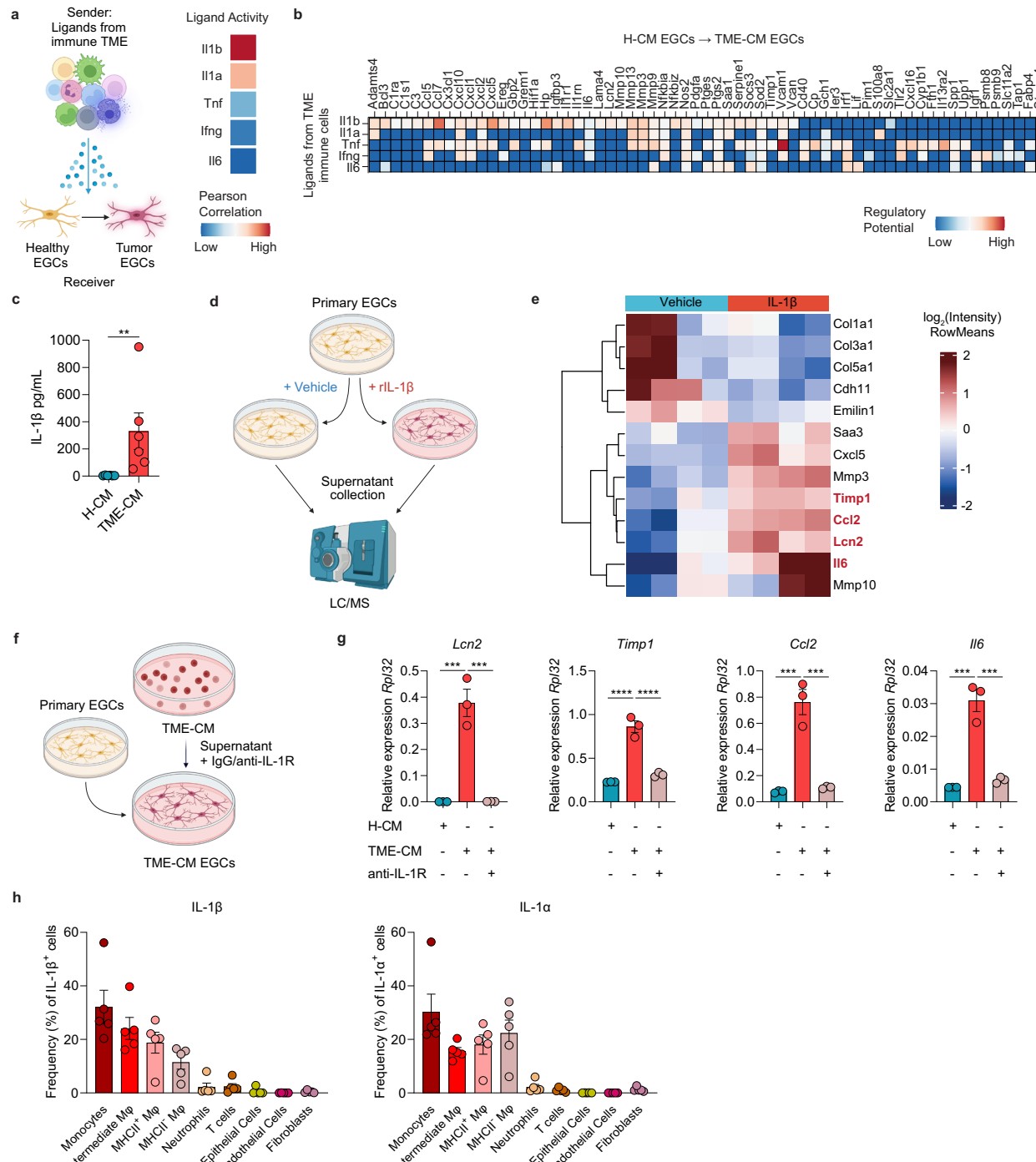

**Fig. 4 | IL-1R activation drives the CRC EGC phenotype. a, b** Top ligands from orthotopic CRC tumor-infiltrating immune cells predicted by NicheNet to be inducing CRC EGC signature. NicheNet analysis was performed considering the ligands expressed by murine orthotopic CRC tumor-infiltrating immune cells, data extracted from CRC orthotopic murine CRC dataset (see Supplementary Fig. 4a) and considering the differentially expressed genes between 24 h healthy conditioned medium (H-CM) EGCs and tumor microenvironment (TME)-CM EGCs as target genes, data extracted from in vitro CRC EGCs bulk RNAseq dataset (see Fig. 2). Schematic representation of TME-derived ligand-EGC interplay (left) and top 5 predicted ligands with their Pearson correlation (right) (**a**). Heatmap of ligand-target pairs showing regulatory potential scores between top ligands and target genes among the differentially expressed genes between in vitro H-CM EGCs and TME-CM EGCs (**b**). **c** Protein level of IL-1β in H-CM and TME-CM (*n* = 6 H-CM and TME-CM). **d, e** Primary adult neurosphere-derived EGCs were isolated from WT C57BL/6J mice and treated with or without recombinant (r)IL-1β (10 ng/mL) for 24 h.

Protein concentration in the culture supernatants was determined by liquid chromatography/mass spectrometry (*n* = 4). Schematic experimental representation (**d**) and heatmap of differentially expressed proteins between vehicle and rIL-1β-treated EGCs (**e**). **f, g** Primary embryonic neurosphere-derived EGCs were stimulated for 24 h with H-CM or TME-CM in the presence or absence of IgG or anti-IL-1R (5 µg/mL each) (*n* = 3 H-CM and TME-CM). Schematic representation of experimental setup (**f**). Relative mRNA levels for *Lcn2, Timp1, Ccl2,* and *Il6* normalized to the housekeeping gene *Rpl32* (**g**). **h** WT C57BL/6J mice were intracolonically injected at day(d)0 with MC38 cells, and both stromal and immune cells were assessed for IL-1β and IL-1α expression at d21. Data are presented as the frequency of total live IL-1β+ or IL-1α+ cells (*n* = 5 mice). Data are represented as mean ± SEM (**c, g, h**). Statistical analysis: unpaired two-tailed Mann-Whitney test (**c**), unpaired one-way Anova test with multiple comparison correction (**g**). **p* < 0.005, ****p* < 0.0005, *****p* < 0.00005. Source data and exact *p* values are provided as a Source Data file.

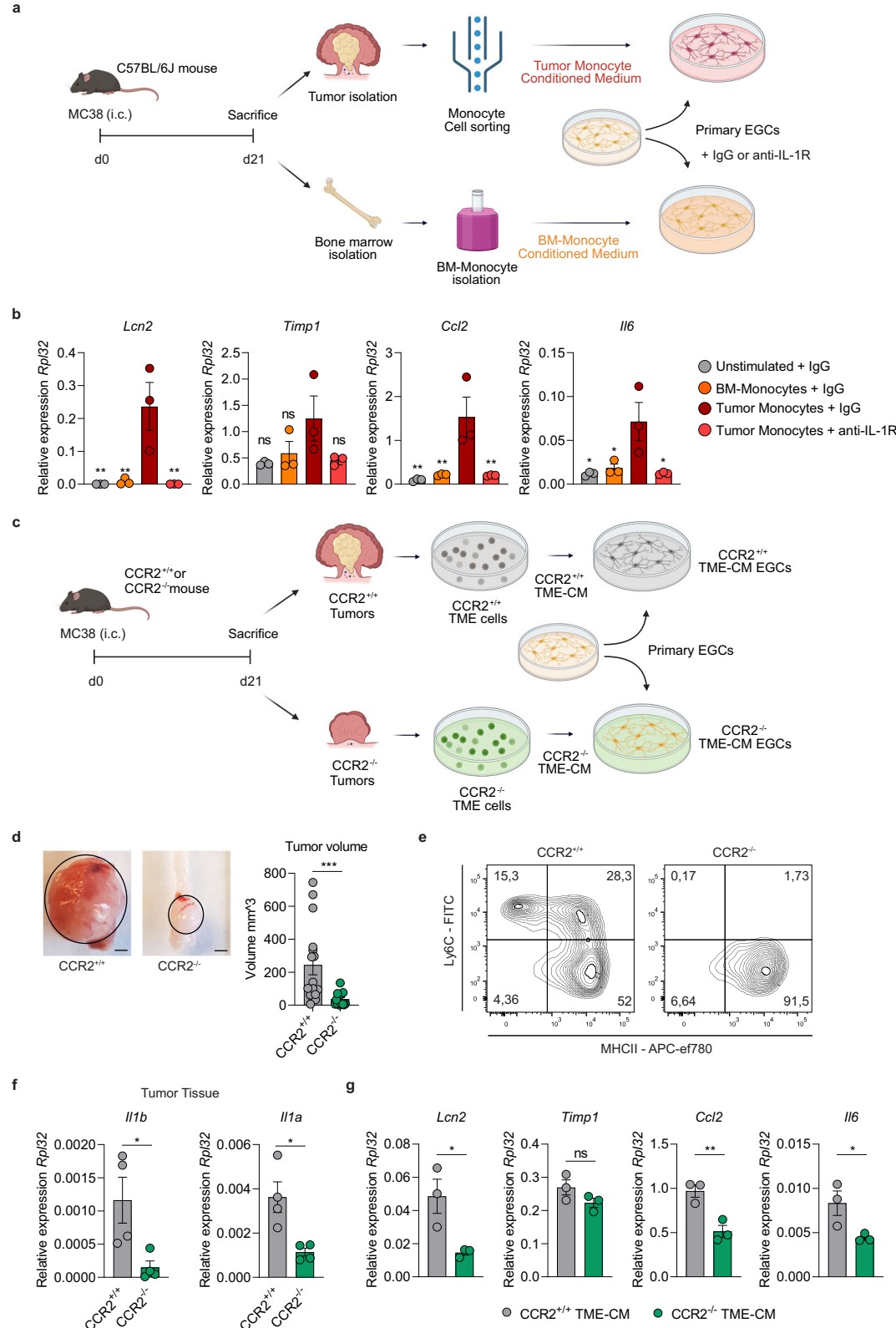

as the principal producers of both IL-1β and IL-1α at the RNA and protein levels (Fig. 4h and Supplementary Fig. 5h).

To define the possible effect of tumor monocyte-derived IL-1 on the immunomodulatory reprogramming of EGCs, we isolated tumor- and bone marrow (BM)-derived monocytes from mice bearing orthotopic colon tumors and exposed primary enteric glia to their supernatant with or without IL-1R blockade (Fig. 5a). Remarkably, the supernatant of tumor monocytes was able to induce a higher expression of CRC EGC marker genes (*Lcn2*, *Timp1*, *Ccl2*, and *Il6*) compared to control BM-derived monocytes in an IL-1R-dependent manner (Fig. 5b).

To further verify the monocyte origin of IL-1 signaling, we examined the effects on primary EGCs of orthotopic TME-CM, sourced from tumors induced in both monocyte-deficient [C-C chemokine receptor type 2 deficient (CCR2$^{-/-}$)] and monocyte-competent mice (CCR2$^{+/+}$)

**Fig. 5 | Monocyte-derived IL-1 promotes the CRC EGC phenotype. a**, **b** Primary embryonic neurosphere-derived EGCs were stimulated for 24 h with IgG or anti-IL-1R (5 μg/mL each) with or without the supernatant of sorted tumor monocytes or bone marrow (BM)-derived monocytes from WT C57BL/6J mice bearing orthotopic CRC tumors. Schematic representation of experimental setup (**a**). Relative mRNA levels of *Lcn2*, *Timp1*, *Ccl2*, and *Il6*, normalized to the housekeeping gene *Rpl32*, in primary embryonic neurosphere-derived EGCs were compared between EGCs stimulated with tumor monocyte supernatant + IgG and all other conditions (*n* = 3 primary EGC cultures and monocytes) (**b**). **c**–**g** CCR2⁺/⁺ and CCR2⁻/⁻ mice were intracolonically injected at day(d)0 with MC38 cells, tumor tissue was collected at d21. Then, in vitro embryonic neurosphere-derived EGCs were cultured for 24 h with the tumor microenvironment-conditioned medium (TME-CM) of CCR2⁺/⁺ and CCR2⁻/⁻ tumors. Schematic representation of experimental setup (**c**).

Representative pictures (left, scale bar 2 mm) and quantitative comparison of tumor volume (right) (*n* = 16 CCR2⁺/⁺, *n* = 17 CCR2⁻/⁻) (**d**). Representative contour plots of tumor-infiltrating monocytes and macrophages gated on live-CD45⁺-CD11b⁺-Ly6G⁻-CD64⁺ cells (**e**). Relative mRNA levels of *Il1b* and *Il1a* normalized to the housekeeping gene *Rpl32* in CCR2⁺/⁺ and CCR2⁻/⁻ CRC tumors (*n* = 3 mice) (**f**). Relative mRNA levels of *Lcn2*, *Timp1*, *Ccl2*, and *Il6* in EGCs stimulated with TME-CM of CCR2⁺/⁺ and CCR2⁻/⁻ mice, normalized to the housekeeping gene *Rpl32* (*n* = 3 TME-CM) (**g**). Data represented as mean ± SEM (**b**, **d**, **f**, **g**). Statistical analysis: One-way ANOVA test with correction for multiple comparisons, compared to tumor monocyte supernatant + IgG condition (**b**), unpaired two-tailed Mann Whitney test (**d**) or unpaired two-tailed t-test (**f**, **g**). *$p < 0.05$, **$p < 0.005$, ***$p < 0.0005$, ns not significant. Source data and exact *p* values are provided as a Source Data file.

(Fig. 5c). Consistent with previous findings[27], the volume of orthotopic tumors grown in the colonic mucosa of CCR2⁻/⁻ mice was significantly reduced (Fig. 5d) as a consequence of the lower number of monocytes and monocyte-derived macrophages in CCR2⁻/⁻ tumors, as confirmed by flow cytometry (Fig. 5e). Consistently, *Il1b* and *Il1a* expressions were significantly decreased in the tumor tissue of CCR2-deficient mice compared to WT mice (Fig. 5f), further corroborating that monocytes and monocyte-derived macrophages are the major source of IL-1 ligands in the colon TME. As expected, TME-CM isolated from the CCR2⁻/⁻ mice failed to induce CRC EGC reprogramming as reflected by the reduced expression of *Lcn2*, *Ccl2*, and *Il6* when compared with EGCs treated with CCR2⁺/⁺ TME-CM (Fig. 5g). A similar trend, although not significant, was observed for *Timp1*. Overall, our findings strongly support the concept that IL-1, derived from tumor-infiltrating monocytes- and monocyte-derived macrophages, provides remodeling of the neighboring enteric glia into activated and immunomodulatory CRC EGCs.

## IL-1R signaling in EGCs promotes SPP1⁺ TAM differentiation

Next, we assessed whether IL-1R blocking in CRC EGCs might directly affect TAM differentiation. For this purpose, EGC cultures were exposed to TME-CM or H-CM in the presence of an IL-1R blocking antibody and subsequently their supernatant was used to treat naive monocytes (Fig. 6a). To exclude any direct impact of anti-IL-1R antibody on monocytes, we removed the antibody from the EGC supernatant by using magnetic beads. Notably, the blockade of IL-1R led to a significantly reduced differentiation of monocytes into SPP1⁺ TAMs with the supernatant of TME-CM EGCs. However, IL-1R blockade did not alter TAM polarization in the context of H-CM EGCs both at RNA and protein level (Fig. 6b, c and Supplementary Fig. 6a-b). As a result, IL-6 levels were markedly reduced in the supernatant of TME-CM-exposed EGCs following IL-1R inhibition (Fig. 6d). These findings further corroborated our hypothesis regarding the critical role of the IL-1R/IL-6 axis in CRC EGCs in directing monocyte differentiation towards the SPP1⁺ TAM phenotype.

Then, to investigate the effect of IL-1R signaling inhibition on the immunomodulatory functions of EGCs in vivo, we co-injected WT or IL-1R1⁻/⁻ EGCs and MC38 cells in C57/BL6J mice (Fig. 6e). Interestingly, co-injection of tumor cells with IL-1R1⁻/⁻ EGCs resulted in lower tumor growth and reduced infiltration of SPP1⁺ TAMs when compared with tumors co-injected with WT EGCs (Fig. 6f-g). Lack of IL-1R in co-injected EGCs had no impact on C1Q⁺ TAMs.

Acknowledging the heterogeneity of CRC with regard to the immune compartment[28], we sought to investigate the glial-TAM crosstalk also in "immune cold" MSS-CRC like tumors, by injecting villinCre^ER Apc^fl/fl Kras^G12D/+ Trp53^fl/fl Trgfbr1^fl/fl (AKPT) tumor-derived cells orthotopically[29]. Indeed, while the MC38 model is representative of "immune hot" tumors, characterized by a large immune infiltrate[30], the AKPT model is characterized by a low tumor-immune compartment[30,31]. Thus, we injected AKPT cells in the colonic mucosa of mice with a glial-specific deletion of IL-1R1 (GFAP^Cre IL-1R1^fl/fl) and

their littermates (GFAP^Wt IL1R1^fl/fl) (Fig. 6h and Supplementary Fig. 6c). Notably, also in this model, deletion of IL-1R in EGCs impacted tumor growth and SPP1⁺ TAM expansion (Fig. 6i, j), suggesting IL-1R triggered EGC-TAM crosstalk as a broad mechanism across independent orthotopic CRC subtypes.

In addition, we investigated the glia-TAM interaction also in a colitis-associated colorectal cancer model based on AOM/DSS administration[32] (Fig. 7a). Here, using the glia reporter mouse line GFAP^Cre Ai14^fl/fl (Supplementary Fig. 6c), we confirmed spatial proximity of EGCs (tdTomato⁺) and TAMs (IBA1⁺) in the tumor regions (Fig. 7b), as we have previously observed in the orthotopic MC38 model. Transcriptomic analysis comparing AOM/DSS-induced tumors to naive tissue in wild-type mice supported the involvement of the IL-1R/IL-6 pathway in EGC-TAM crosstalk (Supplementary Fig. 6d, e, Supplementary Data 3). In particular, transcriptomic differences highlighted an increased expression of CRC EGC markers, as well as TAM signature genes together with increased IL-1 signaling. Consistent with the role of enteric glia in the orthotopic CRC model, the number of colonic tumors was diminished in AOM/DSS-treated GFAP^Cre IL-1R1^fl/fl compared to littermate GFAP^Wt IL-1R1^fl/fl mice, while no differences in weight loss or EGC network morphology were detectable between the two genotypes (Fig. 7c and Supplementary Fig. 6f, g). Interestingly, glial-specific IL-1R deficiency correlated with a decline in SPP1⁺ TAMs, but C1Q⁺ TAM levels remained unchanged (Fig. 7d). Importantly, glial-specific IL-1R deficiency was further associated with reduced *Il6* gene expression in tumor tissues of GFAP^Cre IL-1R1^fl/fl compared to control GFAP^Wt IL-1R1^fl/fl tumor lesions (Fig. 7e).

To further validate the expression of IL-6 in tumor EGCs, we performed fluorescence-activated cell sorting of enteric glia from healthy or tumor colonic mucosa of AOM/DSS-treated GFAP^Cre Ai14^fl/fl mice. In line with our in vitro data, we detected a strong increase of *Il6* expression in tumor versus healthy tdTomato^pos EGCs (Fig. 7f and Supplementary Fig. 6h-i). Consistently, IL-6 staining in tumor sections from the EGC reporter mouse line Sox10^CreERT2 Ai14^fl/fl, revealed that IL-6 protein co-localized with tdTomato⁺ cells in AOM/DSS tumors (Fig. 7g and Supplementary Fig. 6j).

In conclusion, these findings offer further in vivo evidence underscoring the IL-1-dependent interaction between EGCs and SPP1⁺ TAMs within the CRC milieu.

## IL-1R induced-CRC EGC phenotype in patients with CRC

After identifying enteric glia-immune interactions in preclinical models of CRC, we investigated whether a similar process might influence disease progression in CRC patients. Initially, spatial tissue co-localization of EGCs and TAMs was confirmed in patient-derived CRC samples (Fig. 8a). Next, the possible contribution of EGCs in disease outcome was defined using the colon and rectal cancer datasets from The Cancer Genome Atlas (TCGA- COAD, and READ). Here, patients were hierarchically clustered into two groups based on the expression levels (high versus low) of the EGC transcriptomic signature, consisting of genes highly expressed by EGCs as defined in previously published

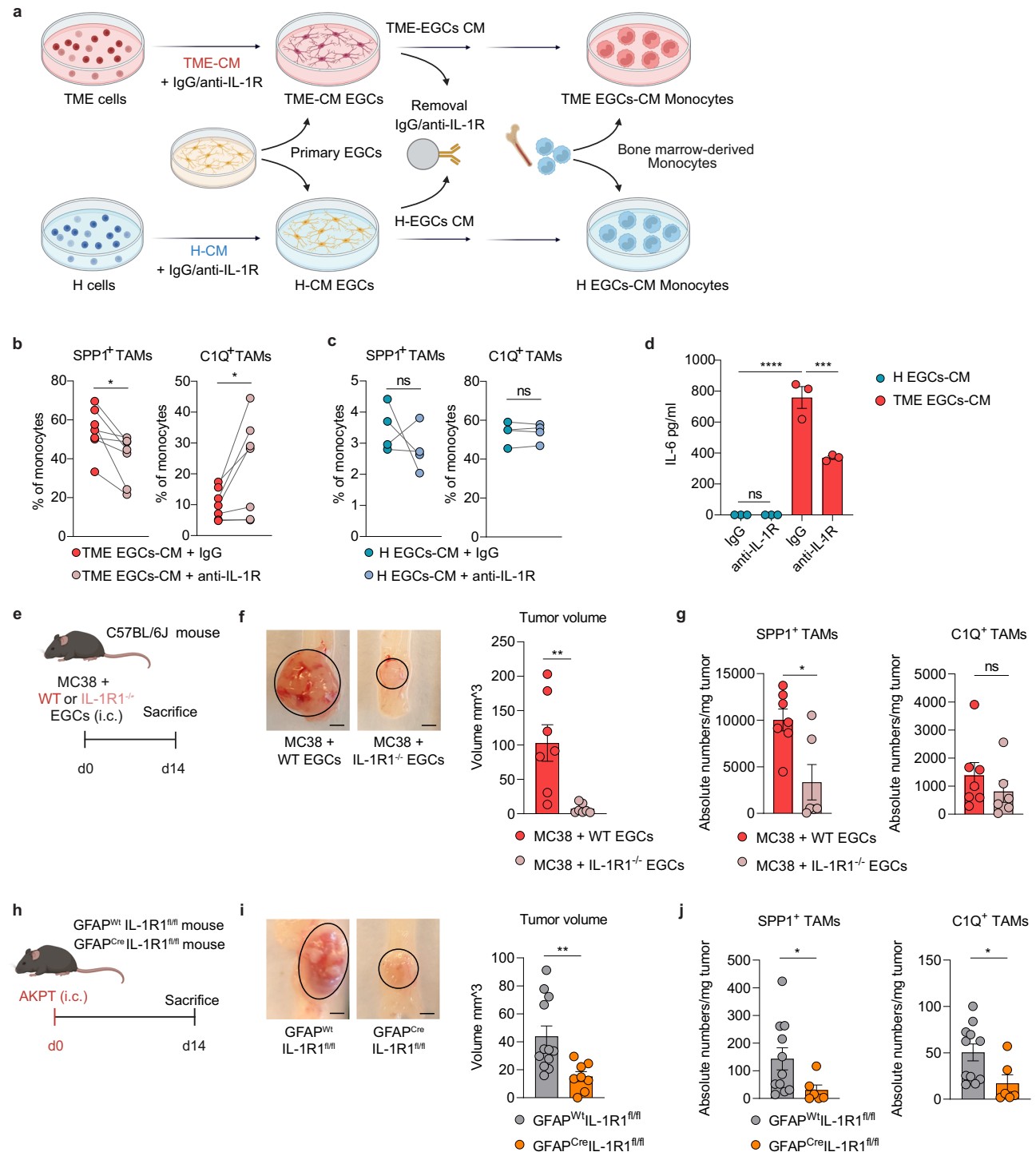

**Fig. 6 | IL-1R signaling in EGCs promotes SPP1⁺ TAM differentiation and tumor progression. a–c** Murine bone marrow-derived monocytes were cultured for 48 h with supernatant of primary embryonic neurosphere-derived EGCs, which were pre-incubated for 24 h with tumor microenvironment conditioned medium (TME-CM) or healthy-CM (H-CM) together with isotype IgG or anti-IL-1R ($5\,\mu g$/mL each). Antibodies were removed from the CM through the application of Dynabeads™ Protein G prior to incubation with monocytes. Experimental design (**a**). Flow cytometry quantification of SPP1⁺ TAMs and C1Q⁺ TAMs after TME-EGCs ($n = 7$) (**b**) or H-EGCs ($n = 4$) (**c**) supernatant stimulation. **d** IL-6 concentration in the conditioned medium of H-EGCs and TME-EGCs pre-incubated with IgG or anti-IL-1R ($n = 3$ primary EGC cultures). **e–g** WT C57BL/6J mice were intracolonically (i.c.) injected with MC38 cells and embryonic neurosphere-derived WT EGCs or IL1R1⁻/⁻ EGCs (1:1 ratio). Tumor growth and immune infiltration were assessed on day (d)14. Schematic representation of EGCs co-injection mouse model (**e**) with representative

pictures (scale bar 2 mm) and quantitative comparison of tumor volume ($n = 7$ mice) (**f**). Data show absolute tumor-infiltrating TAM cell numbers per mg tumor tissue ($n = 7$ MC38 + WT EGCs, $n = 6$ MC38 + IL-1R1⁻/⁻ EGCs) (**g**). **h–j** GFAP^Wt^IL-1R1^fl/fl^ and GFAP^Cre^IL-1R1^fl/fl^ mice were i.c. injected with AKPT cells. Tumor growth and immune infiltration were assessed at d14. Schematic representation of orthotopic CRC mouse model (**h**) with representative pictures (scale bar 2 mm) and quantitative comparison of tumor volume ($n = 12$ GFAP^Wt^IL-1R1^fl/fl^, $n = 8$ GFAP^Cre^IL-1R1^fl/fl^) (**i**). Data show absolute tumor-infiltrating TAM cell numbers per mg tumor tissue ($n = 11$ GFAP^Wt^IL-1R1^fl/fl^, $n = 6$ GFAP^Cre^IL-1R1^fl/fl^) (**j**). Data are represented as mean ± SEM (**d, f, g, i, j**). Statistical analysis: paired two-tailed Wilcoxon test (**b, c**), two-way ANOVA test with correction for multiple comparisons (**d**) or unpaired two-tailed Mann-Whitney test (**f, g, i, j**). *$p < 0.05$, **$p < 0.005$, ***$p < 0.0005$, ****$p < 0.00005$, ns not significant. Source data and exact $p$ values are provided as a Source Data file.

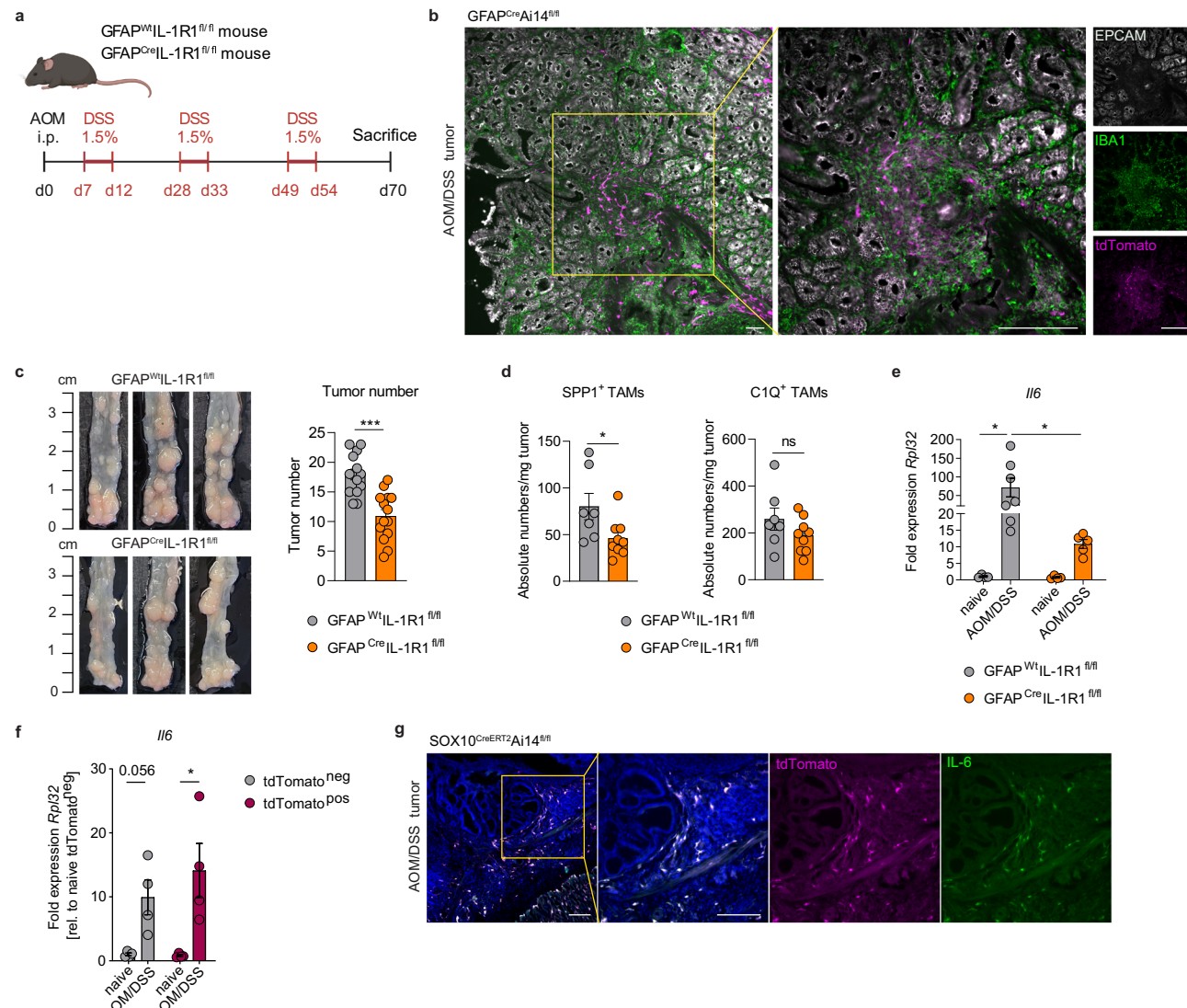

**Fig. 7 | IL-1R deficient EGCs impair tumor progression and SPP1+ TAM differentiation in colitis-associated CRC. a** Schematic representation of the murine AOM/DSS CRC model. Mice were intraperitoneally (i.p.) injected with azoxymethane (AOM, 10 mg/kg body weight) at day(d)0. Starting from d7, mice underwent 3 repetitive cycles of 1.5% dextran sodium sulfate (DSS) in drinking water as indicated. **b** GFAP^Cre^Ai14^fl/fl mice were subjected to the AOM/DSS model (Fig. 7a) using 1% DSS per cycle. Representative images of EPCAM (white), IBA1 (green) and tdTomato (magenta) in tumor sections at d70 (scale bar 100 μm) (*n* = 3). **c-e** GFAP^Wt^IL-1R1^fl/fl and GFAP^Cre^IL-1R1^fl/fl mice were subjected to the AOM/ DSS model (Fig. 7a). Tumors number and TAMs infiltration were assessed at d70. Tumor numbers of GFAP^Wt^IL-1R1^fl/fl and GFAP^Cre^IL-1R1^fl/fl littermates, representative images (left) and quantitative comparison of tumor numbers (right) (*n* = 14 mice per genotype) (**c**). Corresponding absolute numbers of SPP1+ and C1Q+ TAMs per mg tumor tissue (*n* = 7 GFAP^Wt^IL-1R1^fl/fl, *n* = 9 GFAP^Cre^IL-1R1^fl/fl) (**d**). Relative mRNA levels for *Il6*, normalized to the housekeeping gene *Rpl32* in naive (*n* = 4) and AOM/ DSS treated mice (*n* = 7 GFAP^Wt^IL-1R1^fl/fl, *n* = 5 GFAP^Cre^IL-1R1^fl/fl) (**e**). **f** GFAP^Cre^Ai14^fl/fl mice underwent the AOM/DSS model (Fig. 7a) using 1% DSS. Tumor or naive colon cells were isolated at d70 and FACS-sorted. *Il6* expression levels of sorted tdTomato^pos glial cells versus remaining tdTomato^neg cells of naive and AOM/DSS-treated GFAP^Cre^Ai14^fl/fl mice (*n* = 4). Expression displayed as fold to *Rpl32* and relative to naive tdTomato^neg cells. **g** SOX10^CreERT2^Ai14^fl/fl mice were i.p. injected with Tamoxifen (1 mg in 100 μL sterile corn oil) on d−7, −6, and −5. Subsequently, mice were subjected to the AOM/DSS model (Fig. 7a) using 2% DSS per cycle. Representative image of tdTomato (magenta), IL-6 (green), and DAPI (blue) in tumor section at d70 (scale bar 100 μm) (*n* = 3). Data are represented as mean ± SEM (**c**–**f**). Statistical analysis: unpaired two-tailed t-test (**c**), unpaired two-tailed Mann-Whitney test (**d**), and two-way ANOVA with correction for multiple comparisons (**e**, **f**). **p* < 0.05, ***p* < 0.0005, ns not significant. Source data and exact *p* values are provided as a Source Data file.

scRNA-seq datasets[22,33,34] (Supplementary Fig. 7a and Supplementary Table 3). Here, we observed that CRC patients with a higher enteric glia transcriptomic signature presented with a decreased overall survival probability compared to patients with low EGC signature (Fig. 8b and Supplementary Fig. 7b). Importantly, the correlation between the expression of the high EGC gene signature with worse survival probability in CRC patients was confirmed in a second independent CRC dataset, known as the Sidra-LUMC AC-ICAM cohort[35](Supplementary Fig. 7c, d).

In-depth characterization of the COAD-READ patients with high EGC gene signature revealed 79% of this group belonged to the mesenchymal consensus molecular subtype 4 (CMS4) (Supplementary Fig. 8a), defined by the stromal invasion phenotype, which is generally associated with the worst overall and relapse-free survival[36]. In comparison, only minor differences were identified when divided based on stage, microsatellite stability, or intrinsic CMS (iCMS). Interestingly, CRC patients with pronounced EGC involvement also exhibited higher expression of SPP1+ TAM signature genes (Fig. 8c).

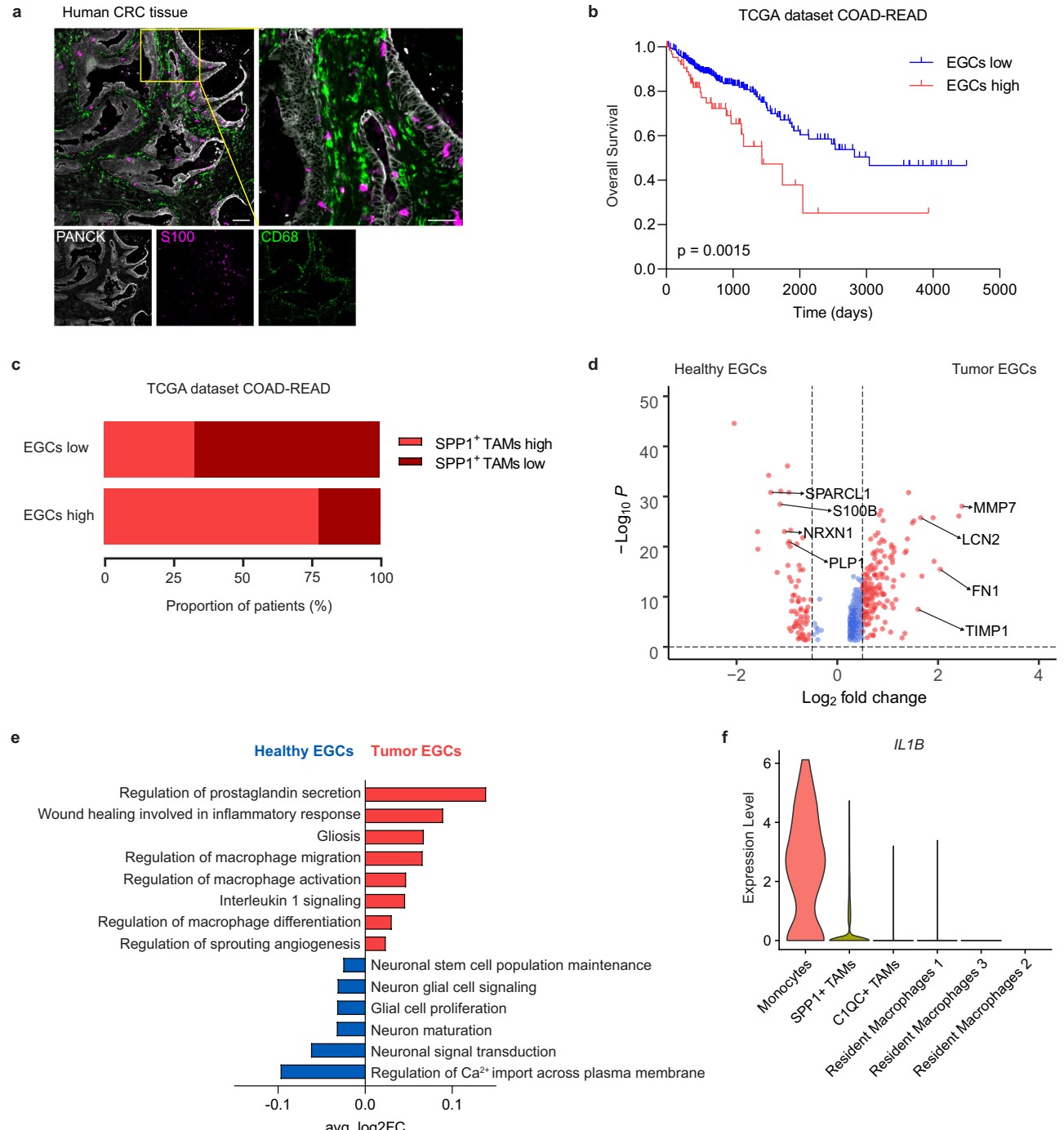

**Fig. 8 | IL-1R induced-CRC EGC phenotype in patients with CRC. a** Representative image showing S100 (magenta), CD68 (green) and Pancytokeratin (PANCK, white) in a human CRC tissue section (scale bar 200 μm and 50 μm). Image representative of ten stainings. **b, c** TCGA COAD and READ patients stratified based on their expression of the EGCs signature genes ($n = 309$ EGCs low, $n = 67$ EGCs high). Kaplan-Meier overall survival curve for EGCs high and low patients (**b**). The proportion of EGCs high and low patients classified based on high or low SPP1+ TAMs gene signature expression (**c**). **d, e** Transcriptome analysis of tumor EGCs in CRC patients (KUL3 Dataset[33], $n = 5$). Volcano plot of differentially expressed genes between healthy and tumor EGCs (**d**), highlighting genes defining the tumor EGCs signature. Gene Set Enrichment Analysis presenting GO terms of interest (**e**). **f** Violin plot showing expression of *IL1B* in the tumor-infiltrating myeloid cell clusters of human CRC in the KUL3 Dataset[33] ($n = 5$). Statistical analysis: Mantel Cox test (**b**) and Wilcoxon test with Bonferroni correction using Seurat (**d**). Source data are provided as a Source Data file.

Finally, to assess if the murine CRC EGC transcriptional signature was conserved also in human CRC EGCs, we analyzed EGCs identified in a scRNA-seq dataset containing both CRC lesions and unaffected colonic tissues[33] (Supplementary Fig. 8b, c). By comparing gene expression profiles of tumor and healthy EGCs, we identified 589 genes specifically expressed in human CRC EGCs (Fig. 8d and Supplementary

Data 2). Strikingly, among the top differentially expressed CRC EGC genes, we identified the two key murine CRC EGC marker genes *Lcn2* and *Timp1*. Using gene set enrichment analysis, we found that human EGC populations differentiate along the same homeostatic and tumor pathogenesis pathway transcriptomic signatures (Fig. 8e) as seen in our murine EGCs (Fig. 2e). Importantly, the significance of IL-1

signaling in the differentiation of patient CRC EGCs was further confirmed by the increased transcriptomic signatures for "*Interleukin-1 signaling*" as well as "*Regulation of macrophage differentiation, activation and migration*" compared to healthy colonic EGCs (Fig. 8e). In alignment with our preclinical findings, cell-population profiling of human CRC samples identified tumor-infiltrating monocytes as the primary source of IL-1β production (Fig. 8f and Supplementary Fig. 8d). Altogether, our data from mice and patients indicate the relevance of IL-1R induced glial-TAM interaction in CRC pathogenesis.

## Discussion

Using a variety of in vitro and in vivo models, we uncover a positive feedback loop between EGCs and TAMs in CRC. More specifically, we found that monocytes and monocyte-derived macrophages within the TME are the main producers of IL-1, inducing a pro-tumorigenic reactive phenotype in EGCs. IL-1-activated CRC EGCs via IL-6, in turn, directly promote the differentiation of tumor-infiltrating monocytes towards SPP1[+] TAMs. Importantly, the abundance of EGCs correlates with worse disease outcomes, as observed in both pre-clinical CRC mouse models and in patients with CRC.

EGCs are highly plastic cells that can rapidly adapt their functions under the influence of microenvironmental cues[7]. Recent studies have identified specific immunomodulatory factors, including IFN-γ, IL-1, and ATP, as triggers of enteric glia phenotypic and functional reprogramming in both homeostasis and diseased conditions[12,15,17]. In particular, IL-1-mediated EGC reactivity and its effects on immune cell modulation have been extensively studied in the context of intestinal inflammation[17,37]. However, the mechanisms underlying these processes in CRC were not yet understood. Using an unbiased approach combining murine bulk and human scRNA-seq, our investigation pinpointed TME-derived IL-1 as the principal initiator of EGC reactivity. Furthermore, we found that this IL-1-triggered EGC activation coincides with a profoundly immunomodulatory transcriptional signature in CRC EGCs. Of note, IL-1R signaling in EGCs may hold relevance for additional functions of CRC EGCs, as in vitro studies indicated the significance of IL-1 in EGC-cancer stem cell interactions[4]. Interestingly, we identified tumor-infiltrating monocytes and macrophages as the main source of IL-1 within the tumor. Nevertheless, although our in vivo data could not verify epithelial cells as a significant IL-1 source in CRC, tumor epithelial IL-1 might also contribute to EGC activation during CRC[4].

Our study utilizes single-cell and bulk RNA-seq techniques to better understand and predict the interactions between EGCs and TAMs within the colonic TME. In our murine orthotopic CRC model, we identified two distinct subsets of TAMs. The C1Q[+] TAMs, which preferentially express genes involved in phagocytosis and antigen presentation, coexist in the TME with SPP1[+] TAMs that are enriched for factors regulating angiogenesis and extracellular matrix, suggesting their key role in colon tumorigenesis. This dichotomy, recently also identified in patients with colorectal cancer[24] supports the relevance of our findings to human disease. In our research, we found that the two types of TAMs−SPP1[+] TAMs and C1Q[+] TAMs−are regulated by EGCs. Specifically, EGCs within tumors were effective in promoting the SPP1[+] TAM phenotype. In contrast, when monocytes were stimulated with healthy EGCs, the C1Q[+] TAM phenotype was induced. Importantly, while C1Q[+] macrophage differentiation was independent of IL-1R signaling in EGCs, we found that IL-1-activated tumor EGCs drive monocyte differentiation into pro-tumorigenic SPP1[+] macrophages via IL-6. Thereby, our findings underscore a bidirectional EGC-TAM interaction which was proven to influence tumor burden and SPP1[+] TAMs in three independent CRC mouse models. Of note, IL-6 can be produced by different cell types in the TME contributing to tumor development through direct and indirect pathways[38]. However, utilizing FACS-sorting and EGC-specific genetic IL-6 and IL-1R depletion models, we confirmed the critical role of IL-1-triggered EGC-IL-6 release as a regulator of tumor growth and SPP1[+] TAM differentiation. Thereby, our

study identified EGCs as an additional important regulator of SPP1[+] TAM differentiation that, together with other cancer-associated stromal cells, may contribute to tumor progression[39,40]. Consistent with this, pan-cancer analysis has pinpointed SPP1[+] TAMs as the most pro-tumorigenic macrophage subset across various cancers[25,41].

Furthermore, our data also point to a functional association between CRC EGCs and monocyte migration, as reflected by increased *Ccl2* expression in CRC glia. Considering that the tumor monocyte population decreased upon EGC depletion, we speculate that CRC EGC-derived chemokines (i.e., CCL2 and CXCL5) could also promote the infiltration of monocytes in the colonic tumor site. This would be in line with our recent findings showing early expression of CCL2 by reactive EGCs in the context of intestinal inflammation[19]. However, further research will need to determine whether the pro-tumorigenic role of EGCs is exerted solely on the SPP1[+] TAMs or whether EGCs also affect other immune or stromal cells via glial-derived factors. Recent work by Progatzky et al. provides supportive evidence for this, demonstrating that EGCs are involved in a protective immune and stromal response to control parasitic insult in the gut[12].

Consistent with the identification of a pro-tumorigenic EGC phenotype in three independent CRC mouse models, we found that also in patients an increased EGC transcriptomic signature was associated with reduced overall survival in two independent CRC cohorts, the TCGA cohort and the Sidra-LUMC AC-ICAM cohort[35]. Therefore, we could speculate that the EGC gene signature might be used as a potential biomarker to predict disease outcomes. In line with the stromal nature of EGCs, we demonstrated that the vast majority of patients with high EGC involvement belonged to the CMS4, which is characterized by a mesenchymal-like phenotype, a strong stromal infiltration and the worst overall and relapse-free survival compared to the other CMS subtypes[36]. Gene ontology analysis revealed that also the human tumor glial cells were enriched for immunomodulatory transcriptional programs related to macrophage differentiation, leading to the assumption that CRC EGC-derived signals modulate TAMs also in patients. In line, human tumor EGCs also displayed enrichment for the GO term "Interleukin-1 signaling" hinting at a similar EGC activation in CRC as in our preclinical models. Moreover, various studies demonstrated increased IL-6 levels in both tumor tissues and serum samples of human CRC patients compared to healthy controls[42,43]. Consistently, immune-related pathways, including IL-6 signaling, were enriched in CRC EGCs in a human single-cell data set published by Qi et al.[39].

Our research elucidates the role of the IL-1/IL-6 axis in glial-immune communication in CRC, which could potentially be of relevance to various other tumors exhibiting neuronal infiltration, a feature often associated with less favorable disease outcomes[44–47]. Apart from EGCs, peripheral glial cells, including Schwann cells, are known to play a crucial role in cancer pathophysiology, as demonstrated in pancreatic ductal adenocarcinoma, lung cancer, and melanoma[48–50]. Consistent with the pro-tumorigenic functions of EGCs in CRC, studies in melanoma models have shown that tumor Schwann cells favor the differentiation of pro-tumorigenic macrophages enhancing tumor growth[50]. Therefore, glial-immune crosstalk might be an overlooked and conserved component of tumor pathophysiology in many cancer types beyond CRC.

In conclusion, our study reveals a critical role for IL-1R signaling in driving enteric glia-macrophage interactions in CRC pathogenesis. Our research provides insight into a complex neuroimmune mechanisms underlying CRC development, underscoring potential biomarkers and specific therapeutic targets that hold the promise of transforming the management of this devastating disease.

## Methods

### Animals
All experimental procedures were approved by the Animal Care and Animal Experiments Ethical Committee of KU Leuven (208/2018, 213/

2018 and 159/2021) or by the Regional Office for Nature, Environment and Consumer Protection of North-Rhine-Westphalia, Germany (81-02.04.2021.A424). WT C57BL/6J (JAX:000664), CCR2$^{-/-}$ (JAX: 004999), IL-1R1$^{-/-}$ (JAX:003245), PLP1$^{CreERT2}$iDTR (JAX:005975 and JAX:007900), PLP1$^{CreERT2}$Ai14$^{fl/fl}$ (JAX:005975 and JAX:007908), GFAP$^{Cre}$IL-1R1$^{fl/fl}$ (JAX:012886 and JAX:028398), GFAP$^{Cre}$Ai14$^{fl/fl}$ (JAX:012886 and JAX:007908) and SOX10$^{CreERT2}$Ai14$^{fl/fl}$ [SOX10$^{CreERT2}$ (kindly provided by Dr. Vassilis Pachnis[51], (Ai14$^{fl/fl}$ JAX:007908)], SOX$^{CreERT2}$IL-6$^{fl/fl}$ (SOX10$^{CreERT2}$ as mentioned previously and IL-6$^{fl/fl}$ were kindly provided by Juan Hidalgo's lab[52]) mice were originally purchased from Jackson Laboratory and bred in our animal facilities. All mice were housed in temperature-controlled specific pathogen–free facilities with ad libitum access to standard chow diet and water under 12-h light–dark cycles at the KU Leuven or University of Bonn. Criteria for euthanasia were >20% weight loss for AOM/DSS model (according to 81-02.04.2021.A424) and tumor growth that interferes with fecal evacuation (>50% of colonic lumen obstruction, Grade 5) for the orthotopic model. In cases where limits have been exceeded due to the natural variety of the models, animals were euthanized according to ethical regulations. In the case of GFAP$^{Cre}$IL-1R1$^{fl/fl}$ mice, following recommendation by The Jackson Laboratory (JAX:012886), we used a strict mating scheme using only Cre$^+$ carrying female with Cre$^-$ males to overcome any issues of germline Cre-expression and to produce only litters with a GFAP-promotor-driven Cre-expression. Specific GFAP$^{Cre}$ recombination was confirmed in the reporter mouse line GFAP$^{Cre}$Ai14$^{fl/fl}$ showing a strong overlap of tdtomato signal with immunolabelled GFAP and SOX10 cells in colonic tissue, confirming Cre activity exclusively in enteric glia (Supplementary Fig. 6c).

### In vitro tumor EGCs model

Both orthotopic tumors and healthy colons of C57BL/6J, CCR2$^{+/+}$ or CCR2$^{-/-}$ mice were digested for 30 min in DMEM with 2.5% FBS, 100 µg/mL Penicillin and Streptomycin, 200 U/mL collagenase IV (Gibco, ThermoFisher Scientific) and 125 µg/mL type II dispase (Gibco, ThermoFisher Scientific) to obtain a single-cell suspension. Tumor microenvironment conditioned medium (TME-CM) and healthy colon conditioned medium (H-CM) were generated by culturing $5 \times 10^5$ cells/mL in DMEM-complete medium overnight. Next, primary murine embryonic neurosphere-derived EGCs were stimulated with the TME-CM or H-CM for 6, 12, or 24 h. For IL-1R blocking experiments, primary embryonic neurosphere-derived EGCs were incubated for 24 h with TME-CM together with 5 µg/mL isotype IgG (BioXCell) or 5 µg/mL anti-IL-1R (BioXCell).

### Orthotopic CRC model

Orthotopic colonic sub-mucosal implantation of CRC cells was performed as previously described[20]. Briefly, MC38 cells[53] or AKPT cells[29,31] were intracolonically (i.c.) injected via endoscopy as a single-cell suspension containing between 75,000–750,000 cells/100 µL PBS depending on the susceptibility of the mouse strain. Both male and female mice were used for orthotopic tumor implantation. AKPT cells were received from Prof. Owen Sansom (UK). For the EGCs supplementation model, primary embryonic neurosphere-derived WT and IL-1R1$^{-/-}$ EGCs were isolated with 0.05% Trypsin-EDTA (Gibco, ThermoFisher Scientific) and treated with HBSS (Gibo, ThermoFisher Scientific) supplemented with 100 µg/mL DNAse I (Roche) and 5 mM MgCl$_2$ (Sigma-Aldrich) for 30 min at RT. Subsequently, EGCs were first washed with HBSS with 5 mM MgCl$_2$ and then with PBS. Finally, EGCs were resuspended in PBS together with MC38 cells in a ratio of 1:1 and orthotopically co-injected in C57BL/6J WT mice. Two weeks prior to the start of each experiment, PLP1$^{CreERT2}$iDTR mice, SOX$^{CreERT2}$IL-6$^{fl/fl}$ and SOX$^{CreERT2}$IL-6$^{wt/wt}$ were injected intraperitoneally (i.p.) 2 times every other day with 100 mg/kg Tamoxifen (Sigma-Aldrich) dissolved in 100 µL MIGLYOL®812 (Sigma-Aldrich). For EGCs in vivo depletion experiments, PLP1$^{CreERT2}$iDTR and C57BL/6J WT mice were injected i.c.

with 2 mg/kg Diphtheria toxin (DT) (Merck, Sigma) dissolved in 100 µL of saline, three and five days prior to the start of the tumor implantation. Tumor volume was determined by caliper measurements and calculated based on the height ($h$), length ($l$) and width ($w$) of the tumor, according to the formula: $(\pi/6)*h*l*w$.

### AOM-DSS model

Female GFAP$^{Cre}$IL-1R1$^{fl/fl}$, GFAP$^{Wt}$IL-1R1$^{fl/fl}$ littermate and GFAP$^{Cre}$Ai14$^{fl/fl}$ mice (10–14 weeks of age) were injected i.p. with azoxymethane (AOM; 10 mg/kg; Sigma-Aldrich) a week prior starting three cycles of DSS colitis using 1.5% DSS in drinking water (MP Biomedicals) for 5 days followed by 16 days of recovery with normal drinking water[54]. On day 70, colonic tumor development was determined. For Sox10$^{CreERT2}$Ai14$^{fl/fl}$ mice subjected to the AOM/DSS model, slight adjustments were made to the protocol. Female Sox10$^{CreERT2}$Ai14$^{fl/fl}$ mice were i.p. injected with Tamoxifen (MP biomedicals, 1 mg in 100 µL sterile corn oil) on days −7, −6, and −5. On day 0, the AOM/DSS model was started as described above, but here mice were subjected to a DSS concentration of 2% in drinking water since this strain was less susceptible to DSS. On day 70, colons were harvested and cryo-embedded as Swiss rolls for immunohistochemistry.

### MC38 cell line

Murine colon adenocarcinoma cell line MC38 (NCI, ENH204-FP) was kindly provided by Prof. Max Mazzone (VIB - KU Leuven). The MC38 cell line was maintained in 5% CO$_2$ at 37 °C in high glucose Dulbecco's Modified Eagle Medium (DMEM) (Gibco, ThermoFisher Scientific) supplemented with 10% Fetal Bovine Serum (FBS) (Biowest), 100 µg/mL Penicillin and Streptomycin, 2 mM L- glutamine, 10 mM N-2-hydroxyethylpiperazine-N-2-ethane sulfonic acid (HEPES), 1 mM sodium pyruvate, 50 µM 2-Mercaptoethanol and 1X Non-Essential Amino Acids (all from Gibco, ThermoFisher Scientific).

### AKPT cell line

Murine small intestine villinCre$^{ER}$ Apc$^{fl/fl}$ Kras$^{G12D/+}$ Trp53$^{fl/fl}$ TrgfbrI$^{fl/fl}$ (AKPT) tumor-derived cells were kindly provided by Prof. Owen J. Sansom (Cancer Research UK Beatson Institute). The AKPT cells were maintained in 5% CO$_2$ at 37 °C in DMEM/F-12 (1:1) with L-glutamine and 2.438 g/L sodium bicarbonate (Gibco, ThermoFisher Scientific), 10% FBS (heat inactivation: 30 min at 56 °C, PAN Biotech) and Penicillin 100 U/ml and streptomycin 100 µg/ml (Gibco, ThermoFisher Scientific).

### Embryonic neurosphere-derived enteric glial cell culture

Embryonic neurosphere-derived EGCs were obtained as previously described[19]. Briefly, total intestines from E13.5 C57BL/6J mice were digested with collagenase D (10 mg/mL; Roche) and DNase I (0.08 mg/mL; Roche) in DMEM/F-12 (Gibco, ThermoFisher Scientific) for 1 h at 37 °C under gentle agitation. After digestion, tissue was filtered through a 70 µm cell strainer and cells were cultured in a CO$_2$ incubator at 37 °C in DMEM/F-12, 100 µg/mL Penicillin and Streptomycin, 2 mM L-glutamine, 10 mM HEPES, 1 mM sodium pyruvate, 50 µM 2-Mercaptoethanol supplemented with 1× B27 (Gibco, ThermoFisher Scientific), 40 ng/mL Epidermal Growth Factor (EGF) (Stemcell Technologies) and 20 ng/mL Fibroblast Growth Factors (FGF) (Invitrogen, ThermoFisher Scientific). After a minimum of 1 week of culture, neurospheres were treated with NeuroCult™ Chemical Dissociation Kit (Stemcell Technologies) according to the manufacturer's protocol and filtered through a 70-µm cell strainer. Cells were seeded on Poly-D-Lysine (PDL solution, 1.0 mg/mL, Sigma Aldrich) coated plates and differentiated in DMEM medium supplemented with 10% FBS, 100 µg/mL Penicillin and Streptomycin, 2 mM L- glutamine, 10 mM HEPES, 1 mM sodium pyruvate, 50 µM 2-Mercaptoethanol (DMEM complete medium) until confluence for 5 days to obtain primary EGCs. For IL-1 stimulation experiments, $5 \times 10^4$ neurosphere-derived EGCs/mL were

stimulated with or without recombinant murine 10 ng/mL IL-1α (Peptrotech) and/or 10 ng/mL IL-1β (Sigma Aldrich) for 24 h.

## Neurosphere-derived adult enteric glial cell culture

Neurosphere-derived adult EGC cultures were obtained as described previously[17]. Briefly, small intestines of 8–16 weeks old C57BL/6 mice were harvested, cleaned, cut into 3–5 cm long segments and kept in oxygenated Krebs-Henseleit buffer (126 mM NaCl; 2.5 mM KCl; 25 mM NaHCO$_3$; 1.2 mM NaH$_2$PO$_4$; 1.2 mM MgCl$_2$; 2.5 mM CaCl$_2$, 100 IU/mL Penicillin, 100 IU/mL Streptomycin and 2.5 μg/mL Amphotericin). For each segment, the muscularis layer was peeled and collected for digestion. Muscularis tissues were incubated for 15 min in DMEM containing Protease Type 1 (0.25 mg/mL, Sigma-Aldrich) and Collagenase A (1 mg/mL, Sigma-Aldrich) at 37 °C and 150 rpm. Digestion was stopped with DMEM containing 10% FBS (Sigma-Aldrich) and cells were cultured in proliferation medium (neurobasal medium with 100 IU/Penicillin, 100 μg/mL Streptomycin, 2.5 μg/mL Amphotericin (all ThermoFisher Scientific), FGF and EGF (both 20 ng/mL, Immunotools) at 37 °C, 5% CO$_2$ to promote neurosphere formation. After 1 week in culture, enteric neurospheres were dissociated with trypsin (0.25%, ThermoFisher Scientific) for 5 min at 37 °C and differentiated at 50% confluency on Matrigel (100 μg/mL, Corning) coated six well plates for 1 week in differentiation medium (neurobasal medium with 100 IU/ Penicillin, 100 μg/mL Streptomycin, 2.5 μg/mL Amphotericin, B27, N2 (all Thermo Scientific) and EGF (2 ng/mL, Immunotools). For liquid chromatography/mass spectrometry (LC/MS) experiments, mature EGCs were treated with or without IL-1β (10 ng/mL, Immunotools) for 24 h. Conditioned media were collected and concentrated using Pierce™ Protein Concentrators, 3 K MWCO (ThermoFisher Scientific) according to the manufacturer's instructions. After denaturation at 95 °C for 5 min, samples were snap-frozen and kept at −80 °C until further processing.

## Bone marrow-derived monocyte isolation and stimulation

Murine bone marrow (BM)-derived monocytes were isolated from C57BL/6 mice. Briefly, the tibia and femur were dissected, and BM cells were flushed with DMEM high glucose supplemented with 10% FBS. After cells were collected and counted, monocytes were isolated with the EasySepTM Mouse monocyte isolation kit (Stemcell Technologies) according to the manufacturer's instructions. Next, $5 \times 10^5$ monocytes were stimulated for 48 h with 1 mL of H-CM, TME-CM, or with the supernatant of in vitro embryonic neurosphere-derived EGCs preincubated for 24 h with H-CM or TME-CM. For IL-1R blocking experiments, 5 μg/mL isotype IgG (BioXCell) or 5 μg/mL anti-IL-1R (BioXCell) was removed from the EGCs supernatant by using 20 μL/mL Dynabeads™ Protein G (ThermoFisher Scientific) according to the manufacturer's instructions. For IL-6 neutralization experiments, 20 μL/mL Dynabeads™ Protein G (ThermoFisher Scientific) together with 5 μg/ mL anti-IL-6 (R&D Systems) or 5 μg/mL isotype IgG (R&D Systems) was added to the supernatant of H-CM EGCs and TME-CM EGCs and incubated for 2 h before removal by taking advantage of magnetization with the DynaMag™−2 Magnet (ThemoFisher Scientific).

## MILAN multiplex immunohistochemistry

Multiplex immunohistochemistry and analysis were performed according to a previously published method[55–57] and https://doi.org/ 10.21203/rs.2.1646/v5. Briefly, tissue sections (3 μm thickness) were prepared from formalin-fixed paraffin-embedded human CRC samples (collected at the UZ/KU Leuven biobank with informed consent from the patients according to protocol S66460, approved by the Ethics Committee Research UZ/KU Leuven). First, FFPE-tissue slides were deparaffinized by sequentially placing them in xylene, 100% ethanol and 70% ethanol. Following dewaxing, antigen retrieval was performed using PT link (Agilent) using 10 mM Ethylenediaminetetraacetic acid (EDTA) in Tris-buffer pH 8. Immunofluorescence staining was performed using Bond RX Fully Automated Research Stainer (Leica Biosystems) with the primary antibodies mouse anti-pan cytokeratin (1:500, Invitrogen, ThermoFisher Scientific), rabbit anti-S100B (1:500, Dako) or mouse anti-CD68 (1:100, Invitrogen, ThermoFisher Scientific). The sections were incubated for 4 h with the primary antibodies, washed several times and afterwards stained for 30 min with fluorescently labeled secondary antibodies (1:800, Alexa fluor 647 donkey anti-rabbit and 1:800, Alexa Fluor 488 goat anti-mouse respectively). Slides were then incubated for 10 min with a buffer containing 4,6-diamidino-2-phenylindole (1:1000, DAPI), after which mounting medium (50% glycerol; 584 mM C12H22O11; 10 mM Phosphate, 154 mM NaCl; pH 7,5) and a coverslip (Agilent, ref. CR12230-7) were manually applied to the slides. Then the slides were scanned using a Zeiss Axio Scan Z.1 (Zeiss) at 10x magnification. After completion of the staining procedure, the coverslips were manually removed after 30 min soaking in the washing buffer. Consecutive washing steps were thereafter performed using TBS. Stripping of the antibodies was performed in a buffer containing 1% SDS and β-mercaptoethanol for 30 min at 56 °C. After this stripping process, the slides were washed in a washing buffer for 45 min with frequent changes in the buffer. The staining procedure was repeated until all markers were stained and scanned on each of the slides. Utilize ImageJ (1.53T) and Qu path (0.3.2) were used for the region selection and to subtract background and tissue autofluorescence. Further analysis was performed by using ImageJ.

## Immunohistochemistry and Immunofluorescence

Orthotopic tumors were fixed overnight (ON) at 4 °C in Periodate-Lysine-Paraformaldehyde (PLP) buffer consisting of Milli-Q Water supplemented with 1% paraformaldehyde, 0.075 M lysine (pH 7.4), 0.037 M sodium phosphate (pH 7.4) and 0.01 M NaIO$_4$ (all from Sigma-Aldrich). Samples were washed three times with Milli-Q Water supplemented with 0.037 M sodium phosphate (pH 7.4), followed by a minimum of 4 h incubation in 30% sucrose (VWR chemicals) in PBS. Then, samples were embedded in OCT (Scigen) and stored at −80 °C until usage.

Preceding immunohistochemical staining, 7-μm orthotopic tumor tissue sections on SuperFrost Plus™ Adhesion slides (Epredia) were exposed to two washes with HistoChoice Cleaning Agent for 2 min each (Sigma-Aldrich) and subsequent hydration with Ethanol 100% for 2 min each (Merck) followed by deionized water. Then haematoxylin and eosin (both from Leica) staining was performed using standard procedures. Imaging was performed with Nikon Marzhauser Slide Express2, processed and analyzed using ImageJ.

Preceding immunofluorescent staining, tissues were sectioned to 7-μm thickness on SuperFrost Plus™ Adhesion slides (Epredia) and blocked with blocking buffer consisting of PBS containing 0.02% Sodium azide (Sigma-Aldrich), 0.3% donkey serum (Jackson), and 3% Bovine Serum Albumin (BSA, Serva) for 2 h at room temperature (RT). Subsequently, samples were incubated ON at 4 °C with the following primary antibodies: rat anti-F4/80, (1:500, BioRad) and rabbit anti-GFAP (1:300, Dako) or chicken anti-IBA1 (1:400, Synaptic Systems), rat anti-CD326 (EPCAM) (1:1000, Biolegend) and rabbit anti-Ki67 (1:400, Abcam) in staining buffer consisting of blocking buffer supplemented with 0.3% Triton X-100 (ThermoFisher, Scientific). Then, samples were washed in PBS and incubated with DAPI (4',6-Diamidine-2'-pheny-lindole dihydrochloride; Sigma-Aldrich) combined with the secondary antibodies: donkey anti-rat AF488 (1:1000, Invitrogen, ThermoFisher Scientific), and donkey anti-rabbit Cy5 (1:400, Jackson) or donkey anti-chicken Cy3 (1:800, Invitrogen, Thermo Fisher Scientific), donkey anti-rat 488 (1:800, Invitrogen, Thermo Fisher Scientific) and donkey anti-rabbit AF647 (1:800, SouthernBiotech) in staining buffer for 2 h at RT. Finally, samples were rinsed three times in PBS and mounted with SlowFade Diamond Antifade mounting (Invitrogen, ThermoFisher Scientific). Imaging was performed on the ZEISS LSM 880 confocal

microscope or the Nikon ECLIPSE Ti2 microscope using NIS-Elements AR software (version 5.41.01). Pictures were analyzed using ImageJ.

For immunofluorescent staining of AOM/DSS or naive Swiss rolls of GFAP^CreAi14^fl/fl, GFAP^CreIL-1R1^fl/fl, and SOX10^CreERT2Ai14^fl/fl mice were fixed ON at 4 °C in 4% PFA. Samples were washed once with PBS followed by ON incubation at 4 °C in 30% sucrose (Sigma) in PBS. Subsequently, swiss rolls were embedded in Tissue-Tek® O.C.T.™ Compound (Sakura) and stored at −80 °C until usage.

Prior to immunofluorescent staining, AOM/DSS or naive GFAP^CreAi14^fl/fl, GFAP^CreIL-1R1^fl/fl, and SOX10^CreERT2Ai14^fl/fl samples were sectioned to 14 μm thickness on SuperFrost Plus™ Adhesion slides (Epredia). Slides were washed three times in PBS and blocked with blocking buffer (PBS containing 3% donkey serum and 0.1% Triton X-100) for 1 h at RT. Subsequently, primary antibody staining was performed ON at 4 °C in staining buffer (blocking buffer diluted with PBS in a 1:1 ratio) using the following antibodies: rabbit anti-IBA1 (1:400, Abcam), chicken anti-IBA1 (1:400, Synaptic Systems) and rat anti-CD326 (EPCAM) (1:1000, Biolegend), or rabbit anti-IL-6 (1:100, Abcam), or chicken anti-GFAP (1:1000, Biolegend). For IgG control staining, rabbit IgG (Dianova) was used in the same antibody concentration as rabbit anti-IL6. Slides were washed three times in PBS and secondary antibody staining was performed for 2 h at RT with donkey anti-rabbit FITC (1:800, Dianova), donkey anti-chicken CF633 (1:800, Sigma), donkey anti-rat 488 (1:800, Invitrogen, Thermo Fisher Scientific), or donkey anti-chicken FITC (1:800, Jackson) in staining buffer. After three more washes with PBS, slides were incubated with DAPI (Sigma-Aldrich) for 5 min at RT. After a final wash with PBS, slides were mounted with Shandon™ Immu-Mount™ (Epredia). Imaging was performed on the Nikon ECLIPSE Ti2 microscope using NIS-Elements AR software (version 5.41.01) or a Leica SP8 with LAS AF v3.x software for confocal images. Pictures were analyzed using ImageJ.

For whole-mount samples of PLP1^CreERT2Ai14^fl/fl mice and GFAP^CreAi14^fl/fl mice, the terminal colon was opened longitudinally, fixed with 4% PFA for 30 min, and washed with PBS. Muscularis externa was peeled off the colonic tissue and blocked with blocking buffer consisting of PBS containing 0.02% Sodium azide (Sigma-Aldrich), 0.3% donkey serum (Jackson), and 3% BSA (Serva) for 2 h at RT. Subsequently, samples were incubated 2 ONs at 4 °C in staining buffer consisting of blocking buffer supplemented with 0.3% Triton X-100 (ThermoFisher, Scientific) with following primary antibodies for PLP1^CreERT2Ai14^fl/fl mice: rabbit anti-GFAP (1:300, Dako) and goat anti-SOX10 (1:300, R&D Systems), and for GFAP^CreAi14^fl/fl mice: chicken anti-GFAP (1:1000, Biolegend) and goat anti-SOX10 (1:1000, custom-made aliquot kindly provided by Prof. Wegner, University of Erlangen[58]). Then, samples were washed in PBS and incubated with DAPI (Sigma-Aldrich) combined with the secondary antibodies in staining buffer ON at 4 °C. Secondary antibodies used for PLP1^CreERT2Ai14^fl/fl mice; donkey anti-rabbit Cy5 (1:400, Jackson) and donkey anti-goat AF488 (1:500, Invitrogen, ThermoFisher Scientific) and for GFAP^CreAi14^fl/fl mice: donkey anti-chicken FITC (1:800, Jackson) and donkey anti-goat CF647 (1:800, Sigma). Lastly, samples were rinsed three times in PBS and mounted with SlowFade Diamond Antifade mounting (Invitrogen, ThermoFisher Scientific). Imaging was performed on the ZEISS LSM 880 confocal microscope (PLP1^CreERT2Ai14^fl/fl) or Leica SP8 with LAS AF v3.x software (GFAP^CreAi14^fl/fl) and the pictures were analyzed using ImageJ.

### Enzyme-linked immunosorbent assay (ELISA)

Healthy and tumor-conditioned medium EGCs supernatants were collected and analyzed for IL-6 and IL-1β content using sensitive commercial ELISA kits (R&D Systems, Minneapolis, MN and V-Plex Proinflammatory panel Meso Scale Discovery; MSD respectively) according to the manufacturer's instructions. The data were analyzed with the Discovery Workbench 4.0 software (MSD).

### Western blot

Total proteins were extracted from mouse colonic tissues in T-PER buffer (ThermoFisher, Scientific) supplemented with 1 mM dithiothreitol, 10 mg/mL aprotinin, 10 mg/mL leupeptin, 1 mM phenylmethylsulfonyl fluoride, 1 mM Na3VO4 and 1 mM NaF (all from ThermoFisher, Scientific), by homogenization for 1 minute at 30 Hz (TissueLyser II, Qiagen). Lysates were clarified by centrifugation at 4 °C, 12,000 g for 30 min and separated on sodium dodecyl sulfate (SDS)-polyacrylamide gel electrophoresis. Blot was incubated with the GFAP antibody (1:500, Cell Signaling) followed by a secondary antibody conjugated to horseradish peroxidase (HRP) (1:5000; Dako Agilent Technologies). HRP was detected using the SuperSignal West Dura Extended Duration Substrate kit (ThermoFisher Scientific). To ascertain equivalent loading of the lanes, the blot was stripped and incubated with an anti-vinculin antibody (1:5000, Sigma-Aldrich) followed by a secondary antibody conjugated to horseradish peroxidase (1:5000; Dako Agilent Technologies) and HRP detection as described above. Computer-assisted scanning densitometry (GE Healthcare ImageQuant LAS 4000 Luminscent Image Analyzer) was used to analyze the intensity of the immunoreactive bands.

### Liquid chromatography/mass spectrometry (LC/MS)

LC/MS analysis of adult neurosphere-derived EGC supernatants treated with or without IL-1β was performed by the Core Facility Analytical Proteomics of the University of Bonn as described in the following. All chemicals from Sigma unless otherwise noted. For LC/MS sample preparation, 70 μg of protein per sample was subjected to in-solution preparation of peptides with the iST-NHS 96x sample preparation kit (Preomics GmbH, Martinsried, Germany) according to the manufacturer's recommendations. 0.4 mg TMT10plex isobaric Mass Tag Labeling reagent (Thermo Scientific) was added to each sample and incubated at room temperature for 1 h. 10 μL 5% hydroxylamine was used to quench the reaction. The preparation procedure was continued according to the iST-NHS kit instructions. Peptide concentration was determined with a colorimetric peptide assay (Thermo Scientific). Equal amounts of peptides were pooled and dried in a vacuum concentrator, dissolved in 20 mM ammonium formate (pH 10) and fractionated by reversed phase chromatography at elevated pH with a Reprosil100 C18 column (3 μm 125 × 4 mm, Dr. Maisch GmbH, Ammerbuch-Entringen, Germany). 60 fractions were combined into 6 pools and dried in a vacuum concentrator.

Before measurement, peptides were re-dissolved in 0.1% formic acid (FA) to yield a 1 g/L solution and separated on a Dionex Ultimate 3000 RSLC nano HPLC system (Dionex GmbH, Idstein, Germany). The autosampler was operated in μL-pickup mode. 1 μL was injected into a C18 analytical column (self-packed 400 mm length, 75 μm inner diameter, ReproSil-Pur 120 C18-AQ, 1.9 μm, Dr. Maisch). Peptides were separated during a linear gradient from 5% to 35% solvent B (90% acetonitrile, 0.1% FA) at 300 nL/min during 150 min. The nano-HPLC was coupled online to an Orbitrap Fusion Lumos Mass Spectrometer (Thermo Fisher Scientific, Bremen, Germany). Peptide ions between 330 and 1600 m/z were scanned in the Orbitrap detector every three seconds with a resolution of 120,000 (maximum fill time 50 ms, AGC target 100%). From MS3-based quantification, peptides were subjected either to collision-induced dissociation for identification (CID: 0.7 Da isolation, normalized energy 30%) and fragments analyzed in the linear ion trap with AGC target 50% and a maximum fill time 35 ms, rapid mode. Fragmented peptide ions were excluded from repeat analysis for 30 s. The top 10 fragment ions were chosen for synchronous precursor selection and fragmented with higher energy CID (HCD: 3 Da MS2 isolation, 65% collision energy) for detection of reporter ions in the Orbitrap analyzer (range 100–180 m/z, resolution 50,000, maximum fill time 86 ms, AGC target 200%). Alternatively, peptides were only fragmented by HCD and fragment ions and reporter ions were analyzed in the same spectrum (Orbitrap resolution 50,000).

Raw data processing and database search were performed with Proteome Discoverer software 2.5.0.400 (Thermo Fisher Scientific). Peptide identification was done with an in-house Mascot server version 2.8.1 (Matrix Science Ltd, London, UK). LC/MS data were searched against the Uniprot reference proteome mouse database (2022/05, 63628 sequences) and contaminants database (cRAP1)[59]. Precursor Ion m/z tolerance was 10 ppm, fragment ion tolerance 0.5 Da (CID). Tryptic peptides with up to two missed cleavages were searched. $C_6H_{11}NO$-modification of cysteines (delta mass of 113.08406) and TMT10plex on N-termini and lysines were set as static modifications. Oxidation was allowed as dynamic modification of methionine. Mascot results were evaluated by the Percolator algorithm version 3.02.12 as implemented in Proteome Discoverer[60]. Spectra with identifications above 1% $q$ value were sent to a second round of database search with semi-tryptic enzyme specificity (one missed cleavage allowed). Protein N-terminal acetylation, methionine oxidation, TMT10plex, and cysteine alkylation were then set as dynamic modifications. Actual false discovery rates (FDR) values were 0.2% (peptide spectrum matches) and 0.9% (peptides). Reporter ion intensities (most confident centroid) were extracted from the MS3 level, with SPS mass match >65%.

The statistical analyses of the peptide-spectrum match (PSM) level data were done by the Core Unit for Bioinformatics Data Analysis of the University of Bonn. Analyses were carried out in R environment (R version 4.2) using an in-house developed workflow. Non-unique peptides and single-hit proteins (proteins identified/quantified by only one peptide) were filtered-out prior to the statistical analysis. From all available fractions, only those with the least number of missing values per feature and maximum average intensity across all TMT labels were selected. The PSM-level data were then log-transformed and scaled such that all the samples have the same median values (median normalization method). Next, the normalized data was aggregated to protein-level by applying the Tukey's median polish method. The statistical analysis was performed using the R package limma[61]. For each statistical contrast, the resulting $P$ values were adjusted for multiple testing. The FDR were calculated by the Benjamini-Hochberg method.

## Isolation of tumor-infiltrating cells

Tumor-bearing mice were sacrificed at the described time points. After peeling off the muscularis layer from the orthotopic tumors, tissues were first cut into 1 mm pieces and then went under mechanical and enzymatic digestion for 30 min in DMEM with 2.5% FBS, 100 µg/mL Penicillin and Streptomycin, 200 U/mL collagenase IV (Gibco, ThermoFisher Scientific) and 125 µg/mL type II dispase (Gibco, Thermo-Fisher Scientific). AOM/DSS-induced tumors and healthy colon samples were peeled off the muscularis layer and underwent epithelial removal by vigorous shaking in Hanks' balanced salt solution (HBSS) with phenol red (Gibco, ThermoFisher Scientific) containing 1% FBS, 100 µg/mL Penicillin and Streptomycin, 1 mM EDTA (Invitrogen, ThermoFisher Scientific) and 1 mM dithiothreitol (DTT) (Sigma-Aldrich) for 8 min at 37 °C. A second incubation step was performed for 8 min at 37 °C in the same medium without DTT. After washing in wash medium (DMEM with 2.5% FBS and 100 µg/mL Penicillin and Streptomycin), the remaining tissue was cut into small pieces and digested for 30 min at 37 °C in pre-warmed alpha Minimum Essential Medium (MEM) (Lonza) containing 5% FBS, 100 µg/mL Penicillin and Streptomycin, 5 U/mL DNase (Roche), 1 mg/mL dispase (Gibco, ThermoFisher Scientific), 1.25 mg/mL Collagenase D (Roche) and 0.85 mg/mL Collagenase V (Sigma-Aldrich). For FACS sorting of tdTomato$^{pos}$ and tdTomato$^{neg}$ cells, tumors of AOM/DSS-treated mice were cut out and collected, while non-tumorous tissue was excluded. Naive and tumor tissues were enzymatically digested with 0.15 U/ml Liberase™ TH (Roche) and 0.15 mg/ml DNase (DN25, Sigma) in HBSS for 35 min at 37 °C in a shaking water bath. Independent of tumor origin, cells were then filtered through a 70-µm cell strainer (BD Falcon), washed with PBS, and stained with fluorophore-conjugated antibodies.

## FACS staining and analysis

Single-cell suspensions (obtained as described above) were incubated for 15 min with mouse FcR Blocking Reagent (1:100 BD Pharmingen) at 4 °C. Next, cells were stained for surface markers (see Supplementary Table 1 for antibodies list) and incubated for 20 min incubation at 4 °C. Then cells were washed with FACS buffer (0.5% FBS and 2 mM EDTA in PBS) and resuspended in FACS buffer containing the viability marker 7-AAD (1:100 BD Pharmingen) before filtering through a 70-µm strainer.

For the intracellular measurement of IL-1α and IL-1β, single-cell suspensions were pre-cultured in DMEM with 2.5% FBS, 100 µg/mL Penicillin and Streptomycin and stimulated with BD GolgiStop™ (1:1000, BD Biosciences) for 4 h in 5% $CO_2$ at 37 °C followed by a pre-incubation with the viability dye eFluor 506 (1:400 eBioscience) for 20 min at 4 °C. Then cell suspensions were washed, blocked with FcR Blocking Reagent (1:100 BD Pharmingen) and stained with surface antibodies (see Supplementary Table 1 for antibodies list) as described above. After a washing step with FACS buffer, cells were incubated for 45 min in Fix/Perm buffer (eBioscience, Invitrogen, ThermoFisher Scientific), followed by 5 min incubation in 1X Permeabilization buffer (eBioscience, Invitrogen, ThermoFisher Scientific). Next, the cells were stained for a minimum of 1 h in 1X Permeabilization buffer containing FcR Blocking Reagent (1:600 BD Pharmingen) and intracellular markers (see Supplementary Table 1 for antibody details). Cells were subsequently washed and resuspended in a Permeabilization buffer before filtering through a 70-µm strainer.

For cell counting goals, counting beads (1:100 Spherotech) were added per sample. Flow cytometry analyses were performed on a BD Symphony A5 Cell Analyzer (BD Biosciences) and subsequently analyzed using FlowJo v.10.6.1.

## FACS sorting of EGCs

FACS-sorting of tdTomato$^{pos}$ and tdTomato$^{neg}$ cells was performed on a BD FACSAria III cell sorter using BD FACSDiva 9.0.1 software (both BD Biosciences). A non-fluorescent sample was used as negative control to determine background fluorescence. Sorted cells were collected, centrifuged and pellets were snap frozen. RNA isolation was done using TRIzol™ reagent (Thermo Fisher Scientific) and the RNeasy Micro Kit (Qiagen) according to the manufacturers' instructions.

## Tumor-infiltrating monocyte sorting for EGCs stimulation

Tumor-infiltrating monocytes were sorted from orthotopic tumors based on the expression of the viability marker 7-AAD, CD45, CD64, Ly6C, MHCII, SiglecF and Ly6G (see Supplementary Table 1 for antibodies details) using a Sony MA9000 sorter. Next, $1 \times 10^5$ tumor or BM-derived monocytes were cultured in a complete DMEM medium overnight in 5% $CO_2$ at 37 °C. The conditioned medium of these monocytes was collected and used to stimulate primary embryonic neurosphere-derived EGCs ($5 \times 10^4$ cells/mL) in the presence of 5 µg/mL IgG (BioXCell) or 5 µg/mL anti-IL-1R (BioXCell) for 24 h in 5% $CO_2$ at 37 °C.

## RNA extraction and gene expression

RNA was isolated using the innuPREP RNA Mini Kit (Analytik Jena) or RNeasy Mini Kit (Qiagen) for tissue and high cell numbers or RNeasy Plus Micro Kit (Qiagen) for low cell numbers according to the manufacturer's instructions. Dependent on RNA concentrations, total RNA was transcribed into cDNA by the qScript™ cDNA SuperMix (Qianta-Bio) or the High-Capacity cDNA Reverse Transcription Kit (Thermo-Fisher Scientific) according to manufacturer's instructions. qRT-PCR was performed with the LightCycler 480 SYBR Green I Master (Roche) on the Light Cycler 480 (Roche) or with the PowerSYBR Green PCR

Master Mix (Thermo Fisher Scientific) on the QuantStudio5 cycler (Thermo Fisher Scientific) (primers listed in Supplementary Table 2). Results were quantified using the $2^{-\Delta Ct}$ method or were applicable to the $2^{-\Delta\Delta Ct}$ method, usage stated by "relative to". The expression levels of the genes of interest were normalized to the expression levels of the reference gene *Rpl32*.

## Bulk RNA sequencing

For Bulk RNA-seq of the in vitro tumor EGCs model, total RNA from in vitro generated unstimulated, H-CM and TME-CM primary embryonic neurosphere-derived EGCs was provided to the Genomics core (KU Leuven). QuantSeq 3' mRNA libraryprep (015, Lexogen) was used to generate cDNA libraries, followed by sequencing on the HiSeq4000 system. Quality control of raw reads was performed with FastQC v0.11.7 (Andrews S. FastQC: a quality control tool for high throughput sequence data. Available online at: http://www.bioinformatics.babraham.ac.uk/projects/fastqc, 2010.). Adapters were filtered with ea-utils fastq-mcf v1.05 (Erik Aronesty. ea-utils: Command-line tools for processing biological sequencing data. Available online at: https://github.com/ExpressionAnalysis/ea-utils, 2011.). Splice-aware alignment was performed with HISAT2[62] against the reference genome mm10 using the default parameters. Reads mapping to multiple loci in the reference genome were discarded. Resulting BAM alignment files were handled with Samtools v1.5[63]. Quantification of reads per gene was performed with HT-seq Count v0.10.0, Python v2.7.14[64]. Count-based differential expression analysis was done with R-based (The R Foundation for Statistical Computing, Vienna, Austria) Bioconductor package DESeq2[65]. Reported *p* values were adjusted for multiple testing with the Benjamini-Hochberg procedure, which controls FDR. Data visualization was prepared using ggplot2 R package (v3.4.1) or pheatmap (v1.0.12).

For 3'mRNA sequencing of naive and AOM/DSS-treated mice, isolated RNA was provided to the Genomics Core Facility of the University Hospital Bonn, which performed library preparation using QuantSeq FWD 3'mRNA-Seq kit (Lexogen) according to the manufacturer's instructions. Sequencing was performed on the NovaSeq6000 with a sequencing depth of 10 M raw reads. Data were analyzed using PartekFlow software (V10.0.23.0720) available from https://www.partek.com/partek-flow/#features with the Lexogen12112017 pipeline and Ensemble transcripts release 102 for mm10 mouse alignment. Briefly, two adapter-trimming steps and one base-trimming step were performed before alignment was done by star2.5.3a. Post-alignment QC was performed and reads were quantified to the annotation model. Gene counts were normalized and data were further analyzed by gene-specific analysis. Visualization was done with PartekFlow software (V10.0.23.0720) and GraphPad Prism 6 (V6.07).

To quantify the transcripts per million (TPM) values from the bulk RNA-seq data of adult derived EGCs (previously published in refs. 15,17, PartekFlow software (V10.0.23.0720) available from https://www.partek.com/partek-flow/#features. The TPM values for bulk RNA-seq of embryonic neurosphere-derived EGCs were obtained by processing fastq files using nf-core/rnaseq pipeline (nf-co.re/rnaseq/3.12.0) where GRCh37 was used as reference genome and Salmon was used for quantification[66].

## Weighted gene correlation network analysis (WGCNA)

First, variance stabilizing transformation was performed on the bulk RNA-seq data generated from unstimulated, H-CM and TME-CM primary neurosphere-derived EGCs using the DESeq2[65] package in R (v4.2.2). PCA was performed after variance stabilizing transformation using DeSeq2 package by selecting samples stimulated for 24 h. The function prcomp() was used for obtaining the gene strength towards each PC. Next, WGCNA[67] was performed using the R package WGCNA (v1.72.1). To distinguish the modules with different expression

patterns, a soft threshold power of 12, which was the lowest power for the scale-free topology fit index of 0.85, was selected to produce a hierarchical clustering tree (dendrogram). The function "blockwiseModules" was used for automatic block-wise network construction and module identification. The number of modules was detected automatically by the algorithm, with the number of genes in a module limited to between 30 and 5000 genes. The co-expression networks were created based on the similarity of expression patterns of genes and the networks were established by merging genes with similar co-expression patterns into modules. The clusterProfiler[68] package was used to implement enrichGO() for gene ontology over-representation test using the genes of modules 4, 7, and 8.

## Single-cell RNA sequencing of orthotopic murine tumors

Cell suspensions of orthotopic murine tumors were processed with a 10x Chromium Next GEM Single Cell 5' kit and loaded on a 10x chromium controller to create Single Cell Gel beads in Emulsion (GEM). A cDNA library was created using a $10 \times 5'$ library kit and was then paired-end sequenced on an Illumina Novaseq device following 10x's guidelines (https://www.10xgenomics.com/support/single-cell-immune-profiling/documentation/steps/sequencing/sequencing-requirements-for-single-cell-v-d-j). Sample demultiplexing and data analysis was performed using 10x's Cellranger suite (https://support.10xgenomics.com/single-cell-vdj/software/pipelines/latest/using/vdj) using the standard parameters.

## scRNA-seq clustering and dimensionality reduction

The count matrices obtained after pre-processing with Cellranger were concatenated to obtain a combined raw count matrix which was then analyzed using the Seurat R package (v3.1.3). Cells with less than 300 or more than 6000 genes and cells with more than 15% mitochondrial genes were discarded from the analysis. Normalization and scaling were done with default variables with top variable genes identified using FindVariableFeatures function. After principal component analysis, 1st 39 principal components were used based on the elbow plot for creating a nearest neighbor graph using FindNeighbours function in Seurat. After clustering at a resolution of 1, clusters were classified into immune and non-immune clusters. Six small doublet clusters with markers of two or more distinct cell types were removed. Also, two clusters with low nUMI and lacking distinguishing markers of any cell types were removed. A subset of Seurat Object with immune clusters alone was created and the same pipeline was followed from Normalization to Clustering (number of principal components used = 32). After clustering at resolution 1, the clusters were manually annotated following Zhang et al.[24]. Clusters annotated as monocytes or macrophages were re-clustered similarly to identify the subclusters. These sub-clusters were annotated manually based on the expression of Ly6c2, Ccr2, H2-Ab1, Spp1, C1qa, Cx3cr1, and Mki67. Further to learn potential differentiation trajectory, Monocle-3 was used. (Parameters: *n* center = 300, minimal branch length = 10, nn.k = 20). Genes upregulated or downregulated in SPP1+ TAMs compared to C1Q+ TAMs were functionally annotated using universal enrichment function "enricher" from the "ClusterProfiler" package (v4.6.0) with a gene annotation database aggregation containing all terms from Human Phenotype, Transcription factor, and Hallmark from Molecular Signature Database (MSigDB), BIOCARTA, REACTOME, GO and KEGG Markers for different clusters were determined using a Wilcoxon rank sum test with FindMarkers or FindAllMarkers functions in Seurat.

## Inferring cell–cell communication using NicheNet

NicheNet (nichenetr R package; v1.1.0) was used to study the interactions between EGCs and tumor-infiltrating monocytes. To identify TME EGC-derived ligands potentially inducing the differentiation of monocytes into SPP1+ TAMs, bulk RNA-seq data from in vitro TME-CM EGCs was used. Ligands were identified after filtering for genes upregulated in 24 h time point TME-CM EGCs with respect to 24 h

timepoint H-CM EGCs (adjusted $p$ value < 0.05). Genes differentially expressed between SPP1$^+$ TAMs and monocytes (adjusted $p$ value < 0.05) were considered as the gene set of interest.

NicheNet was also used to study the interaction between tumor-infiltrating immune cells and EGCs. To identify immune cell-derived ligands potentially inducing differentiation of 24 h time point H-CM EGCs into 24 h time point TME-CM EGCs, scRNA-seq data of the immune compartment of the in vivo murine orthotopic CRC model was used. Using get_expressed_genes function from NichenetR, genes expressed in at least 5 % of immune cell clusters were considered as potential ligands. Genes differentially expressed between TME-CM EGCs and H-CM EGCs (adjusted $p$ value < 0.001) were considered as the gene set of interest.

## Bio-informatic analysis: KUL3 dataset

Bio-informatic analysis of the CRC tumor microenvironment of patients with CRC was performed making use of the published KUL3 dataset[33]. Integration of the data, dimensionality reduction, unsupervised clustering and differential gene expression analysis was performed in R using Seurat with SCTransform - Integration pipeline. For downstream analysis, "border" and "tumor" samples were taken together and considered as tumor samples. Patient KUL31 was excluded from all EGCs analysis, due to extremely low cell numbers. Pathway enrichment analysis was done using Ingenuity pathway Analysis (IPA, Qiagen). Modules identified using WGCNA on mouse bulk RNA-seq data was converted to one-to-one human orthologs and then used for single-sample Gene Set Enrichment Analysis GSEA as implemented in single-cell Gene Set Variation Analysis R package (v0.0.11).

## Bio-informatic analysis: TCGA and LUMC AC-ICAM dataset

The processed gene expression RNA-seq (IlluminaHiSeq) data of the Cancer Genome Atlas (TCGA) colorectal adenocarcinoma (COAD-READ) was downloaded from the University of California SantaCruz (UCSC) Xena using the UCSCXenaTools (https://doi.org/10.21105/joss.01627) R library. The details of data integration and processing are described in UCSC-Xena browser (https://xenabrowser.net/). The clinical information and overall survival (OS) data of the patients were also obtained using UCSCXenaTools (data subtype: "phenotype"). According to their age, the patients were classified as above 65 years (>=65) and below 65 years. Patients with tumor stage I and IA were clustered as stage I, patients with stage II, IIA, IIB as stage II, patients with stage III, IIIA, IIIB, IIIC as stage III, and patients with stage IV, IVA, IVB as stage IV. Patients with microsatellite stability were classified as MSS and patients with microsatellite instability high and low as MSI. The consensus molecular subtypes (CMS) were predicted using the R package CMSClassifier (v1.0.0) and the iCMS classification of the patients was performed as previously described[69]. The 376 patients with CRC were hierarchically clustered according to the high and low expression patterns of the specific gene signatures (Supplementary Table 3) in the tumor samples which resulted in 2 patient categories (EGC high and EGC low). The R packages survival (v3.5.5) and survminer (v0.4.9) were used for survival analysis and plotting the Kaplan-Meier (KM) survival curve. A statistically significant difference in survival was indicated by a log-rank test p-value of $p < 0.05$. Survival analysis with univariate and multivariate proportional hazards regression models (Cox regression) was performed to adjust for age, gender, radiation therapy, stage and EGC signature expression. The R packages pheatmap, gtsummary, and ggplot2 were used for visualization.

The second CRC cohort used was the Sidra-LUMC AC-ICAM dataset, published by Roelands et al.[35]. The 348 patients were hierarchically clustered and survival analysis was performed as described above.

## Reporting summary

Further information on research design is available in the Nature Portfolio Reporting Summary linked to this article.

## Data availability

The scRNA-seq (related to Fig. 3 and Supplementary Fig. 4) and bulk RNA-seq (related to Fig. 2 and Supplementary Fig. 3 and 5b) data generated for this study are deposited in the Gene Expression Omnibus (GEO) database under the GEO accession code GSE231804. The bulk RNA-seq (related to Supplementary Fig. 6), data generated for this study are deposited in the Gene Expression Omnibus (GEO) database under the GEO accession code GSE231709. The mass spectrometry proteomics data generated for this study (related to Fig. 4) have been deposited to the ProteomeXchange Consortium via the PRIDE partner repository with the dataset identifier PXD045911. Bulk RNA-seq data (related to Supplementary Fig 5b) were published previously[15,17] and can be accessed via GSE205610 and GSE134943. The remaining data are available within the Article, Supplementary Information or Source Data file. Source data are provided in this paper.

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

## Acknowledgements

We acknowledge all members of Prof. Matteoli's laboratory and Prof. Wehner's laboratory for the support and scientific discussions. We would like to thank Tine Gomers and Karlien Vranken (TARGID, KU Leuven) and Patrik Efferz and Bianca Schneiker (Department of Surgery, University Hospital Bonn) for technical assistance during experiments. Furthermore, we would like to thank Ally Peddle and Yourae Hong (Molecular Digestive Oncology, Department, KU Leuven), Lukas Ferreira Maciel (Laboratory for Molecular Cancer Biology, VIB-KU Leuven) and Florent Petitprez (MRC Centre for Reproductive Health, University of Edinburgh) for their scientific support on bio-informatic analysis. Within KU Leuven we would like to acknowledge the following core facilities: FACS Core, Genomics Core (UZ Leuven), LiMoNe VIB Bioimaging Core and VIB Center for Brain & Disease Research. We would like to thank the Cell and Tissue Imaging Cluster (KU Leuven) for the usage of the Zeiss LSM 880 – Airyscan (supported by Hercules AKUL/15/ 37_GOH1816N and FWO G.0929.15 to Prof. Pieter Vanden Berghe). We would like to thank the support from the Core Facilities of the Medical Faculty, University of Bonn, specifically, the Analytical Proteomics Core, funded by the Deutsche Forschungsgemeinschaft (DFG)—project 386936527, the Bioinformatics Data Analysis Core, the Microscopy Core funded by the DFG – project 266686698, the Flow Cytometry Core funded by the DFG —project 216372545, and the Next Generation Sequencing Core. We thank Dr. Vassilis Pachnis for sharing the SOX10$^{CreERT2}$ mice, Prof. Juan Hidalgo for sharing the IL-6$^{fl/fl}$ mice and Prof. Michael Wegner for sharing the Sox10 antibody. Figures 1a, 1b, 1e, 2a, 3e, 3g, 3j, 4a, 4d, 4f, 5a, 5c, 6a, 6e, 6h, 7a, S1f, S1o, created with BioRender.com, released under a Creative Commons Attribution-NonCommercial-NoDerivs 4.0 International license. We would like to acknowledge Prof. Owen Sansom for providing us with the AKPT CRC cell lines. V.D.S. was supported by a Stichting Tegen Kanker postdoctoral fellowship. S.S. was supported by KU Leuven-University of Melbourne Global PhD (GPUM/22/020). F.B. was supported by KU LEUVEN INTERNAL FUNDS KU Leuven Global PhD Partnerships with the University of Edinburgh (GPUE/20/003). B.K. was supported by the Taiwan - KU Leuven PhD Scholarship. M.V. was supported by a Fonds voor Wetenschappelijk Onderzoek Vlaanderen (FWO, 11L0822N) PhD fellowship. G.B. is funded by ERC Advanced grant no. 833816-NEUMACS. S.I. was supported by an MSCA-IF (79756–GLIAMAC) and a fellowship from the European Crohn's and Colitis Organization (ECCO). S.V. and S.T. were supported by the FWO grant G067821N and Stichting tegen Kanker grant F/2020/1512. Research in ADG lab is supported by Research Foundation Flanders (FWO) (Fundamental Research Grant, G0B4620N), KU Leuven (C3 grant, C3/21/037 or C3/23/067), and VLIR-UOS (iBOF grant, iBOF/21/048, for 'MIMICRY' consortium). R.S.L. is supported by FWO-SB PhD Fellowship (1S44123N). G.M.'s lab was supported by FWO grants G0D8317N, G0A7919N, G086721N, G088816N, S008419N and W001620N, KU Leuven Internal Funds (C12/15/016 and C14/17/097). S.W. and L.S. were supported by the DFG-funded Immunosensation[2] cluster of excellence EXC2151-190873048. S.W. and R.S. received funding from BONFOR. B.G.R. received funding from the Deutsche Krebshilfe through a Mildred Scheel Nachwuchszentrum Grant (70113307).

## Author contributions

Conceptualization, L.V.B., V.D.S., L.S., R.S., S.W. and G.M.; Methodology, L.V.B., V.D.S., L.S., S.W. and G.M.; Software, S.S., S.A., Z.H., J.H., L.V.B. and L.S.; Validation, L.V.B., V.D.S. and L.S.; Formal Analysis, L.V.B., V.D.S., L.S., S.S. and S.A.; Investigation, L.V.B., V.D.S., L.S., S.S., S.A., F.B., L.Z., S.V., B.K., B.G.R., M.T., M.V., R.S.D.M., N.S., R.S.L. and S.I.; Resources, G.M., S.W., S.T., G.B, A.D.G. and F.D.S.; Data Curation, L.V.B., V.D.S., L.S., S.S., S.A., S.V., Z.H. and J.H.; Writing—original draft, L.V.B., V.D.S. and L.S.; Writing—review & editing, all; Visualization, L.V.B., V.D.S., L.S., S.S., S.A., and Z.H.; Supervision, G.M., S.W., S.T., F.D.S., R.S., M.S. and V.D.S.; Project administration, L.V.B., V.D.S., L.S., R.S., S.W. and G.M.; Funding acquisition, V.D.S., S.I., G.M. and S.W.

## Competing interests

The authors declare no competing interests.

## Additional information

[1]Laboratory of Mucosal Immunology, Department of Chronic Diseases and Metabolism (CHROMETA), Translational Research Center for Gastrointestinal Disorders (TARGID), KU Leuven, Leuven, Belgium. [2]Department of Surgery, University Hospital Bonn, Medical Faculty, Bonn, Germany. [3]Department of Anatomy and Physiology, University of Melbourne, Parkville, VIC, Australia. [4]Centre for Inflammation Research, University of Edinburgh, Edinburgh, UK. [5]Department of Biology and Biotechnology "L. Spallanzani", University of Pavia, Pavia, Italy. [6]Digestive Oncology, Department of Oncology, KU Leuven, Leuven, Belgium. [7]Laboratory for Intestinal Neuro-Immune Interaction, Department of Chronic Diseases and Metabolism (CHROMETA), Translational Research Center for Gastrointestinal Disorders (TARGID), KU Leuven, Leuven, Belgium. [8]Cell Stress and Immunity (CSI) Lab, Department of Cellular and Molecular Medicine, KU Leuven, Leuven, Belgium. [9]Mildred Scheel School of Oncology, Aachen Bonn Cologne Düsseldorf (MSSO ABCD), University Hospital Bonn, Medical Faculty, Bonn, Germany. [10]Department of Pathology, University Hospital Bonn, Medical Faculty, Bonn, Germany. [11]Translational Cell and Tissue Research Unit, Department of Imaging & Pathology, Laboratory for Precision Cancer Medicine, KU Leuven, Leuven, Belgium. [12]Leuven Institute for Single-Cell Omics (LISCO), KU Leuven, Leuven, Belgium. [13]Laboratory of Cell Biology & Histology, Department of Veterinary Sciences, University of Antwerp, Antwerp, Belgium. [14]These authors contributed equally: Lies van Baarle, Veronica De Simone, Linda Schneider. [15]These authors jointly supervised this work: Sven Wehner, Gianluca Matteoli. ✉e-mail: sven.wehner@ukbonn.de; gianluca.matteoli@kuleuven.be

