## [Peer Review File · Nature Communications]

IL-1R signaling drives enteric glia-macrophage interactions in colorectal cancerEditorial note: In this document, the figures that were created with BioRender.com, are released under a Creative Commons Attribution-NonCommercial-NoDerivs 4.0 International license".

REVIEWER COMMENTS

Reviewer #1 (Remarks to the Author): with expertise in bioinformatics, cancer immunology

This study investigates the role of enteric glial cells (EGCs) in colorectal cancer (CRC) development. It primarily focuses on murine models of CRC, using both orthotopic tumor injections and in vitro co-culture with primary cells and multiple mutants to deplete particular cell types or molecules. The final figure extends the mouse results to analyse human patient cohorts to provide preliminary results that the same phenomenon occurs in humans and that it may be associated with prognosis. The primary conclusion is that there exists a positive feedback loop between monocytes/macrophages and EGCs contributing to poor prognosis CRC. Specifically, that a) monocytes and macrophages are the primary source of IL1 in the tumor microenvironment, b) IL1 acts on EGCs, inducing a protumoral/reactive phenotype, c) these reactive EGCs are the primary source of IL6 which drives monocytes towards a SPP1+ tumor associated macrophage phenotype which are known to drive CRC towards a bad prognosis "mesenchymal" state through diverse mechanisms. The manuscript is clearly written and easy to follow and the experiments are well designed. EGCs are currently underexplored in the literature so this study is a timely contribution.

Major comments

1. The PLP1CreERT2iDTR mouse clearly depletes EGCs, however it is important to show that this is specific – ie that there is not leaky expression in other cell types which then also get depleted by DT injection.
2. More details need to be provided about the survival analyses in Fig 7B,G to verify their robustness. Were the patients split by median values for the gene signatures ? Or by some bi-modality of the gene signature distribution ? If the (unfortunately common) practice of sweeping through splits to identify the value which gives the most significant difference in

outcome was performed, this is not meaningful unless it is validated in a second, independent cohort.

3. The authors nicely show that in vivo, monocytes and macrophages appear to be the major source of IL-1 in the tumor microenvironment. They also show that reactive EGCs produce IL-6 and that IL-6 drives monocytes/TAMs towards the SPP1+ TAM state. It would thus be important to determine whether reactive EGCs are the primary source of IL-6 in the TME or not.

Minor comments

1. In Fig 2B the separation of TLE-CM from Unstimulated and H-CM is very striking. As a complement to the WGCNA analysis in panels C and D, it would be nice to show the genes with the biggest (positive or negative) contribution to PC2, eg in the supplemental.

2. The selection of genes for figure 2D is not entirely clear. The methods says that WGCNA modules were restricted to have between 30 and 5000 genes so this must be a subset of modules 4, 7 and 8. How were they chosen. Doing a gene set enrichment analysis of modules 4, 7, and 8 seems straightforward and potentially more informative than highlighting/interpreting just 2 genes per module as is done in the text on page 7.

Reviewer #2 (Remarks to the Author): with expertise in enteric glia, CRC

This is a very compelling study demonstrating that, in response to monocytes of the tumor microenvironment, enteric glial cells undergo a switch of phenotype that influences tumor-associated macrophages differentiation. The identification of the cells of the enteric nervous system as key components of the tumor microenvironment functionally controlling colon cancer development and progression is a major emerging research area in the field of colon cancer. This paper provides powerful evidence that enteric glial cells engage in functional interactions with monocytes and macrophages of the tumor microenvironment, shaping the cellular composition of the latter. The manuscript is very well written and has a nice flow. The experimental design is creative and pertinent, and the data are very convincing and nicely presented. My only major recommendation is to further characterize the data obtained using the murine orthotopic colon cancer model in DT-treated PLP1DTA mice versus controls. Other minor concerns are detailed and listed below.

The murine orthotopic colon cancer model needs to be better presented and/or characterized. Basically, my major concern is whether or not the difference in tumor volume between DT-treated PLP1DTA mice and controls shown in Fig 1C originates from a difference in tumor epithelial cell proliferation or in immune cell recruitment, as this would change the interpretation of this experiment. This concern is furthered by Fig S1 that shows a major immune infiltrate but not a tumor (epithelial) mass. I would recommend co-staining orthotopic tumors with EpCAM (or equivalent), GFAP, F4/80 and a marker of proliferation. In addition, using a bioluminescent or fluorescent reporter stably expressed in tumor epithelial cells would allow for the discrimination of grafted tumor epithelial cells versus host colonic epithelial cells and prove expansion of grafted MC38 cells. While Fig S1B shows the proportion of immune cells vs other cells in orthotopic tumors vs healthy tissue, this does not demonstrate whether the difference in tumor size between DT-treated PLP1DTA mice and controls comes from tumor epithelial cell proliferation or immune cell recruitment. Other questions are: was Fig S1B flow cytometry? were PLP1DTA control mice injected with saline? Figure S1C shows a decrease in GFAP expression in DT-treated PLP1DTA mice at day 0 (when MC38 cells are injected) which is five days after DT injection. Baghdadi et al. (2022; PMID: 34727519) found a small decrease in GFAP expression at one week after PLP1+ cell ablation and an upregulation of GFAP in PLP1DTA mice after two weeks of tamoxifen treatment, suggesting that there is a major increase in GFAP expression between week 1 and week 2. As only day 0 is shown here, it is difficult to interpret whether or not the growth of the orthotopic tumors may have been concomitant with an increase in GFAP+ expressing cells? While the findings from Baghdadi and colleagues are acknowledged in the manuscript, this needs to be further investigated. Can GFAP expression be investigated at day 7? Can other markers of enteric glia be tested to validate these findings?

The study uses both embryonic neurosphere-derived enteric glial cells and adult neurosphere-derived enteric glial cells, and it is unclear why one type of culture is used over the other. A rationale should be provided. Why not consistently use adult neurosphere-derived enteric glial cells -which seem more appropriate in the context of colon cancer?

Minor: Figure 3H shows that enteric glial cells activate monocyte differentiation into C1Q+

macrophages in response to healthy conditioned medium while they activate monocyte differentiation into Spp1+ macrophages in response to tumor conditioned medium. This is very interesting! Should this be further highlighted and discussed? This is in line with data presented in Figures 6B and S4A (although some important controls are missing – see comment below) indicating that blocking IL1-driven activation of enteric glial cells results in increased monocyte differentiation into C1Q+ macrophages. Should this also be further highlighted and discussed?

Minor: Figure 4G. IgG controls?

Figure 6A shows that data obtained using TME EGC-CM will be compared to H EGC-CM – which is elegant. However, the data obtained with H EGC-CM are not shown? Also, in Figures 6A, 6B, 6C and S4A, it seems essential to include the controls TME-CM ± IgG or anti-IL1R (and H-CM ± IgG or anti-IL1R) to prove that the ligands secreted by TME cells do not directly induce SPP1+ TAM differentiation via an IL1/IL1R axis and that IL-6 is not present in TME-CM (and H-CM) at levels comparable to TME EGC-CM (and H EGC-CM) (±IgG ±anti-IL1R).

Minor: Figure 6E shows Tdtomato+ cells originating from (lineage) tracing Sox10+ cells starting approximately one week before AOM injection. I do not understand the need for using this model here or showing this information here. Since the goal is to demonstrate spatial proximity of EGC and TAM, I recommend showing GFAP+ cells (not Tdt) and IBA1+ TAM using IF on either tumors from AOM-DSS-treated wild type mice, or, better in this figure, in tumors from AOM-DSS-treated GFAP-Cre;IL1R1fl/fl vs AOM-DSS-treated IL1R1fl/fl mice? This seems even more relevant as IL1R is knocked out in GFAP+ cells, not Sox10+ cells, and EGC have been shown to be heterogeneous (Baghdadi et al. 2022; PMID: 34727519) (I acknowledge that Figure S4B and S4E tend to show a perfect overlap between TdT (driven by Sox10-CreERT2 or GFAP-Cre), Sox10 and GFAP in normal colon tissue section). It would also answer the question: how the network of IL1R-deficient EGC in the TME looks like in comparison to WT TME EGC?

Figures 6E, 6I and 7A: I recommend co-staining with EpCAM (or equivalent) to allow for

visualization of the tumor.

Minor:

Page 10. Change “the reduce number of recruited monocyte and monocyte-derived macrophage in CCR2-/- tumors” to the reduced number of recruited monocytes and monocyte-derived macrophages in CCR2-/- tumors.

Page 11: Change “associated with a reduce Il6 gene expression” to associated with reduced Il6 gene expression.

Page 14: Do the authors mean: our data also point to a functional association between CRC EGCs and monocyte migration?

Reviewer #3 (Remarks to the Author): with expertise in tumor associated macrophages

This study elucidates a feed-forward circuit involving enteric glial cells (EGCs) and monocytes or tumor-associated macrophages (TAMs) in the context of colorectal cancer, ultimately promoting tumor growth. According to the author's findings, upon exposure to the colorectal tumor microenvironment, EGCs undergo a phenotypic shift, triggering immunomodulatory processes that foster the differentiation of SPP1+ TAMs via IL-6 secretion. In return, tumor-infiltrating monocytes exert an influence on the phenotype and function of CRC EGCs through the IL-1 signaling pathway. While this interplay between EGCs and TAMs mediated by the IL-1R/IL-6 axis represents a novel insight within the CRC context, it's worth noting that the significance of IL-1R signaling in EGCs has been extensively illustrated in intestinal inflammation by others (Schneider et al., 2022, Commun. Biol.), potentially limiting the novelty of this study.

To strengthen the impact and comprehensiveness of this research, several critical questions and suggestions have been raised:

Figure 1: It is suggested that TAM depletion should follow EGC depletion or EGC co-injection to definitively establish whether changes in TAMs are the dominant factor in this model.

Figure 2: Performing RNA-seq from sorted EGCs in both healthy tissues and tumor lesions

could provide a more comprehensive view of their gene expression profiles.

Figure 3: The study could benefit from in vivo IL-6 blockade, followed by an examination of alterations in SPP1+ TAMs. Additionally, it would be helpful to provide data on the expression levels of IL-6 and changes in SPP1+ TAMs before and after EGC depletion.

Figure 4H: To provide a more complete picture, consider comparing the mean fluorescence intensity (MFI) of IL-1 α /IL-1 β among different tumor-infiltrating cells.

Figure 5B: Given that tumor monocytes induced higher expression of EGC marker genes compared to BM-monocytes, consider investigating whether the expression level of IL-6 differs between these two cell types.

Figure 7G: To clarify the observed differences, it would be helpful to explain why the survival curve in Figure 7G is less pronounced compared to that in Figure 7B, even though both gene signatures are associated with EGCs.

Reviewer #4 (Remarks to the Author): with expertise in enteric glia

A series of recent studies have uncovered fundamental roles of enteric glial cells in the regulation of intestinal homeostasis and tissue response to injury or infection, which extend beyond the canonical neuroprotective and neuroregulatory roles of these cells. Despite a burgeoning literature on this topic, how enteric glial cells interact with cancer cells and how they respond to and regulate the tumor microenvironment and growth, is less well understood. The manuscript by van Baarle et al reports interesting experiments that aim to fill this gap in knowledge and provide mechanistic insight into how enteric glial cell-derived signals modulate the tumour immune microenvironment and drive the differentiation of tumour associated immune cells (such as macrophages). The data presented is clear, adequately powered and well analysed. Data interpretations are mostly correct, however, some important statements in the manuscript are not fully supported by the data and essential controls are lacking. Below are some of the main concerns that this reviewer feels they need to be addressed.

1. The authors use different tumour models: the MC38 murine colorectal cancer cells orthotopically transplanted into the colonic submucosa and the AOM-DSS model. Can the authors explain how these models are related and whether the molecular mechanism underlying tumour initiation and growth are similar between these models? Are these models interchangeable in terms of the microenvironment they induce and if not, what are their unique features? Are they expected to induce similar responses and engage enteric glial cells in a similar mechanistic way?

2. It is unclear what exactly happens in the orthotopic model of tumor cell transplantation. Several of the genes upregulated in response to MC38 cell transplantation have been observed in experimentally induced intestinal inflammation. It is therefore unclear whether the recorded response of glial cells is specific to the tumorigenic character of the transplanted cells or alternatively, represents a non-specific reaction of glial cells to the injury and the inevitable inflammatory response associated with the grafting of cells. How do glial cells react to the transplantation of other (tumorigenic or non-tumorigenic) cells? This is an important issue that the authors must consider and address.

3. In the case the Plp1CreERT2iDTR tumour model experiments, it appears that essential controls are lacking. Have the authors tested that the observed effects are independent of potential non-specific DT toxicity? What is the effect on tumor growth of DT injected into WT mice? Following local administration of DT to Plp1CreERT2iDTR mice it is expected that immune cells would be activated in response to the dying (glial) cells. Are the effects of DT administration on the tumor the result of a non-specific response of the immune system to injury and cell death, rather than a specific effect glial cell elimination?

4. The authors argue that EGCs display an activated and immunomodulatory phenotype in the context of CRC. This statement is based on analysing the effects of the tumour microenvironment on transcriptional profile of primary neurosphere-derived enteric glial cells. Given published evidence that EGCs depend on their tissue microenvironment to express key mature EGCs gene modules it is not clear whether neurosphere-derived enteric glial cells used here are (i) pure, (ii) mature and (iii) representative of their in vivo

counterparts? What are the criteria that the authors applied to decide that their primary neurosphere-derived enteric glial cells are suitable substitutes for mature adult EGCs? Further, the authors have chosen acute time-points to analyse EGC-reactivity to TME-CM in vitro (6, 12 and 24 hours). How do these time-points reflect the activation of EGCs in a tumour environment in vivo?

5. The authors state that enteric glia promote the differentiation of monocytes into pro-tumorigenic SPP1+ TAMs, however the impact of EGCs on monocyte-differentiation (Fig. 3A-C) has actually not been studied in the presence of EGCs. The authors only show an increase of SPP1+ TAMs in added presence of EGCs, without studying monocyte differentiation. Could the increase in SPP1+ TAMs simply be a reflection of increased monocyte numbers? The Plp1CreERT2iDTR tumour model, as well as the GFAPCreIL-1R1fl/fl AOM/DSS could serve as an alternative model to test whether specifically SPP1+ TAMs and/ or monocytes are reduced upon glia deletion.

We appreciate the time and effort you have dedicated to providing your valuable feedback and insightful comments, which helped us to improve the quality of the manuscript. We have incorporated new experiments and adaptations to reflect the suggestions provided by the reviewers. All the changes are underlined within the revised manuscript.

Here is a point-by-point response to the reviewers' comments and concerns.

Reviewer #1 (Remarks to the Author): with expertise in bioinformatics, cancer immunology

This study investigates the role of enteric glial cells (EGCs) in colorectal cancer (CRC) development. It primarily focuses on murine models of CRC, using both orthotopic tumor injections and in vitro co-culture with primary cells and multiple mutants to deplete particular cell types or molecules. The final figure extends the mouse results to analyse human patient cohorts to provide preliminary results that the same phenomenon occurs in humans and that it may be associated with prognosis. The primary conclusion is that there exists a positive feedback loop between monocytes/macrophages and EGCs contributing to poor prognosis CRC. Specifically, that a) monocytes and macrophages are the primary source of IL1 in the tumor microenvironment, b) IL1 acts on EGCs, inducing a protumoral/reactive phenotype, c) these reactive EGCs are the primary source of IL6 which drives monocytes towards a SPP1+ tumor associated macrophage phenotype which are known to drive CRC towards a bad prognosis "mesenchymal" state through diverse mechanisms. The manuscript is clearly written and easy to follow and the experiments are well designed. EGCs are currently underexplored in the literature so this study is a timely contribution.

Reply to reviewer: We thank the reviewer for their appreciation of our work and for their suggestions to improve our manuscript. Below is our point-by-point reply to their suggestions.

Major comments

1. The PLP1CreERT2iDTR mouse clearly depletes EGCs, however it is important to show that this is specific – ie that there is not leaky expression in other cell types which then also get depleted by DT injection.

Reply to reviewer: We have addressed this concern by incorporating an immunofluorescent staining (Supplementary Fig. 1c-d) of PLP1^{CreERT2}Ai14^{fl/fl} mice which demonstrates the specificity of PLP1Cre expression in glial cells, without any evidence of leaky expression in other cell types. Our findings indicate that tdTomato signaling coincides with the expression patterns of SOX10⁺ and GFAP⁺ cells, further supporting the specificity of PLP1Cre-mediated labeling in enteric glial cells.

Supplementary Figure 1c-d. Representative image of healthy muscularis tissue in PLP1^{CreERT2}Ai14^{fl/fl} mice showing tdTomato (magenta), SOX10 (green) and GFAP (blue), scale bar 100 μ m (c) and 25 μ m (d).

2. More details need to be provided about the survival analyses in Fig 7B,G to verify their robustness. Were the patients split by median values for the gene signatures? Or by some bi-modality of the gene signature distribution? If the (unfortunately common) practice of sweeping through splits to identify the value which gives the most significant difference in outcome was performed, this is not meaningful unless it is validated in a second, independent cohort.

Reply to reviewer: In response to the concerns raised by reviewer 1 regarding the methodology employed for the survival analysis of enteric glial cells (EGCs), we wish to clarify that in our analysis patients were not split by median values for the gene signatures. Instead, we employed an unsupervised clustering method. The patients were hierarchically clustered (R package: Pheatmap) based on their expression level of EGC specific genes, resulting in two clusters, one with high EGC gene expression and one with low expression of EGC specific genes. To validate our prediction, we would like to emphasize that these results were subsequently tested through an independent dataset, the Sidra-LUMC AC-ICAM dataset (Roelands J et al. Nat Med. 2023 PMID: 37202560). Similar for what obtained using the TCGA COAD-READ data, in this second independent dataset we again found distinct clustering of patients with EGC-high and EGC-low transcriptomic signatures. Additionally, also in the Sidra-LUMC AC-ICAM dataset EGC-high patients were correlated with low survival probability, thereby validating our previous findings in the TCGA dataset.

Regarding the "high gliosis" signature, the survival analysis for the second cohort has proven to not be statistically different. Given that the difference in this signature was not particularly prominent in the first dataset either, we are convinced that it would be more scientifically prudent to exclude it from the current manuscript. Instead, we will focus solely on reporting the data related to the overall EGC signature. This signature demonstrates a strong correlation with patient survival and has been validated in two independent human CRC datasets. The survival analysis of the second cohort is now included in the revised Supplementary Fig. 7c-d.

Supplementary Figure 7c-d: LUMC AC-ICAM derived CRC patients ($n=348$) stratified based on their expression of the EGCs signature genes ($n = 295$ EGCs low, $n = 53$ EGCs high). Heatmap of patients clustering (c) and Kaplan-Meier overall survival curve for EGCs high and low patients (d).

Figure Legend: LUMC AC-ICAM derived CRC patients ($n=348$) stratified based on their expression of the gliosis signature genes ($n = 236$ gliosis low, $n = 112$ gliosis high). Heatmap of patients clustering (a) and Kaplan-Meier overall survival curve for gliosis high and low patients (b). **Not included in the manuscript**

3. The authors nicely show that in vivo, monocytes and macrophages appear to be the major source of IL-1 in the tumor microenvironment. They also show that reactive EGCs produce IL-6 and that IL-6 drives monocytes/TAMs towards the SPP1+ TAM state. It would thus be important to determine whether reactive EGCs are the primary source of IL-6 in the TME or not.

Reply to reviewer: Although we agree that this is an important consideration, we would like to stress that our findings shown in revised figure 7e, in which a glial-specific knockout of IL-1R displayed a significant effect on the total IL-6 amount in the TME, which provides evidence to consider enteric glia as a major contributor to TME IL-6 amounts. Due to the complexity of the continuously changing TME and the fact that different cell subsets are potential IL-6 producers, proving one cell type as the one main producer of IL-6 is not feasible. However, to further elucidate the contribution of EGCs to TME IL-6 levels, we performed fluorescence-activated cell sorting (FACS) on naive colonic mucosal tissues and colonic tumor tissue of AOM/DSS-treated GFAP^{Cre}Ai14^{fl/fl} mice. Sorting efficiency was validated by increased S100 β expression in tdTomato^{pos} glial cell populations compared to tdTomato^{neg} cells in tumor and naive samples (Revised Supplementary Fig. 6g-h). Further underlining EGCs as major IL-6 producers, we detected a strong increase of *Il6* expression in tumor versus healthy tdTomato^{pos} EGCs

(Revised Fig. 7f). Although *Il6* expression also increased in the tumor $\text{tdTomato}^{\text{neg}}$ cells compared to naive, this increase did not reach significance. Of note, the *Il6* gene expression of tumor $\text{tdTomato}^{\text{pos}}$ EGCs being higher than tumor $\text{tdTomato}^{\text{neg}}$ cells, in fact, emphasizes EGCs as powerful IL-6 producers.

Figure 7f: $\text{GFAP}^{\text{Cre}}\text{Ai14}^{\text{fl/fl}}$ mice underwent the AOM/DSS model as described in Fig. 7a using 1% DSS or were kept under naive conditions. Mice were sacrificed, tumor or naive colon cells were isolated and FACS-sorted. Expression levels of *Il6* in sorted cells of naive and AOM/DSS-treated $\text{GFAP}^{\text{Cre}}\text{Ai14}^{\text{fl/fl}}$ mice, displaying *Il6* levels in $\text{tdTomato}^{\text{pos}}$ glial cells versus remaining $\text{tdTomato}^{\text{neg}}$ cells in both tumor and naive samples. Expression displayed as fold to *Rp/32* and relative to naive $\text{tdTomato}^{\text{neg}}$ cells. Data show mean \pm SEM. Statistical analysis: two-way ANOVA with Bonferroni's multiple comparison test * $p < 0.05$.

Supplementary Figure 6 g-h: $\text{GFAP}^{\text{Cre}}\text{Ai14}^{\text{fl/fl}}$ mice underwent the AOM/DSS model as described in Fig. 7a using 1% DSS or were kept under naive conditions. Mice were sacrificed, tumor or naive colon cells were isolated and FACS-sorted. **g** gating strategy of FACS sorting $\text{tdTomato}^{\text{pos}}$ and $\text{tdTomato}^{\text{neg}}$ cells. **h** Expression levels of *S100b* in sorted cells of naive and AOM/DSS-treated $\text{GFAP}^{\text{Cre}}\text{Ai14}^{\text{fl/fl}}$ mice, displaying *S100b* levels in $\text{tdTomato}^{\text{pos}}$ glial cells versus remaining $\text{tdTomato}^{\text{neg}}$ cells in both tumor and naive samples. Expression displayed as fold to *Rp/32* and relative to naive $\text{tdTomato}^{\text{neg}}$ cells. Data show mean \pm SEM. Statistical analysis: two-way ANOVA with Bonferroni's multiple comparison test * $p < 0.05$, ** $p < 0.005$.

Similarly, to the AOM/DSS model, we endeavored to isolate EGCs from the orthotopic tumor. However, the isolation of EGCs from mucosal tissue presents significant challenges, and to date, no research

group has successfully isolated large quantities of EGCs from the mucosa. In the context of the AOM/DSS model, we managed to isolate EGCs by pooling the mucosal tissues of several tumors per mouse and assessed a small subset of relevant genes (*Il6*, *S100b*, and *Rpl32*). However, in the case of the orthotopic model, our attempts to isolate glial cells from the tissue were unsuccessful, probably due to the lower amount of mucosal tissue gained from single orthotopic tumors. Yet, we confirmed the significance of glial-derived IL-6 in the CRC TME with an additional *in vivo* experiment (Revised Fig. 3j-l) utilizing orthotopic MC38 tumor injection in Sox10^{CreERT2}IL-6^{fl/fl} and wildtype littermate mice. Strikingly, glial-specific IL-6 knockout lead to a reduced tumor growth, which correlated with a reduction in SPP1⁺ TAMs. Together, our results highlight the importance of glial-derived IL-6 in both colonic tumor development and TAM differentiation *in vivo*.

Figure 3 j-l: SOX10^{CreERT2}IL-6^{wt/wt} and SOX10^{CreERT2}IL-6^{fl/fl} mice were intraperitoneally (i.p.) injected with tamoxifen at d-16 and d-14, followed by intracolonic (i.c.) injection at d0 with MC38 cells. Tumor growth and immune infiltration were assessed at d14 ($n = 10$). Schematic representation of orthotopic CRC mouse model (j) with representative pictures (scale bar 2 mm) and quantitative comparison of tumor volume ($n = 10$) (k). Data show absolute tumor-infiltrating TAM cell numbers per mg tumor tissue (l). Data are represented as mean \pm SEM (k-l). Statistical analysis: unpaired t-test (k-l) * $p < 0.05$, ns not significant.

Minor comments

1. In Fig 2B the separation of TME-CM from Unstimulated and H-CM is very striking. As a complement to the WGCNA analysis in panels C and D, it would be nice to show the genes with the biggest (positive or negative) contribution to PC2, eg in the supplemental.

Response to reviewer: Following the reviewer's suggestion, to better understand the genes contributing to the separation of TME-CM from unstimulated and H-CM in Revised Figure 2b, we specifically chose samples stimulated for 24 hours as they are indicative of the significant impact of the tumor microenvironment on EGC gene expression. Principal Component Analysis (PCA) was performed on these selected samples, where in the new PCA plot the TME-CM samples are separated from unstimulated and H-CM EGCs in PC1. We analysed the genes contributing to PC1 and interestingly, the elevation of several chemokines and cytokines positively contributes to the segregation of the EGC samples along the PC1 axis. This includes *Cxcl5*, *Ccl7*, *Cxcl1*, *Ccl2*, and *Il6*, suggesting an immunomodulatory role for EGCs within the tumor microenvironment. Additionally, numerous genes associated with tumor pathogenesis, such as *Mmp3*, *Mmp9*, and *Saa3*, were identified as significant contributors to PC1. Furthermore, genes linked to glial cell activation, such as *Lcn2* and *Timp1*, were among those exhibiting a positive contribution to PC1. Conversely, genes associated with the homeostatic functions of EGCs, including muscle contraction (*Acta2* and *Myh11*), and calcium ion transport (*Ntsr1*), were identified as negative contributors to PC1. We included this additional analysis in revised Supplementary Fig. 3a-b.

The full list of genes contributing to PC1 is provided as Supplementary Table 2.

Supplementary Figure 3a-b: **a** Transcriptome analysis of in vitro primary embryonic neurosphere-derived EGCs alone or stimulated with healthy conditioned medium (H-CM) or tumor microenvironment conditioned medium (TME-CM) at different time points (24h, $n = 4$). Principal component analysis (PCA) plot of EGCs gene signature identified by 3'mRNA bulk RNA-seq. Each dot represents an individual sample. **b** Barplot showing genes with positive and negative contributions towards PC1 of the PCA plot with in vitro primary embryonic neurosphere-derived EGCs alone or stimulated with healthy conditioned medium (H-CM) or tumor microenvironment conditioned medium (TME-CM) at 24h time points ($n = 4$).

2. The selection of genes for figure 2D is not entirely clear. The methods says that WGCNA modules were restricted to have between 30 and 5000 genes so this must be a subset of modules 4, 7 and 8. How were they chosen. Doing a gene set enrichment analysis of modules 4, 7, and 8 seems straightforward and potentially more informative than highlighting/interpreting just 2 genes per module as is done in the text on page 7.

Reply to reviewer: The genes displayed in Revised Figure 2 represent a selection guided by their similarity to the human signature (revised Fig. 8) and their functional significance, particularly emphasizing the active state of EGCs (evidenced by *Lcn2* and *Timp1*) and their immunomodulatory functions (including *Cxcl10*, *Ptges*, *Ccl2*, and *Il6*). In line with the reviewer's suggestion to better define the possible functions of the genes in the various modules, we generated a bar graph shown below to illustrate the terms of interest identified through gene set enrichment analysis for modules 4, 7, and 8 and included this as revised Supplementary Fig. 3c.

Supplementary Figure 3c: Transcriptome analysis of *in vitro* primary embryonic neurosphere-derived EGCs alone or stimulated with healthy conditioned medium (H-CM) or tumor microenvironment conditioned medium (TME-CM) at different time points (6h, 12h, and 24h (n = 4)). Barplot showing the gene set enrichment analysis for the transcriptional modules 4, 7 and 8 identified by weighted gene correlation network analysis (WGCNA) (all terms have adjusted p value <0.05).

Reviewer #2 (Remarks to the Author): with expertise in enteric glia, CRC

This is a very compelling study demonstrating that, in response to monocytes of the tumor microenvironment, enteric glial cells undergo a switch of phenotype that influences tumor-associated macrophages differentiation. The identification of the cells of the enteric nervous system as key components of the tumor microenvironment functionally controlling colon cancer development and progression is a major emerging research area in the field of colon cancer. This paper provides powerful evidence that enteric glial cells engage in functional interactions with monocytes and macrophages of the tumor microenvironment, shaping the cellular composition of the latter. The manuscript is very well written and has a nice flow. The experimental design is creative and pertinent, and the data are very convincing and nicely presented. My only major recommendation is to further characterize the data obtained using the murine orthotopic colon cancer model in DT-treated PLP1DTA mice versus controls. Other minor concerns are detailed and listed below.

Reply to reviewer: We thank the reviewer for their appreciation of our work and for their suggestions to improve our manuscript. Below is our point-by-point reply to their suggestions.

1. The murine orthotopic colon cancer model needs to be better presented and/or characterized. Basically, my major concern is whether or not the difference in tumor volume between DT-treated PLP1DTA mice and controls shown in Fig 1C originates from a difference in tumor epithelial cell proliferation or in immune cell recruitment, as this would change the interpretation of this experiment. This concern is furthered by Fig S1 that shows a major immune infiltrate but not a tumor (epithelial) mass. I would recommend co-staining orthotopic tumors with EpCAM (or equivalent), GFAP, F4/80 and a marker of proliferation. In addition, using a bioluminescent or fluorescent reporter stably expressed in tumor epithelial cells would allow for the discrimination of grafted tumor epithelial cells versus host colonic epithelial cells and prove expansion of grafted MC38 cells. While Fig S1B shows the proportion of immune cells vs other cells in orthotopic tumors vs healthy tissue, this does not demonstrate whether the difference in tumor size between DT-treated PLP1DTA mice and controls comes from tumor epithelial cell proliferation or immune cell recruitment. Other questions are: was Fig S1B flow

cytometry? were PLP1DTA control mice injected with saline? Figure S1C shows a decrease in GFAP expression in DT-treated PLP1DTA mice at day 0 (when MC38 cells are injected) which is five days after DT injection. Baghdadi et al. (2022; PMID: 34727519) found a small decrease in GFAP expression at one week after PLP1+ cell ablation and an upregulation of GFAP in PLP1DTA mice after two weeks of tamoxifen treatment, suggesting that there is a major increase in GFAP expression between week 1 and week 2. As only day 0 is shown here, it is difficult to interpret whether or not the growth of the orthotopic tumors may have been concomitant with an increase in GFAP+ expressing cells? While the findings from Baghdadi and colleagues are acknowledged in the manuscript, this needs to be further investigated. Can GFAP expression be investigated at day 7? Can other markers of enteric glia be tested to validate these findings?

Reply to reviewer: To address the reviewers concern whether the decrease in tumor volume is related to a decrease in tumor epithelial cell proliferation or in immune cell recruitment, we used two different approaches. Firstly, we followed the reviewer's suggestion of co-staining orthotopic tumors with markers of tumor cells, macrophages and a marker of proliferation to visualise the proliferating tumor cells in this model. Co-staining revealed double positive KI67⁺/EPCAM⁺ cells in tumor regions. Of note, IBA1⁺ macrophages were located within tumor regions and in proximity to KI67⁺ cells, but not expressing KI67 themselves. Secondly, we further conducted a more in-depth analysis of the flow cytometric data presented in Revised Fig. 1b-d. Our findings reveal a significant decrease in the absolute numbers of both immune and non-immune cells per milligram of tumor tissue within the tumor microenvironment in the absence of glial cells. Notably, despite this decrease, the composition of the tumor microenvironment remains unchanged, as evidenced by the consistent percentage of immune and non-immune cells. Thus, it is evident that the reduction in tumor size is not solely attributable to a decline in immune cells. (not included in the manuscript).

Regarding the question towards Revised Supplementary Fig. 1b we confirm that this figure was performed by using flow cytometry. Furthermore, to clarify on the PLP1-iDTR control mice, in our study, the PLP^{CreERT2}iDTR "Vehicle" group always received 100 µl saline solution, whereas the "DT" group was injected in the colonic mucosa with 40 ng of Diphtheria toxin diluted in 100 µl of saline.

As noted by the reviewer, Baghdadi et al. reported that PLP1^{CreERT2}-DTA depletion results in a substantial reduction of GFAP⁺ EGCs at 7 days, which is in line with our observations. We conducted a time course study to elucidate the duration of GFAP⁺ EGC reduction following DT injection. Our findings revealed that in healthy tissue, GFAP⁺ EGCs remained depleted at least until day 7 post-injection, with markedly reduced GFAP expression at the injection site in DT-treated mice compared to Vehicle-treated counterparts (not included in the manuscript). Consistently, in tumor, GFAP expression was notably diminished in DT-treated mice compared to those receiving vehicle treatment at day 7 (Revised Supplementary Fig. 1i). Moreover, we observed reduced RNA levels of *S100b* and *Sox10* in DT-treated tumors compared to Vehicle-treated counterparts at day 7 (Revised Supplementary Fig. 1j).

Altogether, these findings confirm that the orthotopic MC38 model in PLP1^{CreERT2}iDTR mice is a valuable model to study the immunomodulatory effects of EGCs depletion in CRC.

Figure legend. **a** Immunofluorescent images of an MC38 orthotopic tumor showing co-staining of EPCAM (grey), IBA1 (cyan) and Ki67 (red) (scale bar 50 μ m). **b-d** PLP1^{CreERT2}iDTR mice were intraperitoneally (i.p.) injected with tamoxifen at d-19 and d-17, followed by intracolonic (i.c.) injection at d-5 and d-3 with 40 ng Diphtheria toxin (DT) or saline (Vehicle). At d0, MC38 cells were i.c. injected in both groups. Data show absolute numbers per mg tumor tissue (**b**) and % of live cells (**c**) for immune (CD45+) and non-immune (CD45-) cells on d7. ($n = 13$ Vehicle, $n = 12$ DT). **d** EGC marker expression was assessed at d0, d4 and d7. Representative western blots for GFAP and Vinculin. Data show mean \pm SEM (b,c). Statistical analysis: unpaired Mann-Whitney test (b-c) * $p < 0.05$, ns not significant. **Not included in the manuscript.**

Supplementary Figure 1i-j: PLP1^{CreERT2}iDTR mice were intracolonicly injected at d-5 and d-3 with 40 ng Diphtheria toxin (DT) or saline (Vehicle). At d0, MC38 cells were i.c. injected in both groups, EGC

marker expression was assessed at d7. Representative western blots for GFAP and Vinculin (i) and relative mRNA levels for *S100b* and *Sox10* normalized to the housekeeping gene *Rpl32* at d7 (n = 4 Vehicle, n = 5 DT) (j). Data show mean ± SEM. Statistical analysis: Unpaired t-test (j) *p < 0.05.

2. The study uses both embryonic neurosphere-derived enteric glial cells and adult neurosphere-derived enteric glial cells, and it is unclear why one type of culture is used over the other. A rationale should be provided. Why not consistently use adult neurosphere-derived enteric glial cells -which seem more appropriate in the context of colon cancer?

Reply to reviewer: This study, conducted in collaboration with Professor Wehner's group from Bonn University Hospital, builds upon both groups' previously established *in vitro* models of enteric glial cells, as reported in publications by M. Stakenborg et al. (2022, Mucosal Immunology) and R. Schneider et al. (2021, EMBO Molecular Medicine). It is particularly reassuring to observe that both adult and embryonic EGC cultures exhibit identical phenotypes following IL-1 stimulation, as depicted in revised Fig. 4d-e and Supplementary Fig. 5b-c. This consistency in the IL-1-driven phenotype across both primary EGC models is considered a significant strength of our study. The neurosphere-derived enteric glial cell culture relies on the expansion of precursor cells residing in the enteric nervous system, so we do not expect intrinsic differences in adult or embryonic derived cultures. To test for possible difference in the differentiation pattern between the two cultures, we characterised their transcriptomic signatures, confirming similar expression patterns of glial markers (*Plp1*, *Sox10*, and *S100b*), and minimal to no expression of markers for other cell types, such as epithelial cells (*Epcam* and *Cdh1*), smooth muscle cells (*Prkg1* and *Foxp2*), endothelial cells (*Cdh5* and *Pecam1*), neurons (*Nefl* and *Syp*), mesenchymal cells (*Pdgfra* and *Wt1*), and immune cells (*Ikzf1* and *Itgam*), suggesting a very high purity degree in both EGC culture models. Furthermore, consistent with recent findings by A. Laddach et al. (Nat. Comms., 2023), the expression of *S100β*, indicative of mature EGCs, was observed in both EGC cultures (not included in the manuscript).

Figure legend: Heatmap illustrating the expression levels of marker genes associated with various cell types, including Enteric glial cells (*Plp1*, *Sox10*, and *S100b*), epithelial cells (*Epcam* and *Cdh1*), smooth muscle cells (*Prkg1* and *Foxp2*), endothelial cells (*Cdh5* and *Pecam1*), neurons (*Nefl* and *Syp*), mesenchymal cells (*Pdgfra* and *Wt1*), and immune cells (*Ikzf1* and *Itgam*), in unstimulated samples collected at 6 hours, 12 hours, and 24 hours from bulk RNA-seq data of embryonic neurosphere-

derived EGCs ($n = 4$) (a) and adult neurosphere-derived EGCs ($n = 9$) (b). **Not included in the manuscript.**

Minor: Figure 3H shows that enteric glial cells activate monocyte differentiation into C1Q+ macrophages in response to healthy conditioned medium while they activate monocyte differentiation into Spp1+ macrophages in response to tumor conditioned medium. This is very interesting! Should this be further highlighted and discussed? This is in line with data presented in Figures 6B and S4A (although some important controls are missing – see comment below) indicating that blocking IL1-driven activation of enteric glial cells results in increased monocyte differentiation into C1Q+ macrophages. Should this also be further highlighted and discussed?

Reply to reviewer: Following reviewer's suggestion we have adapted the results section and discussion accordingly.

Minor:

1. Figure 4G. IgG controls?

Reply to reviewer: IgG controls were used when no IL-1R antibody was added as indicated in Revised Fig. 4f.

2. Figure 6A shows that data obtained using TME EGC-CM will be compared to H EGC-CM – which is elegant. However, the data obtained with H EGC-CM are not shown? Also, in Figures 6A, 6B, 6C and S4A, it seems essential to include the controls TME-CM \pm IgG or anti-IL1R (and H-CM \pm IgG or anti-IL1R) to prove that the ligands secreted by TME cells do not directly induce SPP1+ TAM differentiation via an IL1/IL1R axis and that IL-6 is not present in TME-CM (and H-CM) at levels comparable to TME EGC-CM (and H EGC-CM) (\pm IgG \pm anti-IL1R).

Reply to reviewer: We appreciate the reviewer for bringing this to our attention. We have now included the H-CM EGCs stimulated monocytes in the figures as requested (revised Fig. 6a-d and Supplementary Fig. 6a-b). Notably, the blockade of IL-1R led to a significant reduced differentiation of monocytes into SPP1⁺ TAMs with the supernatant of TME-CM EGCs but did not alter the TAM polarization in the context of H-CM EGCs both at RNA and protein level (revised Fig. 6a-c and Supplementary Fig. 6a-b). As a result, IL-6 levels were markedly reduced in the supernatant of TME-CM-exposed EGCs following IL-1R inhibition, while no difference was found in H-CM-exposed EGCs following IL-1R inhibition (revised Fig. 6d). Regarding the proposed addition of extra TME-CM and H-CM controls, we would like to clarify our methodology: we add the antibodies during the EGCs stimulation step and thereafter remove the antibody from the EGC supernatant using magnetic beads. Therefore, comparing TME-CM + IgG with TME-CM + anti-IL-1R would not yield meaningful results, as the antibodies would be removed after an incubation with the medium, without being in contact with any cell. We have rephrased the results paragraph to provide clarity on the experimental procedure, which may have caused confusion. Additionally, we updated the schematic figure illustrating the experimental design accordingly.

Figure 6 a-d: Murine bone marrow-derived monocytes were cultured for 48h with supernatant of primary embryonic neurosphere-derived EGCs, which were pre-incubated for 24h with tumor microenvironment conditioned medium (TME-CM) or healthy-CM (H-CM) together with isotype IgG or anti-IL-1R (5 μ g/mL each). Antibodies were removed from the CM through the application of Dynabeads™ Protein G prior to incubation with monocytes. **a** Experimental design. **b-c** Relative mRNA levels for *Arg1*, *Spp1* and *C1qa*, normalized to the housekeeping gene *Rpl32*, in monocytes cultured with supernatant of TME-EGCs ($n = 7$) (**b**) or H-EGCs ($n = 4$) (**c**). (**d**) IL-6 concentration in the conditioned medium of H EGCs and TME EGCs pre-incubated with IgG or anti-IL-1R ($n = 3$). Data are represented as mean \pm SEM. Statistical analysis: paired Wilcoxon test (b-c) or two-way ANOVA test with correction for multiple comparisons (d). * $p < 0.05$, *** $p < 0.0005$, **** $p < 0.00005$, ns not significant.

Supplementary Figure 6a-b: Murine bone marrow-derived monocytes were cultured for 48h with supernatant of primary embryonic neurosphere-derived EGCs, which were pre-incubated for 24h with tumor microenvironment conditioned medium (TME-CM) or healthy-CM (H-CM) together with isotype IgG or anti-IL-1R (5 μ g/mL each). Antibodies were removed from the CM through the application of Dynabeads™ Protein G prior to incubation with monocytes. FACS quantification of SPP1⁺ TAMs and C1Q⁺ TAMs after TME-EGCs ($n = 7$) (**a**) or H-EGCs ($n = 4$) (**b**) supernatant stimulation. Statistical analysis: paired Wilcoxon test (a-b). * $p < 0.05$, ns not significant.

3. Minor: Figure 6E shows Tdtomato+ cells originating from (lineage) tracing Sox10+ cells starting approximately one week before AOM injection. I do not understand the need for using this model here or showing this information here. Since the goal is to demonstrate spatial proximity of EGC and TAM, I recommend showing GFAP+ cells (not Tdt) and IBA1+ TAM using IF on either tumors from AOM-DSS-treated wild type mice, or, better in this figure, in tumors from AOM-DSS-treated GFAP-Cre;IL1R1fl/fl vs AOM-DSS-treated IL1R1fl/fl mice? This seems even more relevant as IL1R is knocked out in GFAP+ cells, not Sox10+ cells, and EGC have been shown to be heterogeneous (Baghdadi et al. 2022; PMID: 34727519) (I acknowledge that Figure S4B and S4E tend to show a perfect overlap between Tdt (driven by Sox10-CreERT2 or GFAP-Cre), Sox10 and GFAP in normal colon tissue section). It would also answer the question: how the network of IL1R-deficient EGC in the TME looks like in comparison to WT TME EGC?

Reply to reviewer: We thank the reviewer for pointing out the potentially confusing use of a different mouse lines in this purpose. Our intention was indeed to demonstrate the spatial proximity of EGCs and TAMs in the TME. As we previously observed a reduced fluorescent intensity for stained EGCs compared to reporter expressing EGCs, we initially used the Sox10^{CreERT2}Ai14^{fl/fl} line. However, we understand that using a different glial marker here, is not ideal and reproduced these images using GFAP^{Cre}Ai14^{fl/fl} mice, now also including EPCAM as a tumor marker (Revised Fig. 7b). We further addressed the reviewer's second question of comparing glial networks in GFAP^{Cre}IL-1R1^{fl/fl} vs GFAP^{WT}IL-1R1^{fl/fl} mice by immunohistochemistry and found a similar EGC network morphology at the tumor base in both genotypes (not included in the manuscript).

Figure 7b: GFAP^{Cre}Ai14^{fl/fl} mice were subjected to the AOM/DSS model as described in Fig. 7a using 1% DSS per cycle. Representative images of EPCAM (white), IBA1 (green) and GFAPtdTomato (magenta) in tumor sections at d 70 (scale bar 100 μ m).

GFAP^{Cre}IL-1R1^{fl/fl}

GFAP^{Wt}IL-1R1^{fl/fl}

Figure legend: GFAP^{Wt}IL-1R1^{fl/fl} and GFAP^{Cre}IL-1R1^{fl/fl} were subjected to the AOM/DSS model as described in Fig. 7a. Representative immunofluorescence stainings of EPCAM (white), IBA1 (green) and GFAP (magenta) in tumor sections at d 70 (scale bar 100 μ m). **Not included in the manuscript.**

4. Figures 6E, 6I and 7A: I recommend co-staining with EpCAM (or equivalent) to allow visualization of the tumor.

Reply to reviewer: We adapted the figures accordingly in revised Fig. 7b and Fig. 8a.

5. Minor:

Page 10. Change “the reduce number of recruited monocyte and monocyte-derived macrophage in CCR2^{-/-} tumors” to the reduced number of recruited monocytes and monocyte-derived macrophages in CCR2^{-/-} tumors.

Page 11: Change “associated with a reduce Il6 gene expression” to associated with reduced Il6 gene expression.

Page 14: Do the authors mean: our data also point to a functional association between CRC EGCs and monocyte migration?

Reply to reviewer: We have adapted the text in the manuscript accordingly.

Reviewer #3 (Remarks to the Author): with expertise in tumor associated macrophages

This study elucidates a feed-forward circuit involving enteric glial cells (EGCs) and monocytes or tumor-associated macrophages (TAMs) in the context of colorectal cancer, ultimately promoting tumor growth. According to the author's findings, upon exposure to the colorectal tumor microenvironment,

EGCs undergo a phenotypic shift, triggering immunomodulatory processes that foster the differentiation of SPP1⁺ TAMs via IL-6 secretion. In return, tumor-infiltrating monocytes exert an influence on the phenotype and function of CRC EGCs through the IL-1 signaling pathway. While this interplay between EGCs and TAMs mediated by the IL-1R/IL-6 axis represents a novel insight within the CRC context, it's worth noting that the significance of IL-1R signaling in EGCs has been extensively illustrated in intestinal inflammation by others (Schneider et al., 2022, Commun. Biol.), potentially limiting the novelty of this study.

Reply to reviewer: As stated by the reviewer we are convinced that our research substantially expands the understanding of the role of enteric glial cells in the pathogenesis of colon carcinoma. Significantly, our report pioneers this field by being the first to identify enteric glial cells as essential components of the neuroimmune niche. This niche fosters the accumulation of tumor-supportive macrophages via the IL-1R/IL-6 axis, an interaction previously unexplored in this context. Overall, we are convinced that our findings are novel and go far beyond our previous observations about the role of IL-1R in glial activation in non-oncological gut inflammation (Schneider et al 2022, Comm Biol).

Our study identifies an association between the gene signature of enteric glial cells and the reduced overall survival in patients with CRC, representing an important advance in the understanding of CRC pathogenesis. Our data show both *in vitro* and *in vivo* that, upon IL-1 stimulation, enteric glial cells acquire a reactive and highly immunomodulatory phenotype in the colonic tumor microenvironment and secrete IL-6, which in turn promotes differentiation of monocytes into pro-tumorigenic SPP1⁺ tumor macrophages. Although IL-1 β had already been described in glial cell activation, the release of IL-6 leading to SPP1⁺ macrophage accumulation constitutes a novel mechanism of disease progression. Furthermore, our discoveries extend beyond the current knowledge of glial cells in colorectal cancer, as possible interactions between various types of glial cells and immune cells have been proposed in lung and pancreatic tumors. Overall, our findings suggest an evolutionarily conserved mechanism that provides a survival advantage to tumor cells.

Furthermore, we thank the reviewer for their suggestions to improve our manuscript, which we have addressed in our point-by-point reply.

Figure 1: It is suggested that TAM depletion should follow EGC depletion or EGC co-injection to definitively establish whether changes in TAMs are the dominant factor in this model.

Response to reviewer: Several papers have already demonstrated that SPP1⁺ TAMs play a significant pro-tumorigenic role in CRC (Zhang et al., 2020; Cheng et al., 2021; Li et al., 2023). Thus, performing double or consecutive depletion of SPP1⁺ TAMs and EGCs in the CRC injection model seems slightly out of scope. We are convinced that such an experiment would not substantially alter the current message of our study. Furthermore, optimizing the appropriate mouse models and obtaining ethical approval would significantly delay the revision of this manuscript.

Figure 2: Performing RNA-seq from sorted EGCs in both healthy tissues and tumor lesions could provide a more comprehensive view of their gene expression profiles.

Response to reviewer: We agree with the reviewer that performing bulk or scRNA-seq on the tumor and healthy EGCs would increase our understanding of the phenotype of the cells in the context of CRC. However, isolation of EGCs from the mucosa has proven to be very challenging and to our knowledge no research group has successfully isolated large quantities of EGCs only from the mucosa, with sufficient quality for bulk or scRNA-seq. However, in the context of the AOM/DSS model, we managed to isolate sufficient mucosal EGCs in GFAP-tdTomato reported mice to assess the expression of a small subset of genes relevant to our study (*Il6*, *S100b*, and *Rpl32*). In line with our hypothesis, we

could confirm that tumor-associated EGCs produce increased levels of IL-6 compared to the ones in the healthy mucosa. Our attempts to isolate glial cells in the orthotopic model failed, likely due to insufficient tissue from a single tumor, unlike the multiple tumors in the AOM/DSS model that allow for successful cell isolation. The new data of sorted EGCs from naive tissues and AOM/DSS tumors have been incorporated in Revised Fig. 7f and Supplementary Fig. 6 g-h.

Figure 7f: GFAP^{Cre}Ai14^{fl/fl} mice underwent the AOM/DSS model as described in Fig. 7a using 1% DSS or were kept under naive conditions. Mice were sacrificed, tumor or naive colon cells were isolated and FACS-sorted. Expression levels of *Il6* in sorted cells of naive and AOM/DSS-treated GFAP^{Cre}Ai14^{fl/fl} mice, displaying *Il6* levels in tdTomato^{pos} glial cells versus remaining tdTomato^{neg} cells in both tumor and naive samples. Expression displayed as fold to *Rpl32* and relative to naive tdTomato^{neg} cells. Data show mean ± SEM. Statistical analysis: two-way ANOVA with Bonferroni's multiple comparison test *p<0.05.

Supplementary Figure 6 g-h: GFAP^{Cre}Ai14^{fl/fl} mice underwent the AOM/DSS model as described in Fig. 7a using 1% DSS or were kept under naive conditions. Mice were sacrificed, tumor or naive colon cells were isolated and FACS-sorted. **g** Gating strategy of FACS sorting tdTomato^{pos} and tdTomato^{neg} cells. **h** Expression levels of *S100b* in sorted cells of naive and AOM/DSS-treated GFAP^{Cre}Ai14^{fl/fl} mice, displaying *S100b* levels in tdTomato^{pos} glial cells versus remaining tdTomato^{neg} cells in both tumor and naive samples. Expression displayed as fold to *Rpl32* and relative to naive tdTomato^{neg} cells. Data show mean ± SEM. Statistical analysis: two-way ANOVA with Bonferroni's multiple comparison test *p<0.05,

**p<0.005.

Figure 3: The study could benefit from *in vivo* IL-6 blockade, followed by an examination of alterations in SPP1⁺ TAMs. Additionally, it would be helpful to provide data on the expression levels of IL-6 and changes in SPP1⁺ TAMs before and after EGC depletion.

Reply to Reviewer: Following the reviewer's suggestion of *in vivo* IL-6 blockade, we incorporated a new experiment in our study (Revised Fig. 3j-l) utilizing orthotopic MC38 tumor injection in Sox10^{CreERT2}IL-6^{wt/wt} and wildtype littermate mice. Strikingly, glial-specific IL-6 knockout lead to reduced tumor growth, which correlated with a reduction in SPP1⁺TAMs while no effect was detected for C1Q⁺ TAMs. These new results highlight the importance of glial-derived IL-6 in both colonic tumor development and TAM differentiation *in vivo*.

Regarding the reviewer's second point concerning our EGC depletion model, we initiate the depletion of EGCs prior to inducing tumor growth. Consequently, it becomes unfeasible to evaluate alterations in SPP1⁺ TAMs before and after EGC depletion.

Figure 3 j-l: SOX10^{CreERT2}IL-6^{wt/wt} and SOX10^{CreERT2}IL-6^{fl/fl} mice were intraperitoneally (i.p.) injected with tamoxifen at d-16 and d-14, followed by intracolonic (i.c.) injection at d0 with MC38 cells. Tumor growth and immune infiltration were assessed at d14 ($n = 10$). Schematic representation of orthotopic CRC mouse model (j) with representative pictures (scale bar 2 mm) and quantitative comparison of tumor volume ($n = 10$) (k). Data show absolute tumor-infiltrating TAM cell numbers per mg tumor tissue (l). Data are represented as mean \pm SEM (k-l). Statistical analysis: unpaired t-test (k-l) * $p < 0.05$, ns not significant.

Figure 4H: To provide a more complete picture, consider comparing the mean fluorescence intensity (MFI) of IL-1 α /IL-1 β among different tumor-infiltrating cells.

Reply to reviewer: Following the reviewer's suggestion, we compared the MFI of IL-1 α and IL-1 β among the different TME cell types, resulting in a similar outcome as shown for cell frequency, validating highest MFI in monocytes and monocyte-derived macrophages. We believe that the frequency of IL-1 α ⁺ and IL-1 β ⁺ cells is a more representative way of displaying this result and therefore did not include this additional presentation in our manuscript.

Figure legend: WT C57BL/6J mice were intracolonicly injected at day(d)0 with MC38 cells, and both stromal and immune cells were assessed for IL-1 β and IL-1 α expression at d21. Data are presented as Mean fluorescent intensity (MFI) of IL-1 α and IL-1 β in TME cells. ($n = 5$). **Not included in the manuscript.**

Figure 5B: Given that tumor monocytes induced higher expression of EGC marker genes compared to BM-monocytes, consider investigating whether the expression level of IL-6 differs between these two cell types.

Reply to reviewer: We are not sure if we understood the reviewer's question correctly. We think they might want to elaborate on the source of IL-6 in the TME, which we addressed by collecting more information about EGCs as a major producer of IL-6. By sorting of tdTomato^{pos} EGCs from naive and AOM/DSS-treated GFAP^{Cre}Ai14^{fl/fl} mice, we show increased *Il6* expression in tumor EGCs compared to healthy EGCs. This increased expression was of similar, even slightly higher levels than the tdTomato^{neg} remaining TME cells, strengthening the hypothesis of EGCs as a major but not the only IL-6 producer in the colon TME. The data were already mentioned regarding the reviewer's comment to Figure 2 as well as in the revised figures (Fig. 7f and Supplementary Fig. 6g-h).

Additionally, we confirmed the significance of glial-derived IL-6 in the CRC TME with a glial-specific IL-6 knockout *in vivo* (Revised Fig. 3j-l). We used the orthotopic MC38 tumor injection in SOX10^{CreERT2}IL-6^{fl/fl} and wildtype littermate mice and showed that glial-specific IL-6 knockout lead to reduced tumor growth, which correlated with a reduction in SPP1⁺TAMs. The data were already mentioned regarding the reviewer's comment to Figure 3 as well as in the revised figure (Fig. 3j-l).

Figure 7G: To clarify the observed differences, it would be helpful to explain why the survival curve in Figure 7G is less pronounced compared to that in Figure 7B, even though both gene signatures are associated with EGCs.

Reply to reviewer: Despite both signatures being linked to EGCs, it is essential to note the distinction between them. Figure 7B (revised Fig. 8b) depicts a signature based on well-established EGC signature genes, which are independent of disease or health state. In contrast, the survival curve presented in Figure 7G correlates with the expression of genes induced by gliosis in EGCs. To strengthen our findings regarding EGC gene signatures in human CRC, we validated our results in a second independent dataset, known as the Sidra-LUMC AC-ICAM dataset (Roelands J. et al. Nat Med. 2023 PMID: 37202560). Similar to the TCGA COAD-READ data, we found distinct clustering of patients with EGC^{high}

and EGC^{low} transcriptomic signatures (Revised Supplementary Fig. 7c). Additionally, also here EGC-high patients were correlated with worse survival probability, thereby confirming the TCGA data (Revised Supplementary Fig. 7d). Regarding the "high gliosis" signature, the survival analysis for the second cohort has proven to not be statistically different. Given that the difference in this signature was not particularly prominent in the first dataset either, we are convinced that it would be more scientifically prudent to exclude it from the current manuscript. Thus, we will be reporting only the data related to the overall EGC signature, as this has demonstrated a strong correlation with patient survival and has been validated in two independent human CRC datasets. The second cohort is now included in the revised Supplementary Fig. 7c-d.

Supplementary Figure 7c-d: LUMC AC-ICAM derived CRC patients (n=348) stratified based on their expression of the EGCs signature genes (n = 295 EGCs low, n = 53 EGCs high). Heatmap of patients clustering (c) and Kaplan-Meier overall survival curve for EGCs high and low patients (d).

Figure Legend: LUMC AC-ICAM derived CRC patients (n=348) stratified based on their expression of the gliosis signature genes (n = 236 gliosis low, n = 112 gliosis high). Heatmap of patients clustering (a) and Kaplan-Meier overall survival curve for gliosis high and low patients (b).

Reviewer #4 (Remarks to the Author): with expertise in enteric glia

A series of recent studies have uncovered fundamental roles of enteric glial cells in the regulation of intestinal homeostasis and tissue response to injury or infection, which extend beyond the canonical neuroprotective and neuroregulatory roles of these cells. Despite a burgeoning literature on this topic, how enteric glial cells interact with cancer cells and how they respond to and regulate the tumor microenvironment and growth, is less well understood. The manuscript by van Baarle et al reports interesting experiments that aim to fill this gap in knowledge and provide mechanistic insight into how enteric glial cell-derived signals modulate the tumour immune microenvironment and drive the differentiation of tumour associated immune cells (such as macrophages). The data presented is clear, adequately powered and well analysed. Data interpretations are mostly correct, however, some important statements in the manuscript are not fully supported by the data and essential controls are lacking. Below are some of the main concerns that this reviewer feels they need to be addressed.

Reply to reviewer: We thank the reviewer for their appreciation of our work and for their suggestions to improve our manuscript. Below is our point-by-point reply to their suggestions.

1. The authors use different tumour models: the MC38 murine colorectal cancer cells orthotopically transplanted into the colonic submucosa and the AOM-DSS model. Can the authors explain how these models are related and whether the molecular mechanism underlying tumour initiation and growth are similar between these models? Are these models interchangeable in terms of the microenvironment they induce and if not, what are their unique features? Are they expected to induce similar responses and engage enteric glial cells in a similar mechanistic way?

Reply to reviewer: Colorectal cancer encompasses a multifaceted spectrum of diseases, each presenting distinct etiological factors, clinical manifestations, and therapeutic considerations. Therefore, several subtypes of colorectal cancers have been identified and representative murine models have been established. Within this study we have made use of 3 different mouse models;

In regard to the AOM/DSS model, representative of the colitis-associated CRC, the persistent inflammatory microenvironment in colitis provides an optimal niche for the accumulation of genetic mutations and epigenetic changes, ultimately resulting in the development of colitis-induced CRC.

The MC38 model is a representative orthotopic “immune hot” tumor model, characterized by a strong anti-tumor immune infiltrate (Efremova M. et al. *Nat. Commun.*, 2018) which we used as an additional immune-related CRC model allowing us to study the role of EGCs in two different immune-related CRC models.

Finally, the AKPT model is a representative orthotopic “immune cold” MSS-CRC tumor model. This model is characterized by a very poor and tumor-supportive immune contexture (Jackstadt R. et al. *Cancer Cell*, 2019 and Beach C. et al. *Clin. Invest.*, 2023). The AKPT cell line is originally derived from tumors originating from villinCre^{ER} Apc^{fl/fl} Kras^{G12D/+} Trp53^{fl/fl} Trgfbr1^{fl/fl} mice, these mice develop spontaneous tumors with mutations in APC, KRAS, P53 and the TGF- β pathway (Jackstadt R. et al. *Cancer Cell*, 2019), all mutations with high clinical relevance in CRC.

Overall, our findings demonstrate that ablating IL-1R in EGCs impeded tumor growth and SPP1⁺ TAM expansion in all of these models, suggesting IL-1R triggered EGC TAM crosstalk as a universal mechanism across the different CRC subtypes. We thank the reviewer for indicating that the reasoning behind the different models was not clear and we have now incorporated more explanation regarding the murine models also in the results section.

2. It is unclear what exactly happens in the orthotopic model of tumor cell transplantation. Several of the genes upregulated in response to MC38 cell transplantation have been observed in experimentally induced intestinal inflammation. It is therefore unclear whether the recorded response of glial cells is specific to the tumorigenic character of the transplanted cells or alternatively, represents a non-specific reaction of glial cells to the injury and the inevitable inflammatory response associated with the grafting of cells. How do glial cells react to the transplantation of other (tumorigenic or non-tumorigenic) cells? This is an important issue that the authors must consider and address.

Reply to reviewer: The interplay between inflammation and tumor development has already been demonstrated in many tumor contexts, especially in CRC. Thus, it is not surprising to us that the interaction between EGCs and TAMs, involving ligand-receptor signaling also used in the context of inflammation, plays a significant role. As demonstrated in our study the role of EGCs in CRC development is also taking place in AOM/DSS, a model not relying on injection of cells into the mucosa, but representative of colitis-associated cancer. To further solidify our findings in the revised manuscript we have also employed a second less immunogenic (MMS CRC) primary tumor cell line originally generated from tumors with mutating oncogenes such as Apc, Kras, Trp53 and Trgfbr1, the AKPT cell line (Jackstadt R. et al. *Cancer Cell*, 2019), to ascertain whether glial cells respond similarly to

this injection compared to MC38 transplantation. Different from the MC38 “immune hot” tumor model, characterised by a strong anti-tumor immune infiltrate (Efremova M. et al. Nat. Commun., 2018), these AKPT tumors represent an “immune cold” tumor model. This model is characterized by a very poor and tumor-supportive immune infiltrate (Jackstadt R. et al. Cancer Cell, 2019 and Beach C. et al. Clin. Invest., 2023).

Both cell lines underwent testing in murine models with deficient IL-1R signaling in EGCs, revealing that the immunomodulatory functions of EGCs were uniformly affected across both cell lines and similarly to the AOM/DSS model. Note the differences in absolute immune cells numbers between the “immune-hot” MC38 model and the “immune-cold” AKPT model. These findings have been incorporated into the results section and are presented in Revised Fig. 6e-j.

Figure 6 e-j: e-g WT C57BL/6J mice were intracolonic (i.c.) injected with MC38 cells and embryonic neurosphere-derived WT EGCs or IL-1R1^{-/-} EGCs (1:1 ratio). Tumor growth and immune infiltration were assessed at day (d)14. Schematic representation of EGCs co-injection mouse model (e) with representative pictures (scale bar 2 mm) and quantitative comparison of tumor volume ($n = 7$) (f). Data show absolute tumor-infiltrating TAM cell numbers per mg tumor tissue ($n = 7$ MC38 + WT EGCs, $n = 6$ MC38 + IL-1R1^{-/-} EGCs) (g). h-j GFAP^{Wt}IL-1R1^{fl/fl} and GFAP^{Cre}IL-1R1^{fl/fl} mice were i.c. injected with AKPT cells. Tumor growth and immune infiltration were assessed at d14. Schematic representation of orthotopic CRC mouse model (h) with representative pictures (scale bar 2 mm) and quantitative comparison of tumor volume ($n = 12$ GFAP^{Wt}IL-1R1^{fl/fl}, $n = 8$ GFAP^{Cre}IL-1R1^{fl/fl}) (i). Data show absolute tumor-infiltrating TAM cell numbers per mg tumor tissue ($n = 11$ GFAP^{Wt}IL-1R1^{fl/fl}, $n = 6$ GFAP^{Cre}IL-1R1^{fl/fl}) (j). Data are represented as mean \pm SEM. Statistical analysis: unpaired Mann-Whitney test (f-g, i-j). *p < 0.05, ** p < 0.005, ns not significant.

3. In the case the Plp1CreERT2iDTR tumour model experiments, it appears that essential controls are lacking. Have the authors tested that the observed effects are independent of potential non-specific DT toxicity? What is the effect on tumor growth of DT injected into WT mice? Following local administration of DT to Plp1CreERT2iDTR mice it is expected that immune cells would be activated in response to the dying (glial) cells. Are the effects of DT administration on the tumor the result of a non-specific response of the immune system to injury and cell death, rather than a specific effect glial cell elimination?

Reply to reviewer: To address the concerns regarding potential non-specific DT toxicity or unspecific immune activation, we conducted an experiment utilizing WT C57BL/6J mice following the same injection protocol used for PLP-iDTR mice. Specifically, WT C57BL/6J mice were intracolonicly injected with 40 ng of DT or saline solution on days -5 and -3. On day 0, MC38 cells were injected into both groups. After allowing 7 days for tumor growth, we assessed tumor weight and immune infiltration. Our findings revealed no discernible difference in tumor growth or immune infiltration between DT- and saline-injected WT mice. These results underscore the absence of non-specific DT toxicity in this model (Not included in the manuscript). To ensure that our results regarding tumor growth and immune infiltration remain unaffected by immune activation in response to dying glial cells, we administered DT to deplete the EGCs on days -5 and -3. This approach should ensure resolution of inflammation caused by cell death by day 0. Indeed, at day 0, when EGCs were depleted and the tumor injection was initiated, no discernible difference was observed in the immune response between DT- and saline-treated mice (revised Supplementary Fig. 1e and h). Therefore, we can confidently exclude the possibility that the observed immune response in our tumor model at day 7 is a result of a non-specific reaction of the immune system to cell death. Instead, it is specifically attributed to the elimination of glial cells. We have incorporated these data in revised Supplementary Fig. 1e and h.

Figure legend: a-c WT C57BL/6J mice were injected intracolonicly (i.c.) at d-5 and d-3 with 40 ng DT or saline. At d0, MC38 cells were i.c. injected in both groups. Tumor growth and immune infiltration were assessed at d7. Schematic representation of tumor mouse model (a) with quantitative comparison of tumor volume (b). Data show absolute tumor-infiltrating immune cell numbers per mg tumor tissue (c) ($n = 6$ Vehicle, $n = 7$ DT). Data are represented as mean \pm SEM. Statistical analysis: One-way ANOVA test with correction for multiple comparisons (c) or unpaired Mann-Whitney test (b). ns not significant. **(Not included in the manuscript)**

Supplementary Figure 1e and h. (e-h) PLP1^{CreERT2};DTR mice were intraperitoneally (i.p.) injected with tamoxifen at day(d)-19 and d-17, followed by intracolonic (i.c.) injection at d-5 and d-3 with 40 ng Diphtheria toxin (DT) or saline (Vehicle). Schematic representation of EGCs depletion mouse model (**e**). Absolute numbers of immune cells at the site of injection per mg of colon tissue at d0 ($n = 4$ Vehicle, $n = 4$ DT) (**h**). Data are represented as mean \pm SEM. Statistical analysis: One-way ANOVA test with correction for multiple comparisons (h). ns not significant.

4.1 The authors argue that EGCs display an activated and immunomodulatory phenotype in the context of CRC. This statement is based on analysing the effects of the tumour microenvironment on transcriptional profile of primary neurosphere-derived enteric glial cells. Given published evidence that EGCs depend on their tissue microenvironment to express key mature EGCs gene modules it is not clear whether neurosphere-derived enteric glial cells used here are (i) pure (ii) mature and (iii) representative of their *in vivo* counterparts? What are the criteria that the authors applied to decide that their primary neurosphere-derived enteric glial cells are suitable substitutes for mature adults EGCs

Reply to reviewer: To assess the purity of both neurosphere-derived EGCs, we conducted a comparative analysis of the expression levels of well-known EGC markers (Plp1, Sox10, and S100 β) in addition to markers for potential contaminating cell types such as epithelial cells (Epcam and Cdh1), smooth muscle cells (Prkg1 and Foxp2), endothelial cells (Cdh5 and Pecam1), neurons (Nefl and Syp), mesenchymal cells (Pdgfra and Wt1), and immune cells (Ikzf1 and Itgam) in both embryo-derived and adult-derived neurosphere cultures. Our results demonstrated high expression of EGC markers with minimal expression of markers for other cell types, suggesting a pure culture of EGCs in both models. Furthermore, consistent with recent findings by A. Laddach et al. (Nat. Comms., 2023), the expression of S100 β , indicative of mature EGCs, was observed in both EGC cultures (not included in the manuscript).

Finally, we believe the most compelling evidence supporting the fidelity of our primary EGC cultures to their *in vivo* counterparts is our ability to replicate our *in vitro* findings regarding the IL-1R/IL6 immunomodulatory axis of EGCs *in vivo*.

Figure legend: Heatmap illustrating the expression levels of marker genes associated with various cell types, including Enteric glial cells (Plp1, Sox10, and S100β), epithelial cells (Epcam and Cdh1), smooth muscle cells (Prkg1 and Foxp2), endothelial cells (Cdh5 and Pecam1), neurons (Nefl and Syp), mesenchymal cells (Pdgfra and Wt1), and immune cells (Ikzf1 and Itgam), in unstimulated samples collected at 6 hours, 12 hours, and 24 hours from bulk RNA-seq data of embryonic neurosphere-derived EGCs (*n* = 4) (a) and adult neurosphere-derived EGCs (*n* = 9) (b). **Not included in the manuscript.**

Materials and Methods: To quantify the Transcripts per million (TPM) values from the bulk RNA-seq data of adult derived EGCs, PartekFlow software available from <https://www.partek.com/partek-flow/#features> was used. The TPM values for bulk RNA-seq of embryonic neurosphere-derived EGCs were obtained by processing the fastq files using nf-core/rnaseq pipeline (nf-co.re/rnaseq/3.12.0) where GRCh37 was used as reference genome and Salmon was used for quantification (PMID: 32055031).

4.2 Further, the authors have chosen acute time-points to analyse EGC-reactivity to TME-CM in vitro (6, 12 and 24 hours). How do these time-points reflect the activation of EGCs in a tumour environment in vivo?

Reply to reviewer: We appreciate the reviewer's feedback. In our in vitro experimental design, we chose acute time-points due to evidence from the literature indicating the rapid responsiveness of EGCs to changes in their environment. For example, in postoperative ileus (POI), EGCs are recognized as among the first responders, initiating chemokine secretion as early as 3 hours post-injury (M. Stakenborg et al., Mucosal Immunol., 2022, Leven et al., J Neuroinflammation 2023). This swift response of EGCs to microenvironmental cues is reflected in their epigenetic profiles, which exhibit open chromatin structures within immunomodulatory genes (A. Laddach et al., Nat. Comms., 2023). Considering this body of evidence, we selected acute time-points to assess EGC reactivity to the TME-CM.

5. The authors state that enteric glia promote the differentiation of monocytes into pro-tumorigenic SPP1+ TAMs, however the impact of EGCs on monocyte-differentiation (Fig. 3A-C) has actually not been studied in the presence of EGCs. The authors only show an increase of SPP1+ TAMs in added presence of EGCs, without studying monocyte differentiation. Could the increase in SPP1+ TAMs simply be a reflection of increased monocyte numbers? The Plp1CreERT2iDTR tumour model, as well as the

GFAPCreIL-1R1^{fl/fl} AOM/DSS could serve as an alternative model to test whether specifically SPP1+ TAMs and/or monocytes are reduced upon glia deletion.

Reply to reviewer: In line with the reviewer's comment, we acknowledge the possibility that EGCs may also play a role in attracting monocytes to the tumor site. Indeed, our findings demonstrate that EGCs secrete high levels of Ccl2, a chemoattractant for monocytes, upon exposure to the tumor environment in IL-1R dependent fashion (Revised Fig. 4f-g). Additionally, in the Plp1^{CreERT2}iDTR tumor model, we observed a significant decrease in monocytes in EGC-depleted tumors compared to controls (revised Fig. 1d). However, this effect was not observed in the co-injection model (revised Fig. 1g). Furthermore, in the AOM/DSS model in GFAP^{Cre}IL-1R1^{fl/fl} mice, we observed a significant decrease in monocytes in mice depleted of glial IL-1R compared to control mice (not included in the manuscript).

We propose here that, in addition to their possible role in monocyte attraction, EGCs are involved in the differentiation of monocytes into SPP1⁺ TAMs, as we demonstrated *in vitro* by stimulating bone marrow-derived monocytes with TME-CM EGCs supernatant (revised Fig. 3 g-h). Notably, this effect was significantly decreased by adding anti-IL6 antibody to the TME EGC-CM, highlighting the importance of glial-derived IL-6 in SPP1⁺ TAM differentiation. Furthermore, we also demonstrated *in vivo*, that EGC-derived IL-6 is needed for EGC-induced SPP1⁺ TAM differentiation (Revised Fig. 3j-l). Moreover, monocytes were unaltered in glial-specific IL-6 depleted tumors (not included in the manuscript), while we still observed a significant decrease in SPP1⁺ TAMs compared to wildtype tumors, in contrast C1Q⁺ TAM numbers remained unaltered (Revised Fig. 3l). These result highlight that independent of EGC-induced monocyte attraction (via Ccl2), tumor EGCs promote SPP1⁺ TAMs differentiation in an IL-6 dependent fashion.

Figure legend: **a** GFAP^{Wt}IL-1R1^{fl/fl} and GFAP^{Cre}IL-1R1^{fl/fl} mice were subjected to the AOM/DSS model as described in Fig. 7a. Corresponding absolute numbers of monocytes per mg tumor tissue assessed at d70 (n = 7 GFAP^{Wt}IL-1R1^{fl/fl}, n = 9 GFAP^{Cre}IL-1R1^{fl/fl}). **b** SOX10^{CreERT2}IL-6^{wt/wt} and SOX10^{CreERT2}IL-6^{fl/fl} mice were intraperitoneally (i.p.) injected with tamoxifen at d-16 and d-14, followed by intracolonic (i.c.) injection at d0 with MC38 cells. Corresponding absolute numbers of monocytes per mg tumor tissue assessed at d14. Data are represented as mean ± SEM (a-b). Statistical analysis: unpaired Mann-Whitney test (a) and unpaired t-test (b) **p < 0.005, ns not significant. **Not included in the manuscript.**

REVIEWERS' COMMENTS

Reviewer #1 (Remarks to the Author):

The authors have done an excellent job in response to my earlier report. The manuscript is much improved and I have no more questions or comments.

Reviewer #2 (Remarks to the Author):

This is a revised version of a manuscript exploring the consequences of the molecular interactions between monocytes/macrophages and enteric glial cells on colon carcinogenesis. The authors have been overall extremely responsive to the reviewers' critiques and have added critical data that strengthen their findings. The multiple models and experimental systems are highly convincing in demonstrating the validity of the identified mechanism. However, I still think that the orthotopic model should be more comprehensively presented/discussed – see details below. Also, I think that some of the data presented in the rebuttal and labeled as “not included in the manuscript” should actually be included in the manuscript – see below for details (I can only speak for the figures that pertain to my critiques, but I strongly think that this is also true for some data generated to answer other reviewers' critiques).

Comments on the orthotopic model: The authors now show no change in the proportion of CD45+ vs CD45- within the tumors depleted or not of enteric glial cells, but a decrease in absolute numbers/mg of tumor for both populations in enteric glia-KO tumors vs controls. Their data also show that the MC38 based orthotopic model is mostly composed of CD45+ cells (~50,000 cells/mg of tumor) while CD45- cells (including epithelial cells and enteric glial cells) are a minor component (~5,000 cells/mg of tumor).

(1) How is it possible to measure absolute numbers/mg?

(2) While I agree with the authors that the decrease in CD45- cells suggests a decrease in epithelial cells, I also think that this may reflect the decrease in enteric glial cells (as a result of DT treatment). Measuring EpCAM+ cells would have been more convincing.

(3) If both CD45+ and CD45- populations roughly both decrease by 50% after DT in total numbers/mg of tissue, then what makes the rest of the tumor, dead cells?

(4) I still think that showing, using IF, in the same specimen, enteric glia, macrophages and epithelial cells would be better than showing glia and macrophages (Fig 1h) or macrophages and epithelial cells (rebuttal figure).

(5) I encourage the authors to include the data shown in the rebuttal figure into the manuscript as I believe that it is important to define the cellular composition of the orthotopic tumors.

In addition, I recommend highlighting in the text the similar molecular responses to IL-1 in primary adult and embryonic enteric glial cell cultures to justify the interchangeable use of both systems throughout the study, as well as including the gene signature heatmap shown in the rebuttal as a supplemental figure. I also recommend showing the IF illustrating the density and distribution of enteric glial cells and macrophages in tumors from GFAP-Cre;IL1R^{fl/fl} vs controls in a supplementary figure as it seems important to show that the disruption of the IL1R gene does not seem to impact the organization and density of the enteric glial cell network.

Reviewer #4 (Remarks to the Author):

The authors have invested considerable effort in addressing the concerns and comments raised by this and the other reviewers. Through a series of additional experiments, including an additional tumour model and a glial-specific ablation of IL6, the incorporation of new data has significantly strengthened the manuscript. Overall, the reviewer is impressed with the improvements made and believes that the manuscript is now in excellent shape for publication.

One minor comment: in view of the 3Rs, the reviewer strongly suggests including the experiment that addresses the concerns regarding potential non-specific DT toxicity/ unspecific immune activation (comment 3) in the supplementary figures as it represents an important control that will benefit subsequent studies in the field aiming to use similar ablation approaches.

Reviewer's Comments:

Reviewer #1 (Remarks to the Author)

The authors have done an excellent job in response to my earlier report. The manuscript is much improved and I have no more questions or comments.

Reply: *We appreciate the reviewer's positive feedback and acknowledgment of our efforts in addressing the previous comments and improving the manuscript.*

Reviewer #2 (Remarks to the Author)

This is a revised version of a manuscript exploring the consequences of the molecular interactions between monocytes/macrophages and enteric glial cells on colon carcinogenesis. The authors have been overall extremely responsive to the reviewers' critiques and have added critical data that strengthen their findings. The multiple models and experimental systems are highly convincing in demonstrating the validity of the identified mechanism. However, I still think that the orthotopic model should be more comprehensively presented/discussed – see details below. Also, I think that some of the data presented in the rebuttal and labeled as “not included in the manuscript” should actually be included in the manuscript – see below for details (I can only speak for the figures that pertain to my critiques, but I strongly think that this is also true for some data generated to answer other reviewers' critiques).

Reply: *We thank the reviewer for their positive feedback on our revision and for acknowledging our responsiveness to the critiques. We appreciate the constructive suggestions regarding the orthotopic model and the inclusion of additional data. Please find a point-by-point answer to the detailed comments provided below.*

Comments on the orthotopic model: The authors now show no change in the proportion of CD45+ vs CD45- within the tumors depleted or not of enteric glial cells, but a decrease in absolute numbers/mg of tumor for both populations in enteric glia-KO tumors vs controls. Their data also show that the MC38 based orthotopic model is mostly composed of CD45+ cells (~50,000 cells/mg of tumor) while CD45- cells (including epithelial cells and enteric glial cells) are a minor component (~5,000 cells/mg of tumor).

(1) How is it possible to measure absolute numbers/mg?

Reply: *To measure the absolute numbers per milligram of tumor tissue, we first determine the tumor weight of the tissue sample used for flow cytometry staining. By incorporating counting beads into the flow cytometry analysis, we can accurately quantify the number of cells. The combination of tumor weight measurement and counting beads enables us to calculate the absolute cell numbers per milligram of tumor tissue.*

(2) While I agree with the authors that the decrease in CD45- cells suggests a decrease in epithelial cells, I also think that this may reflect the decrease in enteric glial cells (as a result of DT treatment). Measuring EpCAM+ cells would have been more convincing.

Reply: *We understand the reviewer's concern regarding the interpretation of the decrease in CD45- cells. However, using our current and published protocols for isolating cells from the orthotopic tumor, we are unable to effectively isolate and stain enteric glial cells. Thus, it is currently experimental challenging to precisely quantify enteric glial cells in the tumor microenvironment.*

(3) If both CD45+ and CD45- populations roughly both decrease by 50% after DT in total numbers/mg of tissue, then what makes the rest of the tumor, dead cells?

Reply: We observe a general decrease in the total number of cells per milligram of tumor tissue following DT treatment (a), while we do not observe any difference in dead cell numbers (b). In our view, this reduction of total cells reflects a diminished infiltration, accumulation, or proliferation of both CD45+ and CD45- cell populations. Consequently, the overall decrease in cell density accounts for the observed reduction in both populations without implying a significant increase in dead cells within the tumor tissue.

Figure legend: *PLP1^{CreERT2};DTR* mice were intraperitoneally (i.p.) injected with tamoxifen at d-19 and d-17, followed by intracolonic (i.c.) injection at d-5 and d-3 with 40 ng Diphtheria toxin (DT) or saline (Vehicle). At d0, MC38 cells were i.c. injected in both groups. Data show absolute numbers per mg tumor tissue for total cells (a) and dead cells (b) on d7. (n = 13 Vehicle, n = 12 DT). Data show mean \pm SEM. Statistical analysis: unpaired Mann-Whitney test * $p < 0.05$, ns not significant. **Not included in the manuscript.**

(4) I still think that showing, using IF, in the same specimen, enteric glia, macrophages and epithelial cells would be better than showing glia and macrophages (Fig 1h) or macrophages and epithelial cells (rebuttal figure).

Reply: We appreciate the reviewer's suggestion regarding the inclusion of immunofluorescence staining for enteric glia, macrophages, and epithelial cells in the same specimen. While we acknowledge the potential value of such a comprehensive staining, we would like to emphasize that the orthotopic MC38 model is widely used in the field, with established evidence of tumor cell presence in orthotopic tumors (E. Zigmund, et al. 2011).

Given the specific focus of our study on the interaction between enteric glial cells and macrophages, we prioritized the inclusion of these cell types in the staining process to ensure clarity and relevance to our research objectives. Incorporating additional cell types, such as epithelial cells, proved extremely difficult, primarily due to secondary antibody incompatibility. Refining the staining protocol would require a significant time investment and testing of various antibody combinations, without significantly enhancing the primary message of our study. Additionally, setting up a new tumor experiment solely for this purpose would substantially delay the manuscript revision process and might not be in line with the principles of the 3Rs.

(5) I encourage the authors to include the data shown in the rebuttal figure into the manuscript as I believe that it is important to define the cellular composition of the orthotopic tumors.

Reply: We appreciate the appraisal to our experimental efforts and agreed to include those new data into the manuscript. We will incorporate the relevant data into the manuscript to further enhance its comprehensiveness and robustness.

(6) In addition, I recommend highlighting in the text the similar molecular responses to IL-1 in primary adult and embryonic enteric glial cell cultures to justify the interchangeable use of both systems throughout the study, as well as including the gene signature heatmap shown in the rebuttal as a supplemental figure. I also recommend showing the IF illustrating the density and distribution of enteric glial cells and macrophages in tumors from GFAP-Cre;IL1R^{fl/fl} vs controls in a supplementary figure as it seems important to show that the disruption of the IL1R gene does not seem to impact the organization and density of the enteric glial cell network.

Reply: We agree with the reviewer's point and will incorporate the recommendations in the revised manuscript, addressing both the text and the figure.

Reviewer #4 (Remarks to the Author)

The authors have invested considerable effort in addressing the concerns and comments raised by this and the other reviewers. Through a series of additional experiments, including an additional tumour model and a glial-specific ablation of IL6, the incorporation of new data has significantly strengthened the manuscript. Overall, the reviewer is impressed with the improvements made and believes that the manuscript is now in excellent shape for publication.

Reply: We appreciate the reviewer's acknowledgment of the efforts we have invested in addressing their concerns and incorporating additional experiments to strengthen our manuscript. We are grateful for their insightful comments and constructive feedback, which have undoubtedly contributed to enhancing the quality of our work.

One minor comment: in view of the 3Rs, the reviewer strongly suggests including the experiment that addresses the concerns regarding potential non-specific DT toxicity/ unspecific immune activation (comment 3) in the supplementary figures as it represents an important control that will benefit subsequent studies in the field aiming to use similar ablation approaches.

Reply: In agreement with reviewer 4, we have decided to incorporate the rebuttal figures into the manuscript to ensure their availability for reference and reproducibility.